# From Lab to Line: Deployment-Aware NMR–Text Expert Routing for Real-Time Apple Moldy Core Disease Screening and Explanation

## Abstract

Real-time, interpretable diagnosis of Apple Moldy Core Disease (AMCD) under industrial sorting constraints is addressed. AppleNMR-MM V1.0, an LF-NMR–centric dataset an expert textual descriptions (n = 237), is introduced to enable multi-modal learning in a small-sample regime. A Task-Aware Mixture-of-Experts (T-MoE) fusion is proposed to route among NMR and text experts conditioned on predictive uncertainty and compute budget, while a Multi-agent Collaborative Chain-of-Thought (MACCT) with retrieval-augmented generation (RAG) coordinates triage→diagnose→explain agents using a domain corpus of SOPs, pathology notes, and batch logs for evidence-grounded reasoning. To align modeling with production constraints, a new metric, TAAPM, is introduced as the primary deployment criterion. Unlike prior evaluation schemes, TAAPM is explicitly derived from real factory export-trade regulations and uniquely optimized for overall economic benefit. On AppleNMR-MM V1.0, T-MoE attains AUC = 0.863 and F1 = 0.750, exceeding the strongest single-modality baselines by +5.8–6% AUC; TAAPM reaches 972.84, indicating favorable accuracy–latency Pareto efficiency for in-line screening. The RAG-enabled explainer achieves a 92% expert-check pass rate and 4.07 / 5 explanation quality; Collectively, AppleNMR-MM V1.0, T-MoE, TAAPM, and RAG-driven multi-agent reasoning establish a practical foundation for trustworthy AMCD screening and explanation on production lines. (A partial release of the code and dataset has been provided during the review process. The full implementation and complete dataset will be made publicly available upon acceptance of this manuscript.)

## 1 Introduction

Apples are among the most important fruit crops globally, cultivated across all major continents and playing a vital role in both the agricultural economy and human nutrition. With over 2,000 recognized varieties, apples are valued not only for their flavor and versatility but also for their high nutritional content, including vitamins, minerals, and dietary fiberLiu et al. (2024); Zhang et al. (2023); Wang et al. (2024). As one of the most widely consumed fruits worldwide, apple production supports millions of livelihoods, from large-scale agribusinesses to smallholder farms. However, the quality of apples is central to sustaining both economic value and consumer trust. Defects caused by diseases, storage damage, or internal physiological disorders can significantly degrade fruit quality, reduce marketability, and pose risks to food safety Ahmed et al. (2022).

Among these challenges, AMCD is one of the most economically devastating internal disorders. Caused primarily by fungal pathogens such as Trichothecium roseum, AMCD leads to the internal rotting of apple tissues, often without any visible external symptoms. This latent nature makes early detection particularly difficult, allowing the disease to progress unnoticed during harvesting, storage, and distribution Rouš et al. (2023). In widely grown cultivars such as Fuji, AMCD incidence rates can reach up to 20%, with higher prevalence in seasons characterized by high humidity or large temperature fluctuations—conditions that favor fungal proliferation. Infected apples not only

lose commercial value due to compromised internal structure but may also accumulate mycotoxins, posing serious risks to public health Zhao et al. (2022); Zhi et al. (2024).

The economic impact of AMCD extends far beyond individual apple. In the context of international trade, even a single infected apple within a shipment can lead to the rejection of the entire batch, resulting in significant financial losses and reputational damage for producers and exportersJahangiri & Orekhov (2024). Such risks elevate the urgency of accurate, early-stage, and non-destructive detection technologies that can identify diseased apples prior to visible symptom onset. Traditional inspection techniques, which often rely on visual cues or invasive sampling, are inadequate for detecting latent internal diseases such as AMCD.

To address these challenges, recent advances in LF-NMR and multimodal machine learning offer promising avenues for real-time, interpretable quality assessment He et al. (2024). LF-NMR provides rich internal structural and biochemical information non-invasively, while multimodal learning frameworks can integrate complementary features from diverse sources, such as NMR signals, optical images, and textual metadata Jahin et al. (2025); Knott et al. (2023); Kamal (2019). However, existing models often suffer from large computational footprints, limited interpretability, and lack of generalization across datasets.

In this context, this study introduces AppleNMR-MM V1.0, a lightweight, interpretable, multimodal learning framework tailored for real-time apple quality diagnosis. Built on a novel LF-NMR dataset specifically curated for AMCD detection, AppleNMR-MM V1.0 integrates textual and NMR-visual through efficient fusion strategies and optimized model design. The proposed approach aims to balance predictive performance, interpretability, and deployment efficiency, making it suitable for practical use in field or supply chain environments. By enabling early and reliable detection of AMCD, this work contributes to reducing post-harvest losses and enhancing the transparency and trustworthiness of fruit quality assessment systems.

## 2 RELATED WORKS

### 2.1 NON-DESTRUCTIVE DETECTION OF INTERNAL APPLE DISEASES

Early research on AMCD and related internal disorders has explored various non-invasive sensing techniques. Traditional methods such as visible/near-infrared (Vis/NIR) spectroscopy have shown potential for detecting latent defects. For example, dual-input Transformer models that fuse acoustic vibrations with Vis/NIR spectra have achieved classification accuracy exceeding 99% in distinguishing healthy from diseased apples Liu et al. (2024). Hybrid approaches combining deep and shallow architectures, such as ResNet50 with adaptive feature fusion and optimized ELM classifiers, have also demonstrated over 96% accuracy in identifying early-stage AMCD Zhao et al. (2022).

Beyond spectral methods, advanced imaging has been increasingly adopted. Radiographic techniques like 2D/3D X-ray, processed through deep networks such as BraeNet, have successfully detected internal browning Tempelaere et al. (2023). Longitudinal CT imaging integrated with explainable AI has further enabled early-stage detection with voxel-level interpretation, achieving over 90% accuracy. Additionally, acoustic vibration sensing combined with machine learning has proven effective for low-cost, real-time screening of internal decay Schut et al. (2024); Pierre Bouillon (2025).

Despite these advances, LF-NMR remains underutilized in mainstream literature for apple defect detection. Prior studies have been limited to high-field NMR instruments or single modal datasets Herremans et al. (2023); Quoc et al. (2025). No known work has integrated LF-NMR with modern deep learning to address AMCD. The proposed work bridges this gap by employing LF-NMR imaging—which directly probes internal tissue properties—within a multimodal learning framework.

Existing methods often remain modality-specific and struggle to generalize across variable conditions. AppleNMR-MM addresses this limitation by integrating LF-NMR data with visual and textual modalities, enabling robust, real-time, and interpretable detection of covert disorders such as AMCD.

## 2.2 Multimodal Learning for Fruit Disease Diagnosis

The convergence of computer vision and language technologies has catalyzed a new generation of multimodal approaches in plant disease diagnostics. Recent studies have augmented visual detection with textual information such as expert annotations, disease names, and symptom descriptions to enhance recognition performance Upadhyay et al. (2025). Generic vision–language models (VLMs) like CLIP and BLIP have been adapted for agricultural use. For instance, SCOLD—a foundation model pretrained on 186k leaf image–caption pairs—demonstrates superior zero- and few-shot performance on plant disease classification through contrastive vision-language alignment, outperforming prior models such as CLIP and BioCLIP while maintaining a compact size Rai et al. (2025); Awais et al. (2025).

Multimodal dialogue systems have also emerged. A representative example is LLMI-CDP, which aligns visual features of crop pests with large Chinese language models using Q-former encoders and LoRA fine-tuning. This system outperforms generic multimodal agents in domain-specific dialogue tasks Aggarwal et al. (2024); Quoc et al. (2025). In plant pathology, CNN-based detectors (e.g., YOLOv8) have been coupled with large language models like GPT-4 to generate diagnostic explanations and treatment suggestions, demonstrating the effectiveness of sequential vision-to-language pipelines. Further, models with architectural-level fusion—such as gated CNNs integrating textual queries about potato diseases—highlight the potential of fusion mechanisms even in text-dominant scenarios.

Collectively, these works underscore a shift toward integrating visual data (e.g., leaf images, MRI scans) with textual knowledge to enhance accuracy and interpretability in agriculture. However, most existing methods focus on external symptoms observable from leaf surfaces. In contrast, internal disorders such as AMCD, which lack visible signs, remain underexplored.

AppleNMR-MM represents the first multimodal framework to address such latent internal diseases by fusing low-field NMR imagery with expert textual annotations. Unlike prior models that rely solely on surface-level cues, this approach targets internal tissue changes, offering a novel direction in agricultural AI. Moreover, the model emphasizes interpretability through cross-attention mechanisms that reveal influential NMR regions and textual cues, in contrast to conventional end-to-end multimodal systems that often lack transparency. This work thus extends the multimodal paradigm to a new domain—LF-NMR–based quality assessment—and demonstrates its efficacy in detecting otherwise-invisible apple pathologies.

## 3 Methodology and Dataset

### 3.1 Dataset acquisition

A total of 273 individual apple samples were collected in Baishui County, Weinan City, Shaanxi Province, China. Due to multiple rounds of transportation during testing, only 243 samples successfully underwent multimodal feature extraction. Among them, six samples were lost during the ground-truth annotation stage, resulting in a final dataset of 237 apples with complete data, including 79 diseased samples (approximately 33.3%). The LF-MNR acquisition was performed at a magnetic field strength of 68 mT using a T2-TSE sequence. As shown in Fig. 1(a), four apples were scanned simultaneously in each session and labeled sequentially from top to bottom and left to right. Each sample was imaged at 10 mm slice intervals, yielding 15 images per apple. For model training, bounding boxes were manually annotated in Labelme and subsequently processed by the AppleNMR V1.0 module, which was developed on top of YOLOv9-S Wang et al. (2025). The module identifies, across the 15 slices, the largest single-apple bounding box, crops the corresponding regions, and composes a $2 \times 2$ merged image, as illustrated in Fig. 1(b). YOLOv9-S was selected as the detection backbone on the basis of its superior accuracy–efficiency trade-off established in a systematic comparison against alternative YOLO variants and scales; the detailed ablation and curves are provided in the Appendix (see Figs. S1–S5 and the accompanying description).

For the image description task, outputs were generated via API using three vision-language models, including Qwen2.5-VL-72B Bai et al. (2025), Mistral-small-3.2-24b Caminha et al. (2025), and Kimi-VL-a3b-thinking Team et al. (2025b). The generated texts were iteratively reviewed and refined by apple pathology experts to ensure domain accuracy. For dialogue generation, diagnostic

JSON outputs from the same models were converted into multi-turn conversations using Qwen3-235b-a22b Yang et al. (2025), followed by expert-guided validation to finalize the dialogues.

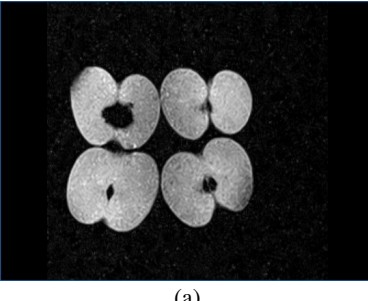 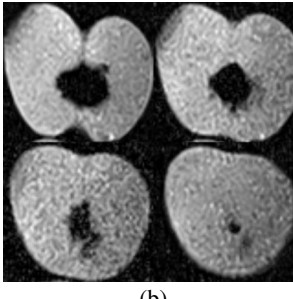

(a)          (b)

Figure 1: LF-MNR image preprocessing workflow: (a) simultaneous scanning of four apples; (b) 2 × 2 merged apple region generated by the AppleNMR V1.0 module.

## 3.2 OVERALL PIPELINE

As shown in Figs. 2, the proposed diagnostic system integrates visual and textual modalities through a modular architecture optimized for both performance and deployment efficiency. The process comprises five main stages:

1. **Image Processing and Visual Feature Extraction.** LF-NMR slice images are cropped using the AppleNMR V1.0 module based on YOLOv9-S and composed into a single composite image per apple. The image $I$ is then passed through a deep convolutional neural network (e.g., ResNet, EfficientNet, MobileNet) to produce a visual representation $\mathbf{v} \in \mathbb{R}^{d_v}$.

2. **Text Tokenization and Encoding.** The expert-generated diagnostic text $T = \{w_1, w_2, \ldots, w_n\}$ is tokenized and embedded using a Transformer-based Vaswani et al. (2017) encoder (e.g., BERT, RoBERTa, ALBERT), yielding a contextualized text feature $\mathbf{t} \in \mathbb{R}^{d_t}$.

3. **Multimodal Fusion Module.** The features $\mathbf{v}$ and $\mathbf{t}$ are combined through one of four fusion mechanisms: Early Fusion with Self-Attention(EF+SA), Gated Fusion, Feature concat and Cross-Attention Chen et al. (2021). The output is a joint representation $\mathbf{h} \in \mathbb{R}^{d_h}$ that captures cross-modal correlations.

4. **Classification Head.** The fused representation $\mathbf{h}$ is fed into a two-layer fully connected network with dropout, producing a predicted probability

$$\hat{y} = \sigma\big(\mathbf{W}_2 \cdot \mathrm{ReLU}(\mathbf{W}_1 \cdot \mathbf{h} + b_1) + b_2\big) \tag{1}$$

where $\sigma(\cdot)$ denotes the sigmoid activation.

5. **Evaluation and Deployment Scoring.** At inference time, the model outputs both the predicted label and a comprehensive evaluation based on three aspects: (i) standard classification metrics such as F1-score, AUC, MCC and Balanced Acc, (ii) model efficiency indicators including parameter count and inference latency, and (iii) a newly proposed metric, TAAPM, which integrates these factors to provide a task-aware and deployment-oriented performance score.

## 3.3 TASK-AWARE APPLE PATHOLOGY METRIC (TAAPM)

To reflect both diagnostic accuracy and real-world deployment value, a task-aware evaluation metric named **TAAPM** is proposed. TAAPM simulates profit in a commercial fruit supply contract, considering classification performance, quality-control constraints, and operational cost. It provides an interpretable scalar score for model selection in practical apple pathology diagnosis.

The TAAPM is computed based on:

- **Recall** $R$ and **Balanced Accuracy** $B$ — used to derive false positive rate $F$;
- **Business requirements:** required delivery quantity $Q$, price per apple $p_{\text{sell}}$, cost per apple $c_{\text{buy}}$ and extra cost $c_{\text{extra}}$, defect tolerance threshold $\alpha$, and background defect rate $r_{\text{bad}}$.

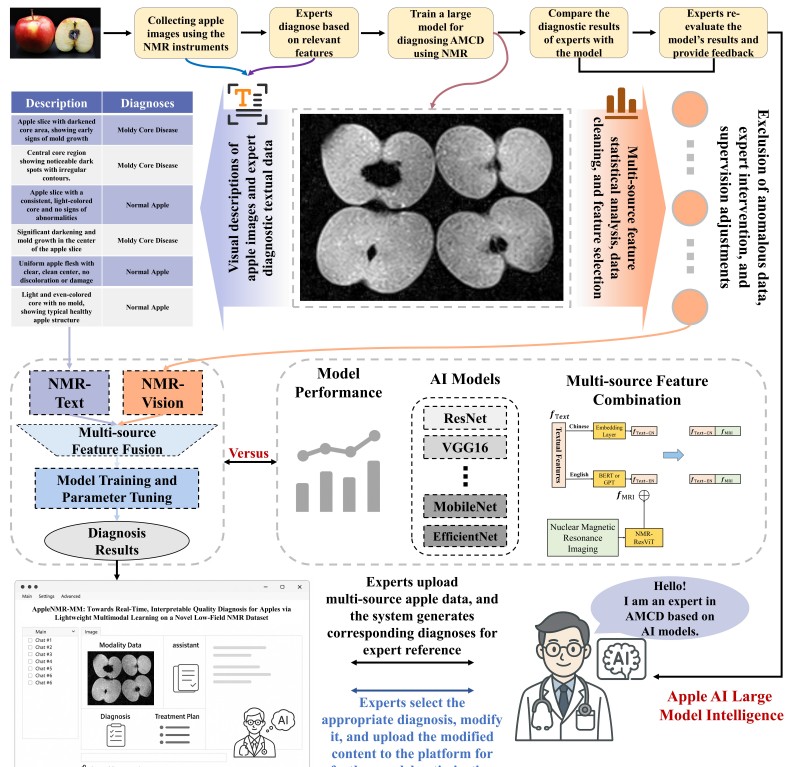

Figure 2: System overview of AppleNMR-MM: Multimodal apple diagnosis pipeline integrating NMR-based imaging and expert textual knowledge. The framework supports expert-in-the-loop diagnosis generation, real-time feedback, and model optimization through multi-source feature fusion and lightweight AI models.

**Quality Gate.**

$$F = \text{clamp}(R + 1 - 2B)$$

$$\text{defect\_prop} = \frac{F\, r_{\text{bad}}}{R\,(1 - r_{\text{bad}}) + F\, r_{\text{bad}}} \tag{2}$$

where $\text{clamp}(\cdot)$ bounds $F$ to $[0, 1]$. The model passes the gate if $\text{defect\_prop} \leq \alpha$.

**Quantity Gate.** Let the expected yield rate

$$k = R \cdot (1 - r_{\text{bad}}) + F \cdot r_{\text{bad}} \tag{3}$$

and required batches

$$x^{\star} = \left\lceil \frac{Q}{k} \right\rceil \tag{4}$$

If quality gate is passed, profit

$$\text{profit} = Q \cdot p_{\text{sell}} - x^{\star} \cdot (c_{\text{buy}} + c_{\text{extra}}) \tag{5}$$

otherwise a penalty of $Q \cdot p_{\text{sell}}$ is applied.

**Final Score.** TAAPM therefore unifies predictive accuracy, quality control, and cost efficiency into a single criterion, enabling selection of models that balance diagnostic power with deployment feasibility. A detailed sensitivity analysis of this metric, examining its behavior under varying defect rates, tolerance thresholds, and order sizes, is provided in the Appendix (see Figs. S6 and the corresponding discussion).

# 4 EXPERIMENT

## 4.1 IMPLEMENTATION DETAILS

All experiments were conducted on a workstation running Windows 11, equipped with a 12th Gen Intel(R) Core(TM) i9-12900H CPU, an NVIDIA RTX 3080 Ti Laptop GPU with 16 GB of dedicated memory, and 64 GB of RAM. Model development and evaluation were implemented using Python 3.10 and the PyTorch deep learning framework, with GPU acceleration enabled via CUDA version 11.8. For all text encoders, input sequences were truncated or padded to a maximum length of 512 tokens. Vision encoders operated on images resized to $224 \times 224$ pixels. Training was performed with a batch size of 32 and a maximum of 50 epochs per run. The Adam optimizer was used with an initial learning rate of 2e-4. To ensure experimental reproducibility, all random number generators were seeded with a fixed value of 42.

## 4.2 MAIN RESULTS

As a preliminary step, conventional machine–learning (ML) classifiers trained on handcrafted descriptors extracted from LF-MNR images were developed and evaluated; their aggregate performance is summarized in Fig. S7. In contrast, deep-learning (DL) models based on CNN backbones (Fig. S8) achieved consistently superior results across repeated runs, with higher ROC/AUC, stronger TAAPM, and reduced variance on both training (out-of-fold) and held-out testing. These findings indicate that fixed, hand-engineered features insufficiently capture the spatial–intensity structure of LF-MNR data, whereas representations learned end-to-end by CNN backbones provide more discriminative and deployment-robust signals. Accordingly, the subsequent subsubsection *Comparison with Text-Only, Vision-Only Models and Different Fusion Methods* focuses on DL backbones and their text/vision fusion variants.

### 4.2.1 COMPARISON WITH TEXT-ONLY, VISION-ONLY MODELS AND DIFFERENT FUSION METHODS

To evaluate the standalone capacity of textual information in apple disease classification, three transformer-based language models—BERT-base-uncased Devlin et al. (2019), RoBERTa-base Liu et al. (2019), DistilBERT-base-uncased Sanh et al. (2019), and ALBERT-base-v2 Lan et al. (2020)—were examined under a text-only setting. Each model was trained using pretrained and scratch initialization and evaluated across different textual corpora, including Qwen2.5-VL-72B, Mistral-small-3.2-24b, and Kimi-VL-a3b-thinking. All configurations were trained for up to 60 epochs and repeated five times to account for performance variance. As shown in Table. 1 and Table. S1, ALBERT-base-v2 (pretrained) consistently achieved the best overall performance on the Kimi-VL-a3b-thinking dataset, with an F1 score of $0.786 \pm 0.015$, AUC of $0.805 \pm 0.019$, MCC of $0.619 \pm 0.015$, and balanced accuracy of $0.828 \pm 0.017$. These results demonstrate the effectiveness of ALBERT's parameter-sharing mechanism in efficiently encoding diagnostic textual cues. In contrast, BERT-base-uncased (pretrained) performed less competitively, particularly in terms of MCC $(0.365 \pm 0.007)$ and balanced accuracy $(0.688 \pm 0.016)$, suggesting that deeper architectures may not necessarily generalize better in low-resource diagnostic settings.

**Table 1.** Text-only results. Top 4 configurations selected from 4 language models $\times$ 3 text sources $\times$ 2 initialization types.

| Src Text | Model | Init | F1 | AUC | MCC | Bal-Acc |
|----------|-------|------|-----|-----|-----|---------|
| kimi | albert-base-v2 | pretrained | **0.786 ± 0.015** | **0.805 ± 0.019** | **0.619 ± 0.015** | **0.828 ± 0.017** |
| kimi | distilbert-base-uncased | pretrained | 0.714 ± 0.014 | 0.770 ± 0.016 | 0.445 ± 0.008 | 0.734 ± 0.013 |
| kimi | bert-base-uncased | pretrained | 0.681 ± 0.015 | 0.766 ± 0.014 | 0.365 ± 0.007 | 0.688 ± 0.016 |
| concat | distilbert-base-uncased | pretrained | 0.706 ± 0.016 | 0.711 ± 0.015 | 0.508 ± 0.013 | 0.766 ± 0.015 |

**Vision-Only Models** To establish the baseline performance of unimodal visual inputs in apple disease classification, a comprehensive evaluation was conducted using eight representative convolutional neural network architectures under a vision-only setting. These included ResNet variants (resnet18, resnet34 and resnet50) He et al. (2016), DenseNet121 Huang et al. (2017), EfficientNet-B0 Tan & Le (2019), shufflenetv2 Ma et al. (2018) and VGG-like backbones Simonyan & Zisserman (2015). Each model was trained under both pretrained and scratch initialization strategies using the same apple image dataset, which was center-cropped and normalized following the standard ImageNet preprocessing pipeline. All models were trained for up to 60 epochs and repeated five times to ensure statistical reliability. As shown in Table. 2 and Table. S2, ResNet18 (pretrained) achieved the highest overall performance across all evaluated models, yielding an F1 score of 0.769±0.013, AUC of 0.798±0.013, MCC of 0.740±0.010, and balanced accuracy of 0.812 ± 0.015. This demonstrates that, despite its relatively shallow depth, ResNet18 is well-suited for extracting discriminative features from high-resolution apple NMR images.

**Table 2.** Vision-only results. Top 5 configurations selected from 8 vision models.

| Model | Init | F1 | AUC | MCC | Balanced Acc |
|---|---|---|---|---|---|
| resnet18 | pretrained | **0.769 ± 0.013** | **0.798 ± 0.013** | **0.740 ± 0.010** | **0.812 ± 0.015** |
| resnet18 | scratch | 0.632 ± 0.014 | 0.786 ± 0.014 | 0.472 ± 0.010 | 0.756 ± 0.011 |
| densenet121 | scratch | 0.571 ± 0.012 | 0.780 ± 0.013 | 0.447 ± 0.008 | 0.702 ± 0.015 |
| resnet50 | pretrained | 0.571 ± 0.008 | 0.762 ± 0.014 | 0.447 ± 0.007 | 0.702 ± 0.015 |
| efficientnet_b0 | scratch | 0.706 ± 0.014 | 0.756 ± 0.015 | 0.587 ± 0.010 | 0.804 ± 0.012 |

**Multimodal Model** To evaluate the effectiveness of multimodal fusion, five representative configurations using EF+SA were compared, as summarized in Table 3 and Table S3. Among them, the best-performing model, ALBERT-base-v2 + EfficientNet-B0, achieved an AUC of 0.863, significantly outperforming the best text-only baseline (AUC: 0.805) and the best vision-only baseline (AUC: 0.798) by +5.8% and +6.5%, respectively. This substantial gain highlights the complementary nature of cross-modal representations. Notably, the RoBERTa-base + ResNet50 fusion achieved a balanced F1 of 0.750, a strong MCC of 0.625, and Balanced Accuracy of 0.812, despite a slightly lower AUC (0.854), suggesting its robustness and generalization under class imbalance. Similarly, DistilBERT-based multimodal variants consistently surpassed their unimodal counterparts, indicating that even lightweight text encoders can benefit from visual alignment.

**Table 3.** Multimodal results using Early Fusion with Self-Attention (EF+SA).

| Src Text | Model | F1 | AUC | MCC | Balanced Acc |
|---|---|---|---|---|---|
| qwen | albert-base-v2+efficientnet_b0 | 0.750 ± 0.010 | **0.863 ± 0.015** | **0.625 ± 0.009** | 0.812 ± 0.011 |
| qwen | distilbert-base-uncased+resnet50 | 0.737 ± 0.014 | 0.859 ± 0.013 | 0.591 ± 0.011 | 0.812 ± 0.015 |
| qwen | roberta-base+resnet34 | 0.718 ± 0.012 | 0.855 ± 0.015 | 0.560 ± 0.010 | 0.797 ± 0.013 |
| kimi | distilbert-base-uncased+resnet34 | 0.667 ± 0.013 | 0.854 ± 0.017 | 0.493 ± 0.007 | 0.750 ± 0.013 |
| kimi | roberta-base+resnet50 | **0.750 ± 0.011** | 0.854 ± 0.015 | **0.625 ± 0.010** | 0.812 ± 0.011 |

**Different Fusion Methods** Having established the effectiveness of multimodal integration, further analysis was conducted to compare different fusion strategies under practical deployment constraints. The TAAPM was adopted as the primary evaluation criterion, as it reflects real-world diagnostic utility by incorporating both predictive performance and model reliability. In addition, the parameter count (params_M) and inference latency (infer_latency_ms) were considered to assess model scalability and efficiency for edge deployment. As summarized in Table 4 and Table S3-Table S6, among all fusion strategies tested, two configurations achieved the highest TAAPM score of 972.84: (1) RoBERTa-base + ResNet18 with Gated Fusion, and (2) ALBERT-base-v2 + MobileNetV3-small with CA fusion. These two models represent contrasting design trade-offs. The Gated Fusion variant exhibited a much larger model size (136.6M parameters) but the lowest latency (2.95 ms), making it suitable for high-throughput server-side inference where compute resources are abundant and response time is critical. In contrast, the CA fusion model offered a lightweight configuration (13.6M parameters) with moderate latency (4.34 ms), enabling efficient deployment in resource-constrained

environments. While other strategies such as EF+SA and Feature Concatenation also delivered competitive TAAPM scores (951.24 and 958.62, respectively), their parameter-efficiency and latency profiles were less favorable under extreme deployment constraints.

**Table 4.** Best configurations per fusion strategy ranked by TAAPM across all 384 configurations.

| Src Text | Model | Params (M) | Inference (ms) | TAAPM | Fusion Type |
|---|---|---|---|---|---|
| qwen | roberta-base+resnet18 | 136.61 | **2.95** | **972.84** | **Gated-Fusion** |
| qwen | albert-base-v2+mobilenet_v3 | **13.61** | 4.34 | **972.84** | **Cross Attention** |
| mistral | albert-base-v2+resnet50 | 35.68 | 4.54 | 958.62 | Feature-concat |
| qwen | albert-base-v2+resnet18 | 25.82 | 4.52 | 951.24 | EF+SA |

## 4.3 ABLATION STUDY AND MODULE CONTRIBUTION ANALYSIS

To systematically quantify the contribution of each modality and assess the robustness of the proposed multimodal fusion architecture, an ablation study was conducted on the two representative model variants: As shown in Table. 5, five configurations were evaluated for each model to measure the impact on classification performance and TAAPM score.

**Table 5.** Ablation Study and Module Contribution Analysis.

| Model Variant | Text | Vision | F1 | AUC | MCC | TAAPM |
|---|---|---|---|---|---|---|
| **NMR-Text-qwen2.5-vl-72b+roberta-base+resnet18-Gate** | | | | | | |
| Full Model | ✓ | ✓ | $0.667 \pm 0.011$ | $0.773 \pm 0.012$ | $0.482 \pm 0.010$ | 972.84 |
| Text Only | ✓ | masked | $0.000 \pm 0.000$ | $0.000 \pm 0.000$ | $0.000 \pm 0.000$ | -3000 |
| Vision Only | masked | ✓ | $0.651 \pm 0.019$ | $0.732 \pm 0.024$ | $0.445 \pm 0.015$ | 826.50 |
| Text + Noise Vision | ✓ | noise | $0.545 \pm 0.018$ | $0.689 \pm 0.024$ | $0.308 \pm 0.009$ | -391.20 |
| Vision + Noise Text | noise | ✓ | $0.667 \pm 0.020$ | $0.771 \pm 0.024$ | $0.472 \pm 0.012$ | 637.68 |
| **NMR-Text-qwen2.5-vl-72b+albert-base-v2+mobilenet_v3_small-CA** | | | | | | |
| Full Model | ✓ | ✓ | $0.667 \pm 0.009$ | $0.785 \pm 0.011$ | $0.482 \pm 0.008$ | 972.84 |
| Text Only | ✓ | masked | $0.500 \pm 0.015$ | $0.460 \pm 0.018$ | $0.000 \pm 0.000$ | -3000 |
| Vision Only | masked | ✓ | $0.698 \pm 0.018$ | $0.811 \pm 0.024$ | $0.535 \pm 0.014$ | 958.62 |
| Text + Noise Vision | ✓ | noise | $0.651 \pm 0.020$ | $0.777 \pm 0.028$ | $0.445 \pm 0.017$ | 826.50 |
| Vision + Noise Text | noise | ✓ | $0.632 \pm 0.010$ | $0.686 \pm 0.008$ | $0.414 \pm 0.004$ | 451.02 |

For the Gated Fusion model, the full multimodal setup yielded a TAAPM of 972.84, with consistent classification metrics (F1: 0.667, AUC: 0.773, MCC: 0.482). Removing the vision modality reduced TAAPM to -3000, indicating that textual information alone was insufficient for meaningful predictions. In contrast, masking the text modality preserved partial functionality (TAAPM: 826.50), suggesting a heavy reliance on visual input in the absence of text. However, introducing visual noise (Text + Noise Vision) caused TAAPM to drop to –391.2, significantly lower than the Vision + Noise Text setting (TAAPM: 637.68), implying that this fusion strategy is more vulnerable to corrupted visual signals.

By comparison, the CA-based model demonstrated greater robustness and balance. The full model also achieved a TAAPM of 972.84, while the vision-only condition (masked text) still retained a high TAAPM of 958.62, with strong AUC (0.811) and MCC (0.535). In contrast, the text-only condition resulted in a TAAPM of –3000, confirming the relatively dominant role of the visual stream in this setup. Importantly, both noisy variants (Text + Noise Vision and Vision + Noise Text) maintained TAAPM scores above 450, indicating robust cross-modal alignment and a degree of error tolerance absent in the gated design.

## 4.4 NMR-APPLE-EXPERT SYSTEM RESULTS

**System overview.** An expert-facing decision pipeline was implemented to operationalize AMCD identification, lot-level quality assessment, and evidence-grounded expert Q&A for LF-MNR apples. The perception front-end uses the AppleNMR V1.0 preprocessor and the best-performing CNN backbones selected to produce calibrated per-apple predictions and uncertainty. On top of

perception, a MACCT with RAG orchestrates three roles—*triage → diagnose → explain*—over a domain corpus comprising standard operating procedures (SOPs), pathology notes, and batch logs.

**Expert Q&A (interactive).** Domain experts can query the system in natural language (e.g., "What are the likely causes of AMCD for Batch 1027?" or "Show all lots where the gate flipped under $\pm 5\%$ cost perturbation"). Queries are compiled into structured filters over predictions and the document store; responses are grounded via RAG with cited passages. When a question requires new evidence, the controller re-enters the diagnose→explain loop to acquire and cite additional support.

To evaluate whether the NMR-Apple-Expert system effectively acquired the capability for apple-specific diagnostic dialogue generation, a QLoRA fine-tuning Dettmers et al. (2023) procedure was applied to the Qwen3-4B model using the proposed vision-language dialogue dataset constructed from expert-validated annotations. Evaluation was conducted on 100 held-out samples, with human experts assessing whether the generated dialogues accurately reflected domain-specific reasoning, interpretability, and diagnostic relevance. Each sample received a binary judgment (*pass/fail*) based on clinical accuracy and logical coherence.

**Human and GPT-Based Evaluation** In addition to expert review, GPT-based scoring was used to quantitatively assess dialogue quality in multiple dimensions, including contextual coherence, factual consistency, domain-specific relevance and language fluency. Each dimension was rated on a 1–5 scale, with a composite score representing overall dialogue quality. The QLoRA-tuned Qwen3-4B model passed 89 out of 100 human-evaluated samples, achieving an expert pass rate of 89%. This strong alignment between model-generated outputs and expert expectations validates the effectiveness of the proposed multimodal dialogue dataset and the fine-tuning strategy.

**Model Comparison** As shown in Table.6, Qwen3-4B consistently outperformed all open-source 4B-scale models in both human evaluation and GPT-based scoring. Specifically, Qwen3-4B surpassed Mistral-4B and Gemma3-4B-IT Team et al. (2025a) by 29.0 and 28.0 percentage points in human pass rate, respectively. When compared to the commercial GPT-4o API, the QLoRA-tuned Qwen3-4B achieved comparable performance, with only a 3-point gap in human evaluation. These results underscore the practical feasibility of lightweight, domain-adapted models for specialized diagnostic dialogue tasks.

**Table 6.** Human and GPT-based evaluation on the Apple Diagnostic Dialogue task. Qwen3-4B ranks highest among open 4B models, within 3.0% Human Pass and 0.43 GPT-Score of GPT-4o. $\Delta$ columns are differences relative to GPT-4o (higher is better).

| Model | Pass Rate(%) | $\Delta$ vs **GPT-4o** | GPT-Score | $\Delta$ vs **GPT-4o** | Type |
|---|---|---|---|---|---|
| **Qwen3-4B** | **89.0** | **–3.0** | **4.07** | **–0.43** | Open |
| Qwen2.5-VL | 75.0 | –17.0 | 3.85 | –0.65 | Open |
| Kimi-VL-a3b | 73.0 | –19.0 | 3.78 | –0.72 | Open |
| Gemma 3-4B | 61.0 | –31.0 | 3.42 | –1.08 | Open |
| Mistral-4B | 60.0 | –32.0 | 3.35 | –1.15 | Open |
| GPT-4o (API) | 92.0 | **0.0** | 4.50 | **0.0** | Proprietary |

## 5 CONCLUSION

This study presents a novel multimodal framework for real-time, interpretable diagnosis of internal apple defects by leveraging LF-NMR imaging and expert textual knowledge. Through the introduction of the AppleNMR-MM dataset and the design of lightweight fusion models, the proposed approach addresses key challenges in agricultural AI, including non-destructive internal quality assessment, cross-modal reasoning, and deployment feasibility under resource constraints. The work contributes to advancing multimodal learning in plant pathology by shifting the focus from surface-level symptoms to latent internal disorders, and by demonstrating that compact, explainable models can be effectively aligned with domain expertise. This direction not only enhances transparency and robustness in apple quality evaluation but also lays the foundation for scalable, real-world applications in smart agriculture and food safety.

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

## STATEMENT ON THE USE OF LARGE LANGUAGE MODELS

The authors used large language models (LLMs)—specifically *GPT-5* and *DeepSeek*—exclusively for (i) English grammar checking and stylistic polishing, and (ii) LaTeX macro refactoring, table/figure caption formatting, and minor code tidying. No sections of scientific content (problem formulation, methods, results, analysis, or conclusions), no figures/tables, and no quantitative claims were directly generated by LLMs. All scientific text, algorithms, experiments, hyperparameter choices, and result interpretations originated from the authors.

**Scope of Assistance.** LLMs were limited to light-touch edits: rephrasing for clarity, fixing typos, harmonizing terminology, and improving LaTeX layout (e.g., environments, caption alignment, column types). Edits were applied only after author drafting and were accepted or rejected by human co-authors.

**Explicit Non-Use.** LLMs were *not* used to (a) design experiments or choose model architectures/hyperparameters; (b) generate, alter, or select results; (c) create or modify figures/tables or quantitative summaries; (d) fabricate, insert, or verify references; or (e) produce domain claims, diagnostic conclusions, or safety-related recommendations.

**Accountability.** All authors independently verified the technical accuracy and integrity of the final manuscript and accept full responsibility for all content, analyses, and conclusions. Any remaining errors are the authors' own.

**Data, Privacy, and IP.** No confidential, proprietary, or personally identifiable data were provided to LLM services. Prompts and model outputs contained only de-identified text and public LaTeX fragments.

**Reproducibility and Audit.** Representative prompts and corresponding LLM outputs related to language polishing and LaTeX formatting have been archived and can be shared for editorial audit upon reasonable request. All code, data, and experimental logs supporting the scientific results are authored by the team and will be released as stated in the Reproducibility Statement.

**Authorship and Acknowledgment.** LLMs did not meet authorship criteria and are not listed as authors. Their limited role (grammar/style and typesetting assistance) is acknowledged here for transparency.

## A APPENDIX

Figure S1 presents the comparative performance of the best-performing models selected from each YOLO version. The curves report the evolution of training loss, validation loss, precision, recall, and mean average precision (mAP) across 100 epochs. While all versions converge to competitive levels of precision and recall, clear differences emerge in convergence speed and stability. In particular, the YOLOv9-S model demonstrates superior overall performance, achieving consistently lower loss values and higher mAP scores compared to its counterparts. These results indicate that YOLOv9-S provides the most effective trade-off between accuracy and robustness, making it the most suitable candidate among the evaluated versions for the present detection task.

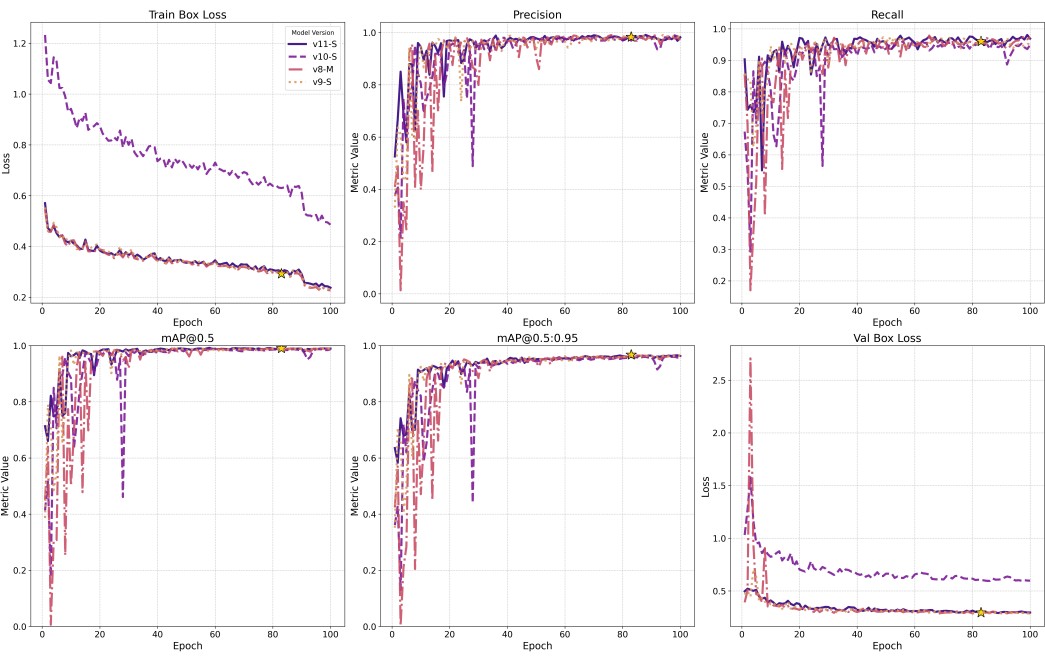

Figure S1: Comparison of the Best-Performing Models Across YOLO Versions

Figure S2 illustrates the performance comparison of YOLOv8 models across different scales (L, M, N, S, X). The training and validation loss curves reveal stable convergence for all model sizes, with smaller variants reaching lower loss values more rapidly. Precision and recall consistently approach high values across epochs, indicating reliable detection capability regardless of scale. The mean average precision (mAP) metrics further confirm strong performance, with only marginal differences between models. Notably, the YOLOv8-M model achieves the most balanced trade-off, delivering competitive accuracy with efficient convergence. These results demonstrate that YOLOv8 provides scalable robustness across different model sizes, enabling flexible adaptation to varying computational and deployment constraints.

Figure S3 depicts the performance comparison of YOLOv9 models across different configurations (C, E, M, S, T). All variants exhibit rapid convergence within the first few epochs, with precision and recall approaching saturation at high values. The mAP@0.5 and mAP@0.5:0.95 curves confirm that each model achieves strong detection performance, with only marginal differences between

**YOLO v8 Model Performance Comparison**

Figure S2: Performance Comparison of YOLOv8 Models with Different Sizes

configurations. Training and validation loss trends indicate stable optimization, although larger models show slightly higher initial variance. Among the evaluated variants, the YOLOv9-S model provides the most favorable balance, achieving high accuracy with reduced computational cost. These findings suggest that YOLOv9 maintains robustness across scales, while the S variant offers an efficient and reliable choice for resource-constrained deployment.

**YOLO v9 Model Performance Comparison**

Figure S3: Performance Comparison of YOLOv9 Models with Different Sizes

Figure S4 shows the performance comparison of YOLOv10 models across different sizes (L, M, N, S, X). All variants achieve rapid convergence, with precision and recall approaching near-perfect values after relatively few epochs. The mAP@0.5 and mAP@0.5:0.95 metrics indicate consistently

high detection performance across scales, confirming the robustness of this version. Training and validation losses gradually decrease, although larger models exhibit greater variance during early optimization. Despite these fluctuations, stability is maintained in later epochs, and performance differences between models remain minor. The YOLOv10-S model emerges as the most effective configuration, sustaining reliable accuracy while offering computational efficiency. These results suggest that YOLOv10 provides strong predictive capability across multiple configurations, with YOLOv10-S being the optimal candidate.

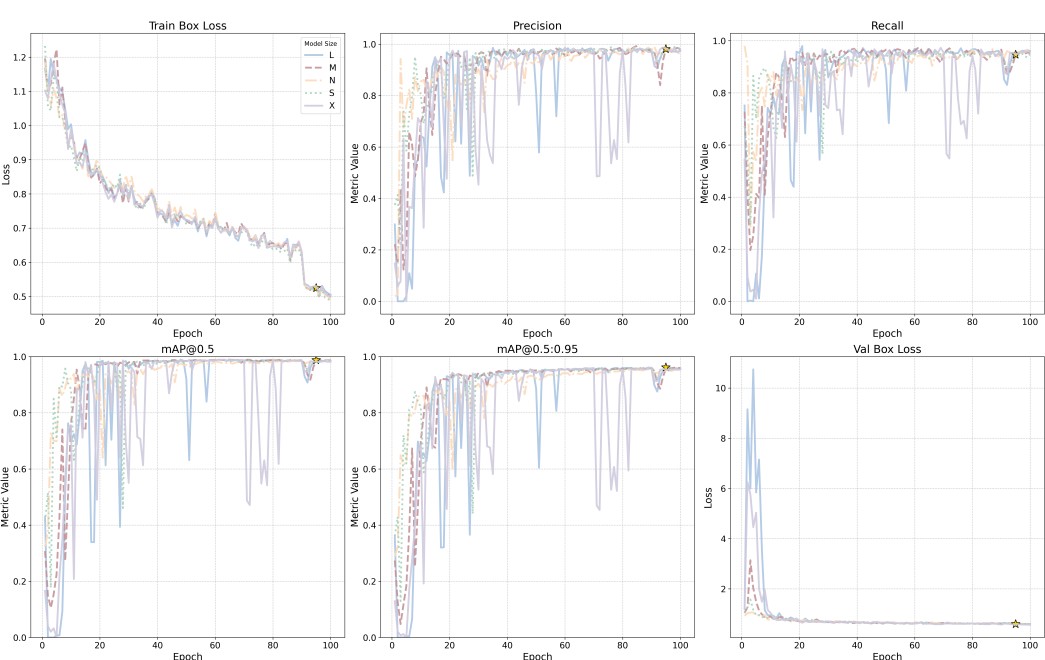

Figure S4: Performance Comparison of YOLOv10 Models with Different Sizes

Figure S5 presents the performance comparison of YOLOv11 models across different sizes (L, M, N, S, X). All configurations demonstrate rapid convergence, with precision and recall consistently exceeding high thresholds after the initial epochs. The mAP@0.5 and mAP@0.5:0.95 results confirm that detection accuracy remains strong across scales, with only subtle variations among model sizes. Training and validation loss curves indicate smooth and stable optimization, and smaller models exhibit particularly efficient convergence with reduced fluctuation. The YOLOv11-S model achieves the best overall results, combining accuracy, stability, and efficiency. These findings highlight the robustness and scalability of YOLOv11, showing that reliable detection performance can be maintained across diverse model sizes, with YOLOv11-S emerging as the most advantageous option for practical deployment.

Figure S6 presents the sensitivity analysis of the newly proposed *TAAPM* metric. The top-left panel shows that as the bad apple rate ($r_{bad}$) increases, profit declines gradually until a critical threshold ($\sim 17\% - 20\%$), where a "profit cliff" emerges due to violation of the batch-level quality gate. The top-right panel indicates that profit exhibits a stepwise transition as defect tolerance $\alpha$ crosses the $1\% - 5\%$ range, shifting between positive and negative returns. The bottom-right panel demonstrates that order size ($Q$) predominantly scales profit or loss linearly, while high defect rates amplify financial risk. The bottom-left heatmap characterizes the feasible region in the (Recall, Balanced Accuracy) plane: insufficient specificity (low BACC) results in negative profit even when recall is high, underscoring the need to prioritize specificity in deployment. Collectively, these findings highlight that *TAAPM* unifies diagnostic performance, quality-gate constraints, and economic outcomes into a single objective function, directly mapping conventional classification metrics to production profit and revealing nonlinear critical phenomena such as profit cliffs. As a novel metric designed in accordance with industrial export regulations and explicitly targeting maximum economic return, *TAAPM* carries significant scientific and practical value, offering quantitative guidance for threshold design, model selection, and production-scale decision-making.

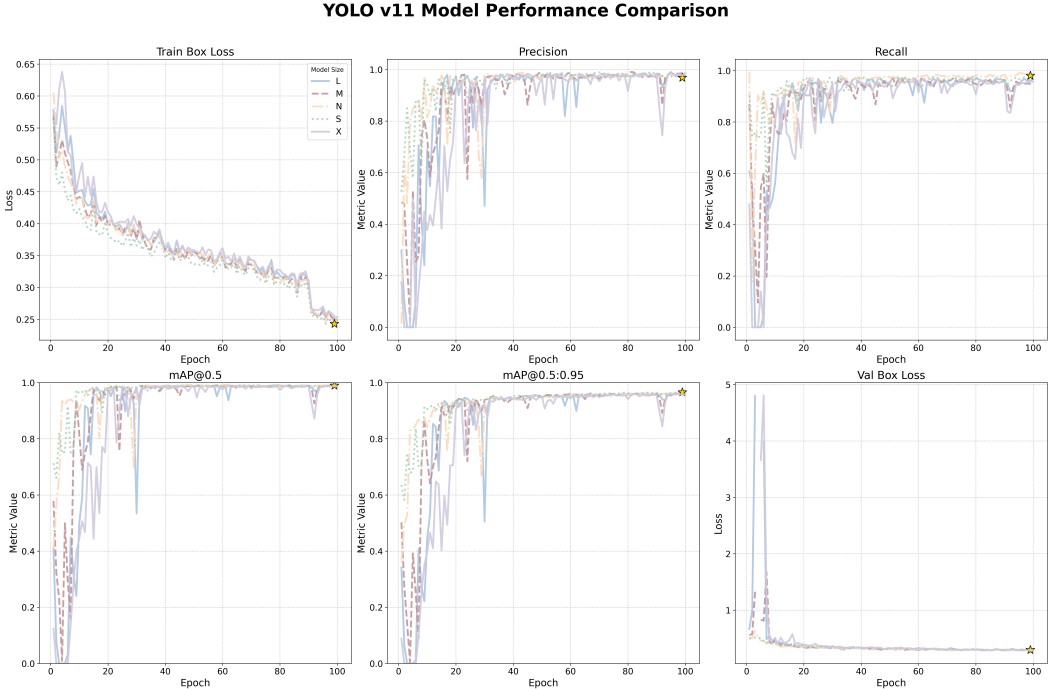

Figure S5: Performance Comparison of YOLOv11 Models with Different Sizes

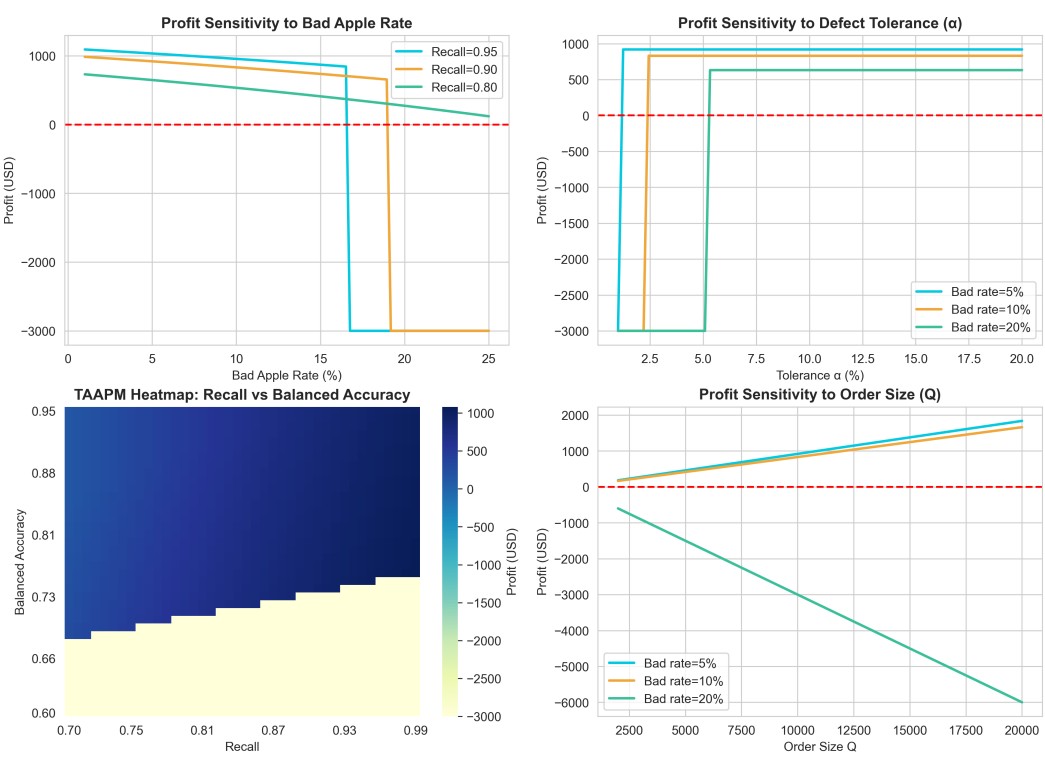

Figure S6: TAAPM Sensitivity analysis

Figure S7 summarizes ROC analyses for traditional machine-learning classifiers trained on hand-crafted features extracted from MNR images; curves show the mean over five repeats and shaded regions denote mean±std. Panel (a) reports out-of-fold training ROC without resampling and yields similar AUCs across models (Random Forest $0.661 \pm 0.012$, SVM $0.655 \pm 0.012$, Logistic $0.669 \pm 0.007$, Gradient Boosting $0.684 \pm 0.039$, KNN $0.668 \pm 0.009$). Panel (b) applies SMOTE and alters the training ranking: Gradient Boosting improves and becomes the strongest ($0.702 \pm 0.027$), whereas KNN degrades markedly ($0.527 \pm 0.013$), consistent with neighborhood distortion under synthetic sampling. Panel (c) shows held-out test ROC without SMOTE, where Gradient Boosting again leads ($0.696 \pm 0.099$), followed by Logistic ($0.688 \pm 0.108$) and SVM ($0.670 \pm 0.101$), with broader uncertainty reflecting the small-sample regime. Panel (d) demonstrates that SMOTE further benefits ensemble and margin-based models on test data (Gradient Boosting $0.772 \pm 0.074$, SVM $0.693 \pm 0.088$, Random Forest $0.680 \pm 0.074$), while KNN deteriorates ($0.520 \pm 0.053$). Collectively, the results indicate that resampling is advantageous for tree-based and linear-margin classifiers but harmful for nearest-neighbor methods, establishing Gradient Boosting as the most reliable baseline for the handcrafted-feature setting.

Panels (e)–(h) report *TAAPM*—the profit-oriented deployment metric, with break-even at 0—computed for the same classifiers; bars show the mean over five repeats and error bars indicate standard deviation. Panel (e) (training, no resampling) shows all models with negative TAAPM; KNN and Logistic Regression incur the least severe losses ($-2515 \pm 418$ and $-2772 \pm 322$), whereas Gradient Boosting underperforms ($-5917 \pm 2549$). Panel (f) (training, SMOTE) reveals marked improvement for ensembles: Gradient Boosting becomes the best on training ($-1822 \pm 1450$), while Random Forest, SVM, and KNN remain near the quality-gate floor (approximately $-3000$). Panel (g) (test, no resampling) indicates that margin/linear models perform best, with SVM and Logistic Regression achieving the highest TAAPM ($-2423 \pm 1155$ and $-2498 \pm 1004$), outperforming tree ensembles. Panel (h) (test, SMOTE) shows a consistent gain for ensembles; Gradient Boosting attains the highest TAAPM and the smallest expected loss ($-1014 \pm 1566$), Random Forest improves to $-3430 \pm 737$, and KNN remains unchanged at the floor. Overall, SMOTE improves profit under TAAPM for tree-based and margin-based classifiers—most notably elevating Gradient Boosting from the worst to the best performer—whereas nearest-neighbor methods are adversely affected. Despite these gains, all models remain below the break-even point, indicating that additional specificity-oriented tuning or feature refinement is required for profitable deployment.

Figure S8 compares ROC curves (mean±std across repeats) for eleven CNN backbones trained on the MNR dataset either from scratch or with pretrained initialization. Panels (a)–(b) summarize training performance: pretraining systematically shifts the curves upward and reduces dispersion, yielding higher AUCs for most architectures (e.g., DenseNet169 rises from $0.604 \pm 0.057$ to $0.718 \pm 0.017$, InceptionV3 from $0.641 \pm 0.029$ to $0.720 \pm 0.013$, MobileNetV3-Large to $0.710 \pm 0.014$). Panels (c)–(d) report held-out test performance: pretraining again improves generalization, with the best AUCs obtained by VGG16 and DenseNet169 ($0.794 \pm 0.041$ and $0.791 \pm 0.057$, respectively), followed by InceptionResNetV2 and MobileNetV3-Large (both $\approx 0.760$). Models that struggle when trained from scratch (e.g., EfficientNetB4/B5: $0.526 \pm 0.051$ and $0.593 \pm 0.134$) become competitive after pretraining ($0.732 \pm 0.032$ and $0.727 \pm 0.058$). Overall, pretrained initialization confers consistent gains and lower variance across backbones, positioning VGG16 and DenseNet169 as the most reliable choices for this dataset under the current protocol.

Panels (e)–(h) report *TAAPM* (profit; break-even at 0) for the same eleven backbones, again summarized as mean±std across repeats. In training from scratch (panel e), most architectures yield negative TAAPM, with several close to break-even—DenseNet169, InceptionV3, and InceptionResNetV2 (all $\approx -100$ to $-70$)—while VGG19 and EfficientNetB4/B5 incur large losses with high variance, indicating instability under small-sample training. With pretrained initialization (panel f), TAAPM shifts upward and dispersion decreases; multiple models approach the profit threshold (InceptionResNetV2 $\approx -12$, InceptionV3 $\approx -49$, EfficientNetB4/B5 within a few hundred, DenseNet169 $\approx -96$), whereas ResNet101 remains markedly below break-even. On held-out tests without pretraining (panel g), two backbones cross or touch break-even (VGG19 $\approx +18$, ResNet50 $\approx +43$) but with substantial uncertainty; EfficientNetB5 shows the lowest TAAPM. With pretraining (panel h), out-of-sample TAAPM further improves and concentrates around zero (approximately $-700$ to $+50$); InceptionResNetV2 attains a slightly positive mean TAAPM, whereas ResNet101 remains strongly negative. Taken together, the TAAPM results corroborate the ROC findings: pretraining consistently enhances profit-oriented behavior and stability. InceptionResNetV2

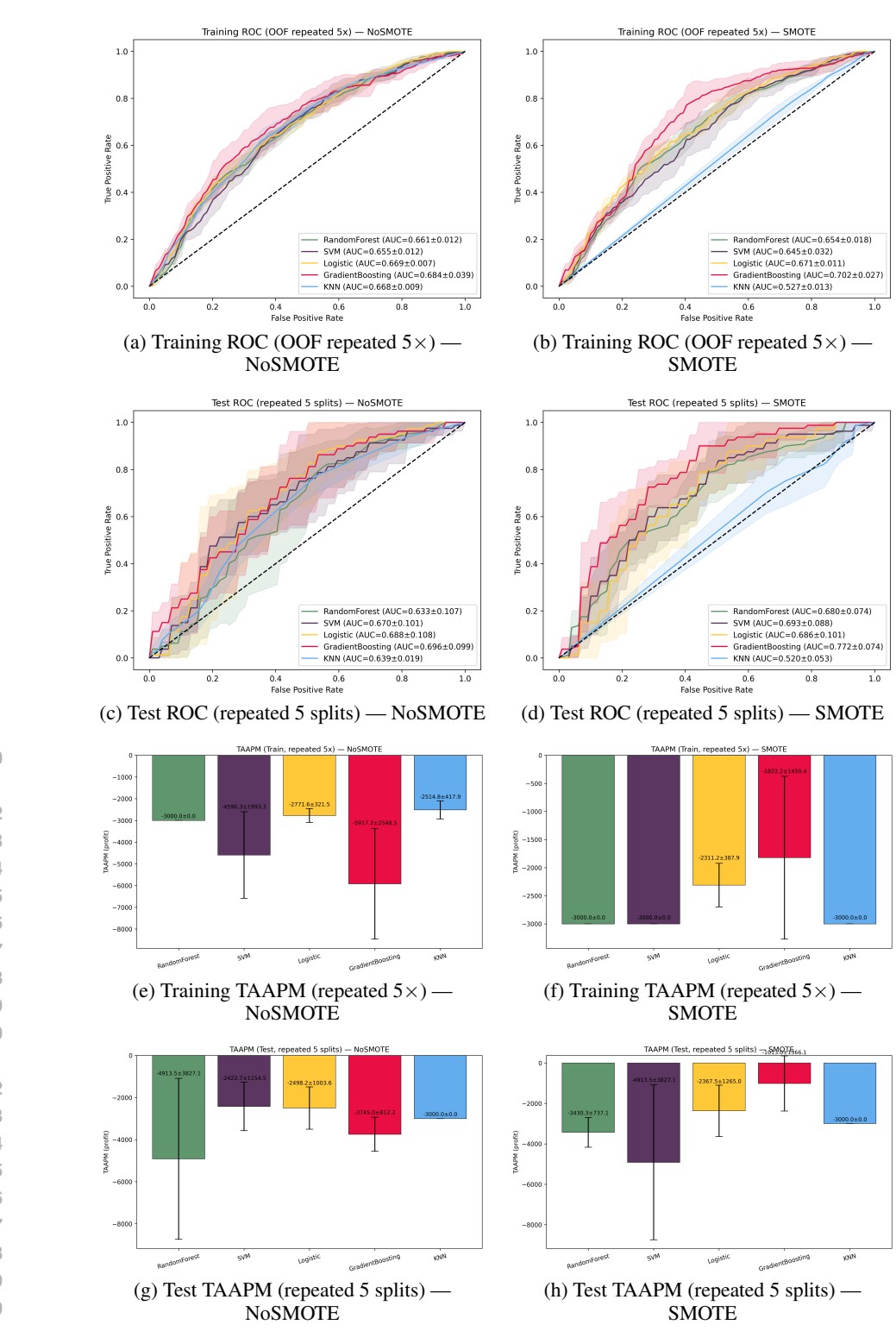

Figure S7: Classification performance of traditional machine-learning models trained on handcrafted features extracted from MNR images, evaluated using ROC analysis and the profit-oriented TAAPM metric under repeated training and testing with and without SMOTE resampling.

emerges as the most promising candidate for profit-aware deployment, whereas large models trained from scratch (e.g., VGG19, EfficientNetB5) are prone to sizeable losses without pretraining.

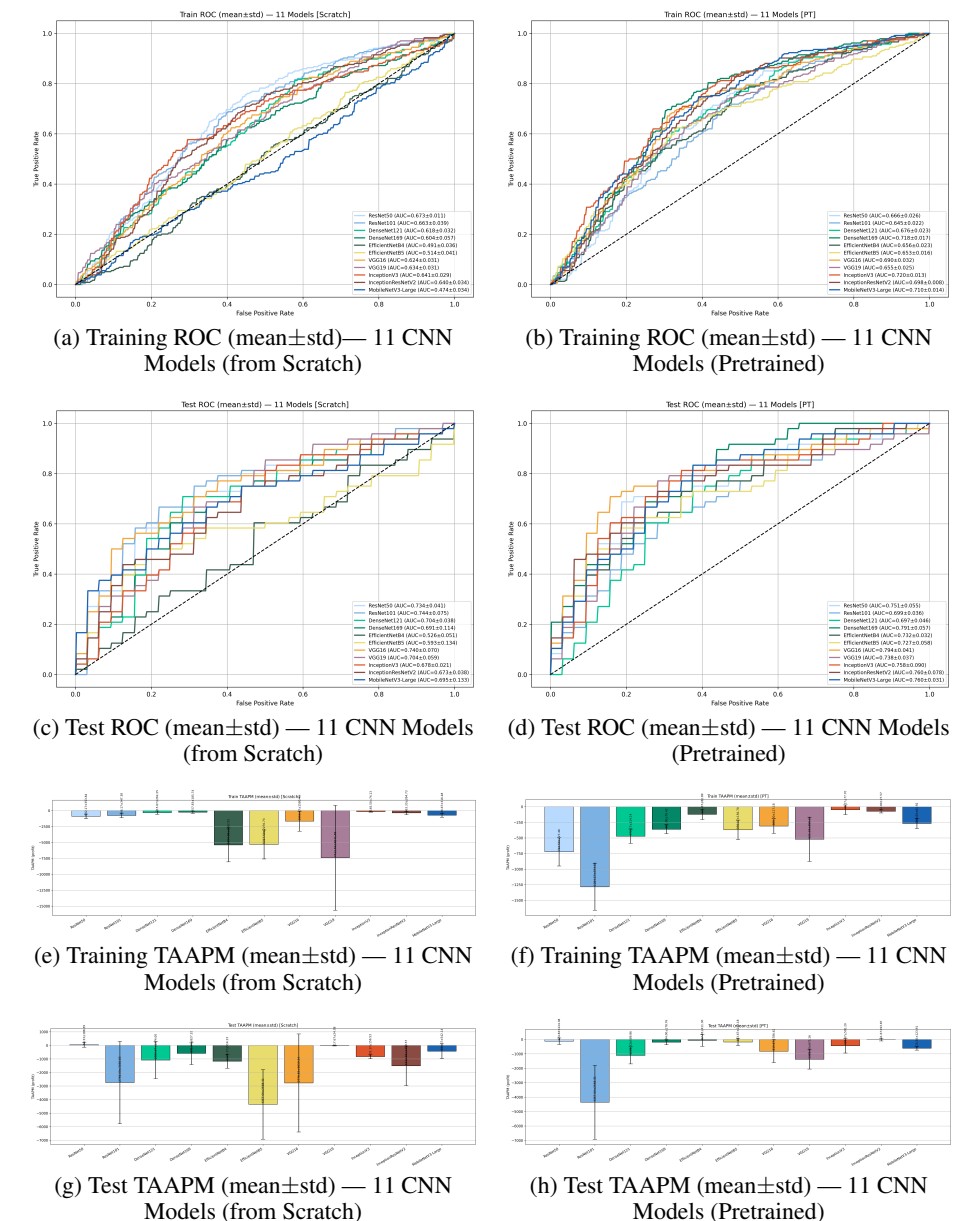

(a) Training ROC (mean±std)— 11 CNN Models (from Scratch)

(b) Training ROC (mean±std) — 11 CNN Models (Pretrained)

(c) Test ROC (mean±std) — 11 CNN Models (from Scratch)

(d) Test ROC (mean±std) — 11 CNN Models (Pretrained)

(e) Training TAAPM (mean±std) — 11 CNN Models (from Scratch)

(f) Training TAAPM (mean±std) — 11 CNN Models (Pretrained)

(g) Test TAAPM (mean±std) — 11 CNN Models (from Scratch)

(h) Test TAAPM (mean±std) — 11 CNN Models (Pretrained)

Figure S8: Classification Performance of CNN Backbones on MNR Images: ROC and TAAPM Analyses (Scratch vs. Pretrained; NoSMOTE vs. SMOTE)

**Table S1.** Text-only classification results for all 31 configurations across three input sources (`kimi`, `qwen`, `mistral`), with various language models and initialization strategies.

| Src Text | Model | Init | F1 | AUC | MCC | Balanced Acc |
|---|---|---|---|---|---|---|
| kimi | albert-base-v2 | pretrained | $0.786 \pm 0.015$ | $0.805 \pm 0.019$ | $0.619 \pm 0.015$ | $0.828 \pm 0.017$ |
| kimi | distilbert-base-uncased | pretrained | $0.714 \pm 0.014$ | $0.770 \pm 0.016$ | $0.445 \pm 0.008$ | $0.734 \pm 0.013$ |
| kimi | bert-base-uncased | pretrained | $0.681 \pm 0.015$ | $0.766 \pm 0.014$ | $0.365 \pm 0.007$ | $0.688 \pm 0.016$ |
| concat | distilbert-base-uncased | pretrained | $0.706 \pm 0.016$ | $0.711 \pm 0.015$ | $0.508 \pm 0.013$ | $0.766 \pm 0.015$ |
| qwen | albert-base-v2 | pretrained | $0.633 \pm 0.013$ | $0.629 \pm 0.015$ | $0.295 \pm 0.007$ | $0.656 \pm 0.017$ |
| mistral | albert-base-v2 | pretrained | $0.400 \pm 0.008$ | $0.592 \pm 0.012$ | $0.000 \pm 0.000$ | $0.500 \pm 0.010$ |

*(end of table)*

*(continued)*

| Src Text | Model | Init | F1 | AUC | MCC | Balanced Acc |
|---|---|---|---|---|---|---|
| kimi | albert-base-v2 | scratch | $0.400 \pm 0.010$ | $0.562 \pm 0.012$ | $0.000 \pm 0.000$ | $0.500 \pm 0.011$ |
| concat | albert-base-v2 | scratch | $0.437 \pm 0.009$ | $0.559 \pm 0.012$ | $0.000 \pm 0.000$ | $0.500 \pm 0.013$ |
| kimi | roberta-base | scratch | $0.400 \pm 0.009$ | $0.547 \pm 0.014$ | $0.000 \pm 0.000$ | $0.500 \pm 0.009$ |
| qwen | bert-base-uncased | pretrained | $0.541 \pm 0.011$ | $0.547 \pm 0.013$ | $0.259 \pm 0.007$ | $0.625 \pm 0.016$ |
| mistral | bert-base-uncased | scratch | $0.400 \pm 0.010$ | $0.539 \pm 0.012$ | $0.000 \pm 0.000$ | $0.500 \pm 0.010$ |
| kimi | distilbert-base-uncased | scratch | $0.400 \pm 0.008$ | $0.525 \pm 0.010$ | $0.000 \pm 0.000$ | $0.500 \pm 0.012$ |
| concat | distilbert-base-uncased | scratch | $0.400 \pm 0.007$ | $0.512 \pm 0.010$ | $0.000 \pm 0.000$ | $0.500 \pm 0.011$ |
| concat | roberta-base | pretrained | $0.612 \pm 0.012$ | $0.512 \pm 0.009$ | $0.227 \pm 0.004$ | $0.609 \pm 0.013$ |
| qwen | albert-base-v2 | scratch | $0.400 \pm 0.009$ | $0.502 \pm 0.013$ | $0.000 \pm 0.000$ | $0.500 \pm 0.011$ |
| mistral | roberta-base | pretrained | $0.250 \pm 0.005$ | $0.479 \pm 0.009$ | $0.000 \pm 0.000$ | $0.500 \pm 0.011$ |
| concat | bert-base-uncased | scratch | $0.400 \pm 0.010$ | $0.479 \pm 0.009$ | $0.000 \pm 0.000$ | $0.500 \pm 0.013$ |
| qwen | distilbert-base-uncased | pretrained | $0.531 \pm 0.012$ | $0.473 \pm 0.013$ | $0.066 \pm 0.001$ | $0.531 \pm 0.013$ |
| concat | albert-base-v2 | pretrained | $0.455 \pm 0.011$ | $0.471 \pm 0.012$ | $-0.030 \pm 0.001$ | $0.484 \pm 0.012$ |
| mistral | roberta-base | scratch | $0.400 \pm 0.011$ | $0.459 \pm 0.012$ | $0.000 \pm 0.000$ | $0.500 \pm 0.012$ |
| concat | roberta-base | scratch | $0.400 \pm 0.009$ | $0.459 \pm 0.010$ | $0.000 \pm 0.000$ | $0.500 \pm 0.009$ |
| mistral | distilbert-base-uncased | scratch | $0.400 \pm 0.008$ | $0.457 \pm 0.011$ | $0.000 \pm 0.000$ | $0.500 \pm 0.009$ |
| qwen | roberta-base | scratch | $0.400 \pm 0.008$ | $0.451 \pm 0.011$ | $0.000 \pm 0.000$ | $0.500 \pm 0.013$ |
| mistral | albert-base-v2 | scratch | $0.435 \pm 0.009$ | $0.447 \pm 0.010$ | $-0.060 \pm 0.001$ | $0.469 \pm 0.009$ |
| qwen | roberta-base | pretrained | $0.400 \pm 0.009$ | $0.446 \pm 0.012$ | $0.000 \pm 0.000$ | $0.500 \pm 0.012$ |
| mistral | distilbert-base-uncased | pretrained | $0.543 \pm 0.014$ | $0.445 \pm 0.009$ | $0.193 \pm 0.005$ | $0.562 \pm 0.011$ |
| qwen | distilbert-base-uncased | scratch | $0.438 \pm 0.008$ | $0.434 \pm 0.011$ | $-0.125 \pm 0.003$ | $0.438 \pm 0.009$ |
| mistral | bert-base-uncased | pretrained | $0.469 \pm 0.011$ | $0.428 \pm 0.008$ | $-0.062 \pm 0.001$ | $0.469 \pm 0.012$ |
| qwen | bert-base-uncased | scratch | $0.400 \pm 0.009$ | $0.410 \pm 0.008$ | $0.000 \pm 0.000$ | $0.500 \pm 0.013$ |
| kimi | bert-base-uncased | scratch | $0.400 \pm 0.011$ | $0.393 \pm 0.010$ | $0.000 \pm 0.000$ | $0.500 \pm 0.011$ |
| kimi | roberta-base | pretrained | $0.400 \pm 0.010$ | $0.277 \pm 0.007$ | $0.000 \pm 0.000$ | $0.500 \pm 0.009$ |

*(end of table)*

**Table S2.** Vision-only results. Top 15 configurations selected from 8 vision backbones $\times$ 2 initialization types.

| Model | Init | F1 | AUC | MCC | Balanced Acc |
|---|---|---|---|---|---|
| resnet18 | pretrained | $0.769 \pm 0.013$ | $0.798 \pm 0.013$ | $0.740 \pm 0.010$ | $0.812 \pm 0.015$ |
| resnet18 | scratch | $0.632 \pm 0.014$ | $0.786 \pm 0.014$ | $0.472 \pm 0.010$ | $0.756 \pm 0.011$ |
| densenet121 | scratch | $0.571 \pm 0.012$ | $0.780 \pm 0.013$ | $0.447 \pm 0.008$ | $0.702 \pm 0.015$ |
| resnet50 | pretrained | $0.571 \pm 0.008$ | $0.762 \pm 0.014$ | $0.447 \pm 0.007$ | $0.702 \pm 0.015$ |
| efficientnet_b0 | scratch | $0.706 \pm 0.014$ | $0.756 \pm 0.015$ | $0.587 \pm 0.010$ | $0.804 \pm 0.012$ |
| efficientnet_b0 | pretrained | $0.632 \pm 0.012$ | $0.738 \pm 0.013$ | $0.472 \pm 0.007$ | $0.756 \pm 0.010$ |
| resnet34 | scratch | $0.632 \pm 0.012$ | $0.708 \pm 0.011$ | $0.472 \pm 0.010$ | $0.756 \pm 0.015$ |
| densenet121 | pretrained | $0.667 \pm 0.009$ | $0.702 \pm 0.011$ | $0.526 \pm 0.010$ | $0.780 \pm 0.014$ |
| resnet50 | scratch | $0.615 \pm 0.009$ | $0.667 \pm 0.010$ | $0.535 \pm 0.007$ | $0.726 \pm 0.015$ |
| shufflenet_v2_x1_0 | pretrained | $0.333 \pm 0.007$ | $0.661 \pm 0.012$ | $0.201 \pm 0.003$ | $0.577 \pm 0.010$ |
| resnet34 | pretrained | $0.667 \pm 0.013$ | $0.661 \pm 0.011$ | $0.553 \pm 0.012$ | $0.765 \pm 0.017$ |
| mobilenet_v3_small | pretrained | $0.526 \pm 0.007$ | $0.577 \pm 0.009$ | $0.313 \pm 0.004$ | $0.670 \pm 0.009$ |
| vgg16 | scratch | $0.308 \pm 0.005$ | $0.506 \pm 0.007$ | $0.127 \pm 0.002$ | $0.554 \pm 0.009$ |
| mobilenet_v3_small | scratch | $0.432 \pm 0.009$ | $0.500 \pm 0.009$ | $0.000 \pm 0.000$ | $0.500 \pm 0.008$ |
| shufflenet_v2_x1_0 | scratch | $0.483 \pm 0.007$ | $0.494 \pm 0.007$ | $0.208 \pm 0.004$ | $0.604 \pm 0.009$ |

*(end of table)*

**Table S3.** Multimodal results using Early Fusion with Self-Attention (EF+SA)

| Src Text | Model | F1 | AUC | MCC | Balanced Acc |
|---|---|---|---|---|---|
| qwen | albert-base-v2+resnet18 | 0.714±0.012 | 0.805±0.016 | 0.562±0.008 | 0.797±0.016 |
| qwen | distilbert-base-uncased+mobilenet_v3_small | 0.732±0.014 | 0.854±0.012 | 0.590±0.008 | 0.812±0.013 |
| kimi | albert-base-v2+resnet34 | 0.667±0.012 | 0.834±0.016 | 0.473±0.007 | 0.750±0.012 |
| mistral | bert-base-uncased+densenet121 | 0.667±0.012 | 0.738±0.010 | 0.473±0.009 | 0.750±0.013 |
| qwen | bert-base-uncased+densenet121 | 0.667±0.011 | 0.805±0.012 | 0.473±0.008 | 0.750±0.014 |
| qwen | distilbert-base-uncased+efficientnet_b0 | 0.667±0.013 | 0.752±0.012 | 0.473±0.009 | 0.750±0.014 |
| mistral | distilbert-base-uncased+mobilenet_v3_small | 0.683±0.010 | 0.795±0.016 | 0.501±0.010 | 0.766±0.014 |
| qwen | distilbert-base-uncased+efficientnet_b0 | 0.683±0.013 | 0.730±0.012 | 0.501±0.008 | 0.766±0.010 |
| qwen | distilbert-base-uncased+resnet18 | 0.700±0.014 | 0.801±0.016 | 0.530±0.008 | 0.781±0.015 |
| qwen | roberta-base+resnet34 | 0.718±0.012 | 0.855±0.015 | 0.560±0.010 | 0.797±0.013 |
| kimi | bert-base-uncased+resnet18 | 0.718±0.011 | 0.850±0.015 | 0.560±0.009 | 0.797±0.016 |
| mistral | roberta-base+resnet34 | 0.718±0.013 | 0.836±0.015 | 0.560±0.010 | 0.797±0.014 |
| qwen | distilbert-base-uncased+resnet50 | 0.737±0.014 | 0.859±0.013 | 0.591±0.011 | 0.812±0.015 |
| qwen | albert-base-v2+resnet50 | 0.757±0.015 | 0.861±0.013 | 0.624±0.011 | 0.828±0.011 |
| qwen | albert-base-v2+resnet18 | 0.757±0.011 | 0.807±0.014 | 0.624±0.012 | 0.828±0.014 |
| kimi | distilbert-base-uncased+densenet121 | 0.634±0.010 | 0.713±0.009 | 0.413±0.007 | 0.719±0.011 |
| kimi | roberta-base+resnet18 | 0.634±0.009 | 0.760±0.010 | 0.413±0.006 | 0.719±0.013 |
| kimi | bert-base-uncased+resnet18 | 0.650±0.010 | 0.717±0.010 | 0.442±0.008 | 0.734±0.014 |
| kimi | distilbert-base-uncased+densenet121 | 0.667±0.009 | 0.729±0.013 | 0.472±0.007 | 0.750±0.013 |
| qwen | bert-base-uncased+resnet18 | 0.684±0.010 | 0.803±0.013 | 0.503±0.007 | 0.766±0.011 |
| qwen | bert-base-uncased+resnet34 | 0.684±0.012 | 0.768±0.014 | 0.503±0.009 | 0.766±0.012 |
| kimi | roberta-base+efficientnet_b0 | 0.684±0.012 | 0.809±0.014 | 0.503±0.009 | 0.766±0.012 |
| mistral | distilbert-base-uncased+resnet34 | 0.703±0.013 | 0.805±0.016 | 0.535±0.010 | 0.781±0.012 |
| kimi | roberta-base+resnet34 | 0.703±0.013 | 0.805±0.011 | 0.535±0.007 | 0.781±0.016 |
| mistral | roberta-base+efficientnet_b0 | 0.703±0.011 | 0.756±0.014 | 0.535±0.009 | 0.781±0.011 |
| kimi | roberta-base+resnet18 | 0.703±0.011 | 0.812±0.014 | 0.535±0.009 | 0.781±0.015 |
| qwen | bert-base-uncased+mobilenet_v3_small | 0.722±0.010 | 0.820±0.016 | 0.568±0.008 | 0.797±0.014 |
| kimi | distilbert-base-uncased+resnet50 | 0.722±0.011 | 0.732±0.013 | 0.568±0.010 | 0.797±0.015 |
| mistral | distilbert-base-uncased+resnet18 | 0.722±0.010 | 0.844±0.017 | 0.568±0.011 | 0.797±0.014 |
| mistral | roberta-base+resnet50 | 0.722±0.011 | 0.816±0.011 | 0.568±0.008 | 0.797±0.015 |
| mistral | bert-base-uncased+efficientnet_b0 | 0.722±0.012 | 0.822±0.011 | 0.568±0.010 | 0.797±0.014 |
| kimi | distilbert-base-uncased+resnet18 | 0.743±0.010 | 0.840±0.013 | 0.602±0.009 | 0.812±0.013 |
| kimi | roberta-base+mobilenet_v3_small | 0.743±0.012 | 0.799±0.014 | 0.602±0.011 | 0.812±0.015 |
| kimi | distilbert-base-uncased+efficientnet_b0 | 0.765±0.015 | 0.789±0.011 | 0.639±0.010 | 0.828±0.012 |
| mistral | albert-base-v2+resnet50 | 0.765±0.015 | 0.824±0.013 | 0.639±0.010 | 0.828±0.013 |
| kimi | bert-base-uncased+resnet34 | 0.788±0.011 | 0.820±0.013 | 0.678±0.011 | 0.844±0.015 |
| mistral | bert-base-uncased+resnet50 | 0.615±0.008 | 0.721±0.014 | 0.383±0.007 | 0.703±0.011 |
| qwen | distilbert-base-uncased+resnet34 | 0.615±0.010 | 0.658±0.011 | 0.383±0.006 | 0.703±0.010 |
| mistral | roberta-base+densenet121 | 0.632±0.011 | 0.646±0.009 | 0.414±0.006 | 0.719±0.011 |
| qwen | bert-base-uncased+densenet121 | 0.632±0.011 | 0.811±0.015 | 0.414±0.006 | 0.719±0.010 |
| kimi | distilbert-base-uncased+resnet50 | 0.632±0.010 | 0.717±0.011 | 0.414±0.006 | 0.719±0.012 |
| qwen | albert-base-v2+efficientnet_b0 | 0.649±0.009 | 0.811±0.012 | 0.445±0.008 | 0.734±0.011 |
| kimi | bert-base-uncased+mobilenet_v3_small | 0.649±0.011 | 0.773±0.013 | 0.445±0.006 | 0.734±0.013 |
| qwen | distilbert-base-uncased+resnet18 | 0.649±0.010 | 0.777±0.013 | 0.445±0.009 | 0.734±0.011 |
| mistral | distilbert-base-uncased+efficientnet_b0 | 0.649±0.013 | 0.721±0.011 | 0.445±0.007 | 0.734±0.014 |
| qwen | distilbert-base-uncased+resnet34 | 0.667±0.011 | 0.766±0.015 | 0.478±0.007 | 0.750±0.012 |
| mistral | albert-base-v2+resnet34 | 0.686±0.011 | 0.807±0.012 | 0.512±0.007 | 0.766±0.014 |
| mistral | roberta-base+resnet34 | 0.686±0.010 | 0.789±0.011 | 0.512±0.007 | 0.766±0.011 |
| mistral | albert-base-v2+resnet18 | 0.686±0.010 | 0.781±0.014 | 0.512±0.008 | 0.766±0.010 |
| kimi | roberta-base+efficientnet_b0 | 0.686±0.013 | 0.770±0.014 | 0.512±0.008 | 0.766±0.013 |
| mistral | bert-base-uncased+resnet18 | 0.706±0.012 | 0.836±0.013 | 0.548±0.008 | 0.781±0.015 |

*(end of table)*

*(continued)*

| Src Text | Model | F1 | AUC | MCC | Balanced Acc |
|---|---|---|---|---|---|
| qwen | albert-base-v2+mobilenet_v3_small | 0.706±0.014 | 0.848±0.014 | 0.548±0.011 | 0.781±0.013 |
| qwen | distilbert-base-uncased+densenet121 | 0.706±0.011 | 0.809±0.014 | 0.548±0.011 | 0.781±0.013 |
| kimi | roberta-base+resnet34 | 0.706±0.013 | 0.820±0.015 | 0.548±0.010 | 0.781±0.011 |
| mistral | bert-base-uncased+densenet121 | 0.706±0.011 | 0.781±0.011 | 0.548±0.010 | 0.781±0.014 |
| kimi | bert-base-uncased+resnet50 | 0.706±0.013 | 0.791±0.012 | 0.548±0.010 | 0.781±0.012 |
| qwen | distilbert-base-uncased+resnet18 | 0.706±0.013 | 0.842±0.016 | 0.548±0.009 | 0.781±0.013 |
| mistral | albert-base-v2+efficientnet_b0 | 0.727±0.013 | 0.773±0.014 | 0.585±0.011 | 0.797±0.014 |
| mistral | albert-base-v2+densenet121 | 0.727±0.010 | 0.807±0.011 | 0.585±0.011 | 0.797±0.013 |
| kimi | albert-base-v2+vgg16 | 0.727±0.014 | 0.785±0.013 | 0.585±0.008 | 0.797±0.013 |
| qwen | bert-base-uncased+resnet18 | 0.727±0.013 | 0.842±0.014 | 0.585±0.008 | 0.797±0.013 |
| qwen | distilbert-base-uncased+resnet34 | 0.727±0.014 | 0.828±0.012 | 0.585±0.009 | 0.797±0.012 |
| qwen | albert-base-v2+efficientnet_b0 | 0.750±0.010 | 0.863±0.015 | 0.625±0.009 | 0.812±0.011 |
| kimi | roberta-base+resnet50 | 0.750±0.011 | 0.854±0.015 | 0.625±0.010 | 0.812±0.011 |
| kimi | distilbert-base-uncased+resnet34 | 0.750±0.010 | 0.848±0.017 | 0.625±0.008 | 0.812±0.013 |
| mistral | roberta-base+efficientnet_b0 | 0.774±0.012 | 0.803±0.012 | 0.667±0.013 | 0.828±0.015 |
| mistral | roberta-base+densenet121 | 0.595±0.009 | 0.680±0.009 | 0.356±0.006 | 0.688±0.012 |
| kimi | distilbert-base-uncased+resnet18 | 0.595±0.011 | 0.758±0.012 | 0.356±0.006 | 0.688±0.010 |
| kimi | albert-base-v2+densenet121 | 0.611±0.012 | 0.695±0.011 | 0.388±0.007 | 0.703±0.010 |
| kimi | roberta-base+densenet121 | 0.611±0.009 | 0.664±0.013 | 0.388±0.006 | 0.703±0.014 |
| qwen | roberta-base+resnet50 | 0.611±0.012 | 0.752±0.010 | 0.388±0.007 | 0.703±0.011 |
| mistral | distilbert-base-uncased+resnet18 | 0.611±0.009 | 0.689±0.011 | 0.388±0.005 | 0.703±0.012 |
| mistral | distilbert-base-uncased+resnet18 | 0.629±0.009 | 0.732±0.013 | 0.422±0.008 | 0.719±0.013 |
| qwen | distilbert-base-uncased+densenet121 | 0.629±0.010 | 0.703±0.010 | 0.422±0.008 | 0.719±0.013 |
| kimi | roberta-base+resnet50 | 0.629±0.012 | 0.777±0.013 | 0.422±0.007 | 0.719±0.011 |
| mistral | distilbert-base-uncased+densenet121 | 0.647±0.008 | 0.711±0.013 | 0.456±0.006 | 0.734±0.012 |
| mistral | albert-base-v2+resnet18 | 0.647±0.011 | 0.738±0.011 | 0.456±0.006 | 0.734±0.012 |
| kimi | roberta-base+shufflenet_v2_x1_0 | 0.647±0.012 | 0.688±0.011 | 0.456±0.008 | 0.734±0.010 |
| mistral | bert-base-uncased+resnet34 | 0.647±0.009 | 0.689±0.012 | 0.456±0.007 | 0.734±0.011 |
| qwen | roberta-base+efficientnet_b0 | 0.647±0.009 | 0.756±0.013 | 0.456±0.008 | 0.734±0.014 |
| qwen | bert-base-uncased+resnet18 | 0.647±0.009 | 0.736±0.015 | 0.456±0.006 | 0.734±0.010 |
| kimi | roberta-base+resnet34 | 0.647±0.011 | 0.734±0.011 | 0.456±0.006 | 0.734±0.015 |
| kimi | albert-base-v2+densenet121 | 0.667±0.013 | 0.752±0.013 | 0.493±0.010 | 0.750±0.013 |
| kimi | bert-base-uncased+resnet50 | 0.667±0.013 | 0.730±0.012 | 0.493±0.008 | 0.750±0.015 |
| qwen | roberta-base+densenet121 | 0.667±0.011 | 0.781±0.010 | 0.493±0.007 | 0.750±0.011 |
| kimi | distilbert-base-uncased+resnet34 | 0.667±0.013 | 0.854±0.017 | 0.493±0.007 | 0.750±0.013 |
| mistral | bert-base-uncased+efficientnet_b0 | 0.667±0.012 | 0.773±0.013 | 0.493±0.009 | 0.750±0.010 |
| mistral | albert-base-v2+shufflenet_v2_x1_0 | 0.667±0.013 | 0.803±0.016 | 0.493±0.008 | 0.750±0.014 |
| mistral | albert-base-v2+resnet34 | 0.667±0.010 | 0.750±0.012 | 0.493±0.009 | 0.750±0.013 |
| qwen | bert-base-uncased+resnet50 | 0.688±0.010 | 0.838±0.011 | 0.531±0.009 | 0.766±0.010 |
| kimi | albert-base-v2+resnet18 | 0.688±0.011 | 0.777±0.013 | 0.531±0.007 | 0.766±0.011 |
| mistral | albert-base-v2+densenet121 | 0.688±0.011 | 0.822±0.014 | 0.531±0.008 | 0.766±0.014 |
| qwen | distilbert-base-uncased+mobilenet_v3_small | 0.688±0.013 | 0.781±0.012 | 0.531±0.008 | 0.766±0.014 |
| mistral | distilbert-base-uncased+resnet50 | 0.688±0.013 | 0.812±0.016 | 0.531±0.011 | 0.766±0.015 |
| qwen | bert-base-uncased+resnet34 | 0.688±0.013 | 0.826±0.011 | 0.531±0.010 | 0.766±0.015 |
| kimi | bert-base-uncased+resnet18 | 0.688±0.014 | 0.818±0.013 | 0.531±0.009 | 0.766±0.014 |
| kimi | distilbert-base-uncased+efficientnet_b0 | 0.688±0.012 | 0.793±0.013 | 0.531±0.008 | 0.766±0.012 |
| kimi | distilbert-base-uncased+densenet121 | 0.688±0.013 | 0.779±0.015 | 0.531±0.008 | 0.766±0.015 |
| kimi | bert-base-uncased+efficientnet_b0 | 0.688±0.013 | 0.762±0.011 | 0.531±0.010 | 0.766±0.013 |
| qwen | roberta-base+densenet121 | 0.710±0.013 | 0.820±0.011 | 0.572±0.010 | 0.781±0.012 |
| mistral | albert-base-v2+efficientnet_b0 | 0.710±0.010 | 0.828±0.013 | 0.572±0.008 | 0.781±0.011 |
| kimi | albert-base-v2+resnet50 | 0.710±0.011 | 0.812±0.015 | 0.572±0.011 | 0.781±0.014 |
| qwen | roberta-base+resnet18 | 0.710±0.010 | 0.820±0.016 | 0.572±0.010 | 0.781±0.015 |

*(end of table)*

*(continued)*

| Src Text | Model | F1 | AUC | MCC | Balanced Acc |
|---|---|---|---|---|---|
| kimi | albert-base-v2+resnet18 | 0.710±0.011 | 0.818±0.011 | 0.572±0.010 | 0.781±0.010 |
| kimi | albert-base-v2+densenet121 | 0.710±0.012 | 0.791±0.013 | 0.572±0.010 | 0.781±0.015 |
| mistral | albert-base-v2+resnet18 | 0.733±0.013 | 0.850±0.014 | 0.616±0.009 | 0.797±0.011 |
| qwen | albert-base-v2+densenet121 | 0.571±0.009 | 0.711±0.012 | 0.331±0.007 | 0.672±0.010 |
| qwen | roberta-base+densenet121 | 0.571±0.010 | 0.684±0.010 | 0.331±0.005 | 0.672±0.011 |
| qwen | roberta-base+mobilenet_v3_small | 0.588±0.009 | 0.822±0.015 | 0.365±0.005 | 0.688±0.009 |
| qwen | distilbert-base-uncased+densenet121 | 0.588±0.009 | 0.723±0.010 | 0.365±0.006 | 0.688±0.011 |
| kimi | albert-base-v2+efficientnet_b0 | 0.588±0.009 | 0.730±0.011 | 0.365±0.006 | 0.688±0.012 |
| mistral | roberta-base+resnet34 | 0.588±0.009 | 0.740±0.012 | 0.365±0.007 | 0.688±0.013 |
| kimi | albert-base-v2+resnet34 | 0.588±0.010 | 0.785±0.011 | 0.365±0.007 | 0.688±0.011 |
| mistral | bert-base-uncased+densenet121 | 0.588±0.009 | 0.682±0.011 | 0.365±0.006 | 0.688±0.011 |
| mistral | distilbert-base-uncased+densenet121 | 0.606±0.012 | 0.709±0.011 | 0.400±0.007 | 0.703±0.010 |
| mistral | bert-base-uncased+resnet18 | 0.606±0.009 | 0.773±0.015 | 0.400±0.006 | 0.703±0.011 |
| qwen | albert-base-v2+resnet34 | 0.606±0.012 | 0.648±0.012 | 0.400±0.007 | 0.703±0.010 |
| qwen | bert-base-uncased+resnet34 | 0.606±0.008 | 0.729±0.012 | 0.400±0.005 | 0.703±0.010 |
| mistral | distilbert-base-uncased+densenet121 | 0.606±0.010 | 0.686±0.009 | 0.400±0.006 | 0.703±0.014 |
| qwen | bert-base-uncased+densenet121 | 0.606±0.009 | 0.742±0.013 | 0.400±0.005 | 0.703±0.012 |
| mistral | distilbert-base-uncased+efficientnet_b0 | 0.606±0.011 | 0.748±0.015 | 0.400±0.005 | 0.703±0.013 |
| qwen | albert-base-v2+densenet121 | 0.625±0.009 | 0.713±0.014 | 0.438±0.008 | 0.719±0.012 |
| kimi | distilbert-base-uncased+mobilenet_v3_small | 0.625±0.010 | 0.768±0.011 | 0.438±0.006 | 0.719±0.011 |
| qwen | distilbert-base-uncased+resnet34 | 0.625±0.010 | 0.709±0.010 | 0.438±0.007 | 0.719±0.013 |
| kimi | roberta-base+densenet121 | 0.625±0.010 | 0.828±0.014 | 0.438±0.006 | 0.719±0.011 |
| kimi | roberta-base+resnet18 | 0.645±0.013 | 0.812±0.016 | 0.477±0.007 | 0.734±0.015 |
| qwen | bert-base-uncased+efficientnet_b0 | 0.645±0.009 | 0.764±0.015 | 0.477±0.008 | 0.734±0.014 |
| qwen | bert-base-uncased+resnet50 | 0.645±0.009 | 0.814±0.014 | 0.477±0.009 | 0.734±0.011 |
| kimi | distilbert-base-uncased+resnet18 | 0.645±0.012 | 0.750±0.012 | 0.477±0.009 | 0.734±0.010 |
| mistral | distilbert-base-uncased+resnet50 | 0.645±0.010 | 0.748±0.014 | 0.477±0.008 | 0.734±0.012 |
| kimi | albert-base-v2+resnet18 | 0.645±0.009 | 0.701±0.009 | 0.477±0.008 | 0.734±0.013 |
| qwen | albert-base-v2+resnet34 | 0.645±0.011 | 0.803±0.013 | 0.477±0.008 | 0.734±0.013 |
| kimi | bert-base-uncased+efficientnet_b0 | 0.667±0.012 | 0.793±0.013 | 0.519±0.009 | 0.750±0.011 |
| qwen | roberta-base+resnet50 | 0.667±0.009 | 0.781±0.014 | 0.519±0.008 | 0.750±0.012 |
| kimi | distilbert-base-uncased+resnet50 | 0.667±0.010 | 0.799±0.011 | 0.519±0.007 | 0.750±0.012 |
| qwen | albert-base-v2+densenet121 | 0.667±0.011 | 0.826±0.012 | 0.519±0.010 | 0.750±0.014 |
| qwen | albert-base-v2+resnet50 | 0.690±0.011 | 0.820±0.014 | 0.564±0.008 | 0.766±0.011 |
| mistral | roberta-base+resnet50 | 0.690±0.009 | 0.805±0.014 | 0.564±0.010 | 0.766±0.013 |
| kimi | bert-base-uncased+resnet34 | 0.690±0.011 | 0.783±0.011 | 0.564±0.008 | 0.766±0.015 |
| mistral | roberta-base+densenet121 | 0.690±0.009 | 0.754±0.011 | 0.564±0.008 | 0.766±0.011 |
| kimi | albert-base-v2+efficientnet_b0 | 0.690±0.009 | 0.773±0.011 | 0.564±0.011 | 0.766±0.014 |
| mistral | bert-base-uncased+resnet34 | 0.690±0.012 | 0.783±0.015 | 0.564±0.008 | 0.766±0.013 |
| kimi | albert-base-v2+resnet50 | 0.690±0.013 | 0.783±0.012 | 0.564±0.008 | 0.766±0.012 |
| qwen | albert-base-v2+resnet34 | 0.714±0.010 | 0.807±0.014 | 0.612±0.010 | 0.781±0.016 |
| mistral | distilbert-base-uncased+resnet34 | 0.545±0.007 | 0.658±0.010 | 0.308±0.005 | 0.656±0.012 |
| mistral | roberta-base+resnet18 | 0.545±0.007 | 0.641±0.011 | 0.308±0.005 | 0.656±0.009 |
| qwen | bert-base-uncased+efficientnet_b0 | 0.562±0.009 | 0.676±0.011 | 0.344±0.006 | 0.672±0.010 |
| kimi | bert-base-uncased+densenet121 | 0.562±0.011 | 0.672±0.013 | 0.344±0.006 | 0.672±0.010 |
| kimi | distilbert-base-uncased+resnet34 | 0.581±0.010 | 0.723±0.012 | 0.381±0.006 | 0.688±0.014 |
| kimi | bert-base-uncased+densenet121 | 0.581±0.011 | 0.762±0.010 | 0.381±0.007 | 0.688±0.012 |
| mistral | roberta-base+resnet18 | 0.581±0.009 | 0.709±0.011 | 0.381±0.006 | 0.688±0.012 |
| mistral | bert-base-uncased+resnet50 | 0.581±0.008 | 0.719±0.011 | 0.381±0.007 | 0.688±0.013 |
| kimi | albert-base-v2+densenet121 | 0.600±0.010 | 0.715±0.012 | 0.421±0.006 | 0.703±0.010 |
| kimi | bert-base-uncased+resnet18 | 0.600±0.008 | 0.768±0.015 | 0.421±0.006 | 0.703±0.010 |
| mistral | bert-base-uncased+densenet121 | 0.600±0.009 | 0.719±0.014 | 0.421±0.006 | 0.703±0.010 |

*(end of table)*

*(continued)*

| Src Text | Model | F1 | AUC | MCC | Balanced Acc |
|---|---|---|---|---|---|
| qwen | albert-base-v2+resnet18 | 0.621±0.009 | 0.777±0.010 | 0.464±0.009 | 0.719±0.011 |
| mistral | bert-base-uncased+resnet18 | 0.621±0.011 | 0.727±0.012 | 0.464±0.007 | 0.719±0.012 |
| qwen | roberta-base+efficientnet_b0 | 0.621±0.010 | 0.783±0.013 | 0.464±0.007 | 0.719±0.011 |
| mistral | roberta-base+resnet18 | 0.621±0.010 | 0.775±0.014 | 0.464±0.009 | 0.719±0.014 |
| qwen | albert-base-v2+resnet34 | 0.643±0.012 | 0.814±0.016 | 0.510±0.009 | 0.734±0.013 |
| mistral | albert-base-v2+resnet34 | 0.692±0.011 | 0.799±0.013 | 0.617±0.011 | 0.766±0.014 |
| mistral | distilbert-base-uncased+resnet34 | 0.692±0.012 | 0.787±0.012 | 0.617±0.011 | 0.766±0.012 |
| kimi | albert-base-v2+mobilenet_v3_small | 0.516±0.008 | 0.783±0.014 | 0.286±0.005 | 0.641±0.012 |
| kimi | albert-base-v2+shufflenet_v2_x1_0 | 0.516±0.007 | 0.490±0.007 | 0.286±0.005 | 0.641±0.010 |
| qwen | albert-base-v2+densenet121 | 0.571±0.010 | 0.848±0.017 | 0.408±0.007 | 0.688±0.013 |
| qwen | bert-base-uncased+resnet34 | 0.571±0.009 | 0.727±0.014 | 0.408±0.005 | 0.688±0.013 |
| kimi | bert-base-uncased+resnet34 | 0.571±0.008 | 0.738±0.014 | 0.408±0.006 | 0.688±0.010 |
| mistral | albert-base-v2+resnet50 | 0.571±0.011 | 0.785±0.011 | 0.408±0.007 | 0.688±0.011 |
| kimi | roberta-base+densenet121 | 0.593±0.011 | 0.729±0.014 | 0.456±0.007 | 0.703±0.010 |
| qwen | roberta-base+resnet34 | 0.593±0.008 | 0.795±0.013 | 0.456±0.006 | 0.703±0.012 |
| qwen | distilbert-base-uncased+resnet50 | 0.615±0.010 | 0.812±0.012 | 0.508±0.008 | 0.719±0.009 |
| qwen | roberta-base+resnet18 | 0.483±0.009 | 0.635±0.012 | 0.265±0.005 | 0.625±0.012 |
| qwen | roberta-base+resnet18 | 0.519±0.008 | 0.777±0.013 | 0.350±0.006 | 0.656±0.013 |
| qwen | roberta-base+mobilenet_v3_small | 0.522±0.008 | 0.773±0.010 | 0.459±0.008 | 0.672±0.010 |
| qwen | bert-base-uncased+mobilenet_v3_small | 0.000±0.000 | 0.000±0.000 | 0.000±0.000 | 0.000±0.000 |
| qwen | bert-base-uncased+shufflenet_v2_x1_0 | 0.000±0.000 | 0.000±0.000 | 0.000±0.000 | 0.000±0.000 |
| qwen | bert-base-uncased+resnet50 | 0.500±0.008 | 0.348±0.006 | 0.000±0.000 | 0.500±0.010 |
| qwen | roberta-base+efficientnet_b0 | 0.000±0.000 | 0.000±0.000 | 0.000±0.000 | 0.000±0.000 |
| qwen | bert-base-uncased+mobilenet_v3_small | 0.500±0.009 | 0.558±0.008 | 0.000±0.000 | 0.500±0.010 |
| kimi | roberta-base+resnet18 | 0.561±0.011 | 0.656±0.010 | 0.292±0.005 | 0.609±0.010 |
| qwen | bert-base-uncased+densenet121 | 0.542±0.010 | 0.607±0.008 | 0.219±0.004 | 0.609±0.009 |
| kimi | bert-base-uncased+shufflenet_v2_x1_0 | 0.558±0.011 | 0.596±0.011 | 0.267±0.004 | 0.641±0.012 |
| kimi | bert-base-uncased+shufflenet_v2_x1_0 | 0.000±0.000 | 0.000±0.000 | 0.000±0.000 | 0.000±0.000 |
| kimi | bert-base-uncased+vgg16 | 0.500±0.009 | 0.480±0.009 | 0.000±0.000 | 0.500±0.007 |
| qwen | bert-base-uncased+efficientnet_b0 | 0.000±0.000 | 0.000±0.000 | 0.000±0.000 | 0.000±0.000 |
| qwen | bert-base-uncased+efficientnet_b0 | 0.500±0.009 | 0.478±0.007 | 0.000±0.000 | 0.500±0.009 |
| qwen | roberta-base+densenet121 | 0.585±0.011 | 0.629±0.012 | 0.324±0.006 | 0.672±0.011 |
| qwen | roberta-base+resnet50 | 0.000±0.000 | 0.000±0.000 | 0.000±0.000 | 0.000±0.000 |
| qwen | bert-base-uncased+resnet50 | 0.500±0.009 | 0.639±0.012 | 0.000±0.000 | 0.500±0.007 |
| qwen | roberta-base+resnet50 | 0.526±0.008 | 0.607±0.009 | 0.237±0.003 | 0.625±0.012 |
| qwen | bert-base-uncased+vgg16 | 0.000±0.000 | 0.000±0.000 | 0.000±0.000 | 0.000±0.000 |
| qwen | bert-base-uncased+shufflenet_v2_x1_0 | 0.000±0.000 | 0.000±0.000 | 0.000±0.000 | 0.000±0.000 |
| kimi | bert-base-uncased+vgg16 | 0.508±0.008 | 0.604±0.011 | 0.096±0.001 | 0.531±0.011 |
| qwen | roberta-base+resnet34 | 0.609±0.009 | 0.719±0.010 | 0.365±0.005 | 0.688±0.011 |
| qwen | roberta-base+resnet34 | 0.564±0.008 | 0.672±0.013 | 0.295±0.006 | 0.656±0.012 |
| kimi | bert-base-uncased+vgg16 | 0.000±0.000 | 0.000±0.000 | 0.000±0.000 | 0.000±0.000 |
| qwen | roberta-base+resnet18 | 0.638±0.009 | 0.715±0.010 | 0.431±0.008 | 0.719±0.012 |
| qwen | bert-base-uncased+shufflenet_v2_x1_0 | 0.424±0.006 | 0.518±0.008 | 0.123±0.002 | 0.562±0.008 |
| kimi | bert-base-uncased+vgg16 | 0.000±0.000 | 0.000±0.000 | 0.000±0.000 | 0.000±0.000 |
| qwen | bert-base-uncased+vgg16 | 0.000±0.000 | 0.000±0.000 | 0.000±0.000 | 0.000±0.000 |
| qwen | bert-base-uncased+vgg16 | 0.000±0.000 | 0.000±0.000 | 0.000±0.000 | 0.000±0.000 |
| qwen | bert-base-uncased+vgg16 | 0.448±0.007 | 0.551±0.011 | -0.134±0.002 | 0.453±0.006 |
| qwen | bert-base-uncased+shufflenet_v2_x1_0 | 0.500±0.007 | 0.773±0.012 | 0.000±0.000 | 0.500±0.009 |
| kimi | distilbert-base-uncased+vgg16 | 0.000±0.000 | 0.000±0.000 | 0.000±0.000 | 0.000±0.000 |
| qwen | roberta-base+efficientnet_b0 | 0.000±0.000 | 0.000±0.000 | 0.000±0.000 | 0.000±0.000 |
| qwen | albert-base-v2+resnet50 | 0.475±0.008 | 0.609±0.010 | -0.048±0.001 | 0.484±0.006 |
| qwen | distilbert-base-uncased+shufflenet_v2_x1_0 | 0.000±0.000 | 0.000±0.000 | 0.000±0.000 | 0.000±0.000 |

*(end of table)*

*(continued)*

| Src Text | Model | F1 | AUC | MCC | Balanced Acc |
|---|---|---|---|---|---|
| qwen | distilbert-base-uncased+shufflenet_v2_x1_0 | 0.553±0.009 | 0.637±0.012 | 0.246±0.005 | 0.625±0.010 |
| qwen | distilbert-base-uncased+shufflenet_v2_x1_0 | 0.000±0.000 | 0.000±0.000 | 0.000±0.000 | 0.000±0.000 |
| qwen | distilbert-base-uncased+vgg16 | 0.000±0.000 | 0.000±0.000 | 0.000±0.000 | 0.000±0.000 |
| qwen | distilbert-base-uncased+vgg16 | 0.000±0.000 | 0.000±0.000 | 0.000±0.000 | 0.000±0.000 |
| qwen | distilbert-base-uncased+vgg16 | 0.500±0.007 | 0.555±0.008 | 0.000±0.000 | 0.500±0.010 |
| qwen | distilbert-base-uncased+vgg16 | 0.000±0.000 | 0.000±0.000 | 0.000±0.000 | 0.000±0.000 |
| kimi | bert-base-uncased+resnet50 | 0.531±0.010 | 0.529±0.008 | 0.191±0.004 | 0.594±0.011 |
| kimi | bert-base-uncased+resnet50 | 0.591±0.009 | 0.758±0.014 | 0.329±0.005 | 0.672±0.012 |
| qwen | albert-base-v2+resnet18 | 0.605±0.011 | 0.760±0.011 | 0.356±0.006 | 0.688±0.012 |
| kimi | bert-base-uncased+resnet34 | 0.553±0.008 | 0.656±0.011 | 0.246±0.003 | 0.625±0.008 |
| qwen | albert-base-v2+resnet50 | 0.000±0.000 | 0.000±0.000 | 0.000±0.000 | 0.000±0.000 |
| qwen | distilbert-base-uncased+mobilenet_v3_small | 0.000±0.000 | 0.000±0.000 | 0.000±0.000 | 0.000±0.000 |
| qwen | albert-base-v2+efficientnet_b0 | 0.000±0.000 | 0.000±0.000 | 0.000±0.000 | 0.000±0.000 |
| qwen | albert-base-v2+efficientnet_b0 | 0.500±0.007 | 0.565±0.010 | 0.000±0.000 | 0.500±0.009 |
| qwen | albert-base-v2+mobilenet_v3_small | 0.000±0.000 | 0.000±0.000 | 0.000±0.000 | 0.000±0.000 |
| qwen | albert-base-v2+mobilenet_v3_small | 0.500±0.009 | 0.473±0.007 | 0.000±0.000 | 0.500±0.007 |
| qwen | albert-base-v2+shufflenet_v2_x1_0 | 0.000±0.000 | 0.000±0.000 | 0.000±0.000 | 0.000±0.000 |
| qwen | albert-base-v2+shufflenet_v2_x1_0 | 0.000±0.000 | 0.000±0.000 | 0.000±0.000 | 0.000±0.000 |
| qwen | albert-base-v2+shufflenet_v2_x1_0 | 0.488±0.009 | 0.605±0.010 | 0.147±0.003 | 0.578±0.010 |
| qwen | albert-base-v2+shufflenet_v2_x1_0 | 0.500±0.009 | 0.438±0.007 | 0.000±0.000 | 0.500±0.009 |
| qwen | albert-base-v2+vgg16 | 0.500±0.009 | 0.631±0.010 | 0.000±0.000 | 0.500±0.008 |
| qwen | albert-base-v2+vgg16 | 0.000±0.000 | 0.000±0.000 | 0.000±0.000 | 0.000±0.000 |
| qwen | albert-base-v2+vgg16 | 0.500±0.010 | 0.551±0.009 | 0.000±0.000 | 0.500±0.007 |
| qwen | distilbert-base-uncased+shufflenet_v2_x1_0 | 0.500±0.008 | 0.508±0.008 | 0.000±0.000 | 0.500±0.007 |
| qwen | distilbert-base-uncased+mobilenet_v3_small | 0.000±0.000 | 0.000±0.000 | 0.000±0.000 | 0.000±0.000 |
| qwen | roberta-base+mobilenet_v3_small | 0.000±0.000 | 0.000±0.000 | 0.000±0.000 | 0.000±0.000 |
| kimi | bert-base-uncased+mobilenet_v3_small | 0.500±0.008 | 0.525±0.008 | 0.000±0.000 | 0.500±0.007 |
| qwen | roberta-base+mobilenet_v3_small | 0.000±0.000 | 0.000±0.000 | 0.000±0.000 | 0.000±0.000 |
| qwen | roberta-base+shufflenet_v2_x1_0 | 0.600±0.012 | 0.729±0.012 | 0.354±0.005 | 0.688±0.010 |
| qwen | roberta-base+shufflenet_v2_x1_0 | 0.000±0.000 | 0.000±0.000 | 0.000±0.000 | 0.000±0.000 |
| qwen | roberta-base+shufflenet_v2_x1_0 | 0.438±0.008 | 0.578±0.009 | 0.156±0.002 | 0.578±0.009 |
| qwen | roberta-base+shufflenet_v2_x1_0 | 0.415±0.008 | 0.436±0.007 | -0.140±0.002 | 0.438±0.006 |
| qwen | roberta-base+vgg16 | 0.500±0.007 | 0.523±0.010 | 0.000±0.000 | 0.500±0.008 |
| qwen | roberta-base+vgg16 | 0.000±0.000 | 0.000±0.000 | 0.000±0.000 | 0.000±0.000 |
| qwen | roberta-base+vgg16 | 0.000±0.000 | 0.000±0.000 | 0.000±0.000 | 0.000±0.000 |
| qwen | roberta-base+vgg16 | 0.100±0.001 | 0.549±0.009 | -0.053±0.001 | 0.484±0.008 |
| kimi | bert-base-uncased+shufflenet_v2_x1_0 | 0.000±0.000 | 0.000±0.000 | 0.000±0.000 | 0.000±0.000 |
| kimi | bert-base-uncased+shufflenet_v2_x1_0 | 0.000±0.000 | 0.000±0.000 | 0.000±0.000 | 0.000±0.000 |
| qwen | distilbert-base-uncased+resnet18 | 0.553±0.009 | 0.682±0.011 | 0.246±0.004 | 0.625±0.009 |
| qwen | distilbert-base-uncased+efficientnet_b0 | 0.000±0.000 | 0.000±0.000 | 0.000±0.000 | 0.000±0.000 |
| kimi | bert-base-uncased+mobilenet_v3_small | 0.000±0.000 | 0.000±0.000 | 0.000±0.000 | 0.000±0.000 |
| kimi | bert-base-uncased+mobilenet_v3_small | 0.500±0.008 | 0.830±0.014 | 0.000±0.000 | 0.500±0.008 |
| kimi | bert-base-uncased+efficientnet_b0 | 0.000±0.000 | 0.000±0.000 | 0.000±0.000 | 0.000±0.000 |
| kimi | roberta-base+resnet50 | 0.528±0.007 | 0.678±0.010 | 0.175±0.003 | 0.578±0.009 |
| kimi | bert-base-uncased+efficientnet_b0 | 0.000±0.000 | 0.000±0.000 | 0.000±0.000 | 0.000±0.000 |
| qwen | distilbert-base-uncased+resnet50 | 0.550±0.010 | 0.627±0.010 | 0.265±0.005 | 0.641±0.010 |
| qwen | distilbert-base-uncased+resnet50 | 0.000±0.000 | 0.000±0.000 | 0.000±0.000 | 0.000±0.000 |
| qwen | distilbert-base-uncased+densenet121 | 0.550±0.010 | 0.605±0.008 | 0.265±0.004 | 0.641±0.009 |
| kimi | bert-base-uncased+densenet121 | 0.545±0.010 | 0.541±0.010 | 0.239±0.004 | 0.625±0.010 |
| kimi | bert-base-uncased+densenet121 | 0.549±0.008 | 0.672±0.012 | 0.232±0.003 | 0.609±0.008 |
| qwen | distilbert-base-uncased+efficientnet_b0 | 0.000±0.000 | 0.000±0.000 | 0.000±0.000 | 0.000±0.000 |
| kimi | roberta-base+resnet34 | 0.564±0.008 | 0.602±0.010 | 0.295±0.005 | 0.656±0.009 |

*(end of table)*

*(continued)*

| Src Text | Model | F1 | AUC | MCC | Balanced Acc |
|---|---|---|---|---|---|
| kimi | roberta-base+efficientnet_b0 | 0.000±0.000 | 0.000±0.000 | 0.000±0.000 | 0.000±0.000 |
| kimi | roberta-base+resnet50 | 0.000±0.000 | 0.000±0.000 | 0.000±0.000 | 0.000±0.000 |
| kimi | distilbert-base-uncased+mobilenet_v3_small | 0.000±0.000 | 0.000±0.000 | 0.000±0.000 | 0.000±0.000 |
| mistral | bert-base-uncased+efficientnet_b0 | 0.500±0.010 | 0.465±0.008 | 0.000±0.000 | 0.500±0.008 |
| mistral | bert-base-uncased+efficientnet_b0 | 0.500±0.010 | 0.472±0.009 | 0.000±0.000 | 0.500±0.009 |
| mistral | bert-base-uncased+mobilenet_v3_small | 0.000±0.000 | 0.000±0.000 | 0.000±0.000 | 0.000±0.000 |
| mistral | bert-base-uncased+mobilenet_v3_small | 0.578±0.011 | 0.645±0.009 | 0.301±0.004 | 0.656±0.009 |
| mistral | bert-base-uncased+mobilenet_v3_small | 0.410±0.007 | 0.471±0.008 | 0.029±0.001 | 0.516±0.008 |
| mistral | bert-base-uncased+mobilenet_v3_small | 0.500±0.010 | 0.456±0.008 | 0.000±0.000 | 0.500±0.010 |
| mistral | bert-base-uncased+shufflenet_v2_x1_0 | 0.500±0.008 | 0.590±0.009 | 0.000±0.000 | 0.500±0.009 |
| mistral | bert-base-uncased+shufflenet_v2_x1_0 | 0.000±0.000 | 0.000±0.000 | 0.000±0.000 | 0.000±0.000 |
| mistral | bert-base-uncased+shufflenet_v2_x1_0 | 0.468±0.009 | 0.559±0.010 | 0.062±0.001 | 0.531±0.009 |
| mistral | bert-base-uncased+shufflenet_v2_x1_0 | 0.474±0.007 | 0.668±0.013 | 0.148±0.002 | 0.578±0.008 |
| mistral | bert-base-uncased+vgg16 | 0.000±0.000 | 0.000±0.000 | 0.000±0.000 | 0.000±0.000 |
| mistral | bert-base-uncased+vgg16 | 0.000±0.000 | 0.000±0.000 | 0.000±0.000 | 0.000±0.000 |
| mistral | bert-base-uncased+vgg16 | 0.000±0.000 | 0.000±0.000 | 0.000±0.000 | 0.000±0.000 |
| mistral | bert-base-uncased+vgg16 | 0.000±0.000 | 0.000±0.000 | 0.000±0.000 | 0.000±0.000 |
| mistral | roberta-base+resnet18 | 0.636±0.011 | 0.793±0.015 | 0.418±0.006 | 0.719±0.010 |
| kimi | distilbert-base-uncased+efficientnet_b0 | 0.000±0.000 | 0.000±0.000 | 0.000±0.000 | 0.000±0.000 |
| kimi | distilbert-base-uncased+efficientnet_b0 | 0.000±0.000 | 0.000±0.000 | 0.000±0.000 | 0.000±0.000 |
| mistral | roberta-base+resnet34 | 0.525±0.010 | 0.566±0.008 | 0.183±0.003 | 0.547±0.010 |
| mistral | roberta-base+resnet50 | 0.483±0.007 | 0.449±0.007 | 0.000±0.000 | 0.500±0.007 |
| mistral | roberta-base+resnet50 | 0.489±0.007 | 0.598±0.010 | 0.120±0.002 | 0.562±0.009 |
| mistral | roberta-base+densenet121 | 0.577±0.008 | 0.787±0.016 | 0.306±0.004 | 0.641±0.013 |
| kimi | distilbert-base-uncased+densenet121 | 0.500±0.007 | 0.631±0.011 | 0.177±0.003 | 0.594±0.010 |
| kimi | distilbert-base-uncased+resnet50 | 0.564±0.011 | 0.650±0.011 | 0.295±0.004 | 0.656±0.012 |
| mistral | roberta-base+efficientnet_b0 | 0.000±0.000 | 0.000±0.000 | 0.000±0.000 | 0.000±0.000 |
| mistral | roberta-base+efficientnet_b0 | 0.000±0.000 | 0.000±0.000 | 0.000±0.000 | 0.000±0.000 |
| mistral | roberta-base+mobilenet_v3_small | 0.000±0.000 | 0.000±0.000 | 0.000±0.000 | 0.000±0.000 |
| mistral | roberta-base+mobilenet_v3_small | 0.500±0.008 | 0.516±0.007 | 0.000±0.000 | 0.500±0.007 |
| kimi | distilbert-base-uncased+mobilenet_v3_small | 0.000±0.000 | 0.000±0.000 | 0.000±0.000 | 0.000±0.000 |
| kimi | distilbert-base-uncased+mobilenet_v3_small | 0.000±0.000 | 0.000±0.000 | 0.000±0.000 | 0.000±0.000 |
| qwen | bert-base-uncased+resnet18 | 0.564±0.009 | 0.641±0.010 | 0.295±0.004 | 0.656±0.011 |
| mistral | bert-base-uncased+resnet50 | 0.542±0.007 | 0.537±0.009 | 0.219±0.004 | 0.609±0.010 |
| kimi | albert-base-v2+resnet18 | 0.622±0.012 | 0.750±0.010 | 0.392±0.005 | 0.703±0.009 |
| kimi | distilbert-base-uncased+vgg16 | 0.000±0.000 | 0.000±0.000 | 0.000±0.000 | 0.000±0.000 |
| kimi | albert-base-v2+resnet34 | 0.533±0.008 | 0.652±0.009 | 0.213±0.004 | 0.562±0.007 |
| kimi | albert-base-v2+resnet34 | 0.517±0.008 | 0.611±0.010 | 0.134±0.002 | 0.547±0.009 |
| kimi | albert-base-v2+resnet50 | 0.500±0.008 | 0.533±0.009 | 0.149±0.002 | 0.578±0.010 |
| kimi | albert-base-v2+resnet50 | 0.000±0.000 | 0.000±0.000 | 0.000±0.000 | 0.000±0.000 |
| kimi | distilbert-base-uncased+vgg16 | 0.000±0.000 | 0.000±0.000 | 0.000±0.000 | 0.000±0.000 |
| kimi | distilbert-base-uncased+vgg16 | 0.000±0.000 | 0.000±0.000 | 0.000±0.000 | 0.000±0.000 |
| kimi | albert-base-v2+efficientnet_b0 | 0.000±0.000 | 0.000±0.000 | 0.000±0.000 | 0.000±0.000 |
| kimi | albert-base-v2+efficientnet_b0 | 0.000±0.000 | 0.000±0.000 | 0.000±0.000 | 0.000±0.000 |
| kimi | albert-base-v2+mobilenet_v3_small | 0.000±0.000 | 0.000±0.000 | 0.000±0.000 | 0.000±0.000 |
| kimi | albert-base-v2+mobilenet_v3_small | 0.500±0.010 | 0.541±0.011 | 0.000±0.000 | 0.500±0.008 |
| kimi | albert-base-v2+mobilenet_v3_small | 0.000±0.000 | 0.000±0.000 | 0.000±0.000 | 0.000±0.000 |
| kimi | albert-base-v2+shufflenet_v2_x1_0 | 0.500±0.009 | 0.439±0.008 | 0.000±0.000 | 0.500±0.007 |
| kimi | albert-base-v2+shufflenet_v2_x1_0 | 0.000±0.000 | 0.000±0.000 | 0.000±0.000 | 0.000±0.000 |
| kimi | albert-base-v2+shufflenet_v2_x1_0 | 0.488±0.007 | 0.562±0.010 | 0.147±0.003 | 0.578±0.012 |
| kimi | albert-base-v2+vgg16 | 0.500±0.009 | 0.465±0.008 | 0.000±0.000 | 0.500±0.008 |
| kimi | distilbert-base-uncased+shufflenet_v2_x1_0 | 0.000±0.000 | 0.000±0.000 | 0.000±0.000 | 0.000±0.000 |

*(end of table)*

*(continued)*

| Src Text | Model | F1 | AUC | MCC | Balanced Acc |
|---|---|---|---|---|---|
| kimi | albert-base-v2+vgg16 | 0.000±0.000 | 0.000±0.000 | 0.000±0.000 | 0.000±0.000 |
| kimi | albert-base-v2+vgg16 | 0.459±0.006 | 0.508±0.008 | -0.183±0.003 | 0.453±0.009 |
| kimi | distilbert-base-uncased+shufflenet_v2_x1_0 | 0.529±0.007 | 0.615±0.012 | 0.274±0.005 | 0.641±0.009 |
| mistral | bert-base-uncased+resnet18 | 0.500±0.008 | 0.613±0.008 | 0.102±0.002 | 0.547±0.010 |
| kimi | distilbert-base-uncased+shufflenet_v2_x1_0 | 0.000±0.000 | 0.000±0.000 | 0.000±0.000 | 0.000±0.000 |
| mistral | bert-base-uncased+resnet34 | 0.571±0.008 | 0.684±0.012 | 0.316±0.004 | 0.625±0.010 |
| mistral | bert-base-uncased+resnet34 | 0.600±0.012 | 0.666±0.010 | 0.354±0.007 | 0.688±0.010 |
| kimi | distilbert-base-uncased+shufflenet_v2_x1_0 | 0.000±0.000 | 0.000±0.000 | 0.000±0.000 | 0.000±0.000 |
| mistral | bert-base-uncased+resnet50 | 0.512±0.007 | 0.586±0.010 | 0.178±0.002 | 0.594±0.012 |
| mistral | roberta-base+mobilenet_v3_small | 0.000±0.000 | 0.000±0.000 | 0.000±0.000 | 0.000±0.000 |
| mistral | roberta-base+mobilenet_v3_small | 0.000±0.000 | 0.000±0.000 | 0.000±0.000 | 0.000±0.000 |
| mistral | roberta-base+shufflenet_v2_x1_0 | 0.000±0.000 | 0.000±0.000 | 0.000±0.000 | 0.000±0.000 |
| mistral | distilbert-base-uncased+vgg16 | 0.000±0.000 | 0.000±0.000 | 0.000±0.000 | 0.000±0.000 |
| kimi | roberta-base+shufflenet_v2_x1_0 | 0.500±0.009 | 0.357±0.007 | 0.000±0.000 | 0.500±0.008 |
| kimi | roberta-base+shufflenet_v2_x1_0 | 0.000±0.000 | 0.000±0.000 | 0.000±0.000 | 0.000±0.000 |
| mistral | albert-base-v2+resnet18 | 0.619±0.008 | 0.697±0.009 | 0.384±0.007 | 0.703±0.014 |
| kimi | roberta-base+shufflenet_v2_x1_0 | 0.000±0.000 | 0.000±0.000 | 0.000±0.000 | 0.000±0.000 |
| mistral | albert-base-v2+resnet34 | 0.529±0.007 | 0.758±0.011 | 0.274±0.005 | 0.641±0.008 |
| kimi | roberta-base+mobilenet_v3_small | 0.500±0.009 | 0.478±0.008 | 0.000±0.000 | 0.500±0.007 |
| kimi | roberta-base+mobilenet_v3_small | 0.500±0.008 | 0.663±0.011 | 0.000±0.000 | 0.500±0.007 |
| mistral | albert-base-v2+resnet50 | 0.500±0.008 | 0.596±0.010 | 0.000±0.000 | 0.500±0.007 |
| mistral | albert-base-v2+resnet50 | 0.500±0.007 | 0.676±0.013 | 0.000±0.000 | 0.500±0.009 |
| kimi | roberta-base+mobilenet_v3_small | 0.508±0.009 | 0.701±0.012 | 0.103±0.002 | 0.516±0.008 |
| kimi | roberta-base+efficientnet_b0 | 0.000±0.000 | 0.000±0.000 | 0.000±0.000 | 0.000±0.000 |
| mistral | albert-base-v2+densenet121 | 0.585±0.011 | 0.664±0.012 | 0.324±0.005 | 0.672±0.011 |
| mistral | albert-base-v2+densenet121 | 0.636±0.009 | 0.748±0.012 | 0.418±0.008 | 0.719±0.010 |
| mistral | albert-base-v2+efficientnet_b0 | 0.000±0.000 | 0.000±0.000 | 0.000±0.000 | 0.000±0.000 |
| mistral | albert-base-v2+efficientnet_b0 | 0.000±0.000 | 0.000±0.000 | 0.000±0.000 | 0.000±0.000 |
| mistral | albert-base-v2+mobilenet_v3_small | 0.542±0.010 | 0.803±0.014 | 0.241±0.003 | 0.578±0.008 |
| mistral | albert-base-v2+mobilenet_v3_small | 0.542±0.008 | 0.656±0.012 | 0.241±0.005 | 0.578±0.011 |
| mistral | albert-base-v2+mobilenet_v3_small | 0.190±0.003 | 0.510±0.007 | 0.048±0.001 | 0.516±0.009 |
| mistral | albert-base-v2+mobilenet_v3_small | 0.000±0.000 | 0.000±0.000 | 0.000±0.000 | 0.000±0.000 |
| mistral | albert-base-v2+shufflenet_v2_x1_0 | 0.605±0.008 | 0.705±0.014 | 0.356±0.005 | 0.688±0.012 |
| mistral | albert-base-v2+shufflenet_v2_x1_0 | 0.000±0.000 | 0.000±0.000 | 0.000±0.000 | 0.000±0.000 |
| mistral | albert-base-v2+shufflenet_v2_x1_0 | 0.526±0.010 | 0.584±0.010 | 0.237±0.004 | 0.625±0.012 |
| kimi | roberta-base+densenet121 | 0.537±0.008 | 0.676±0.012 | 0.236±0.005 | 0.625±0.010 |
| mistral | albert-base-v2+vgg16 | 0.500±0.009 | 0.535±0.010 | 0.000±0.000 | 0.500±0.010 |
| mistral | albert-base-v2+vgg16 | 0.000±0.000 | 0.000±0.000 | 0.000±0.000 | 0.000±0.000 |
| mistral | albert-base-v2+vgg16 | 0.000±0.000 | 0.000±0.000 | 0.000±0.000 | 0.000±0.000 |
| mistral | albert-base-v2+vgg16 | 0.000±0.000 | 0.000±0.000 | 0.000±0.000 | 0.000±0.000 |
| mistral | roberta-base+shufflenet_v2_x1_0 | 0.000±0.000 | 0.000±0.000 | 0.000±0.000 | 0.000±0.000 |
| mistral | distilbert-base-uncased+vgg16 | 0.000±0.000 | 0.000±0.000 | 0.000±0.000 | 0.000±0.000 |
| mistral | distilbert-base-uncased+vgg16 | 0.000±0.000 | 0.000±0.000 | 0.000±0.000 | 0.000±0.000 |
| mistral | distilbert-base-uncased+vgg16 | 0.571±0.009 | 0.723±0.011 | 0.296±0.005 | 0.656±0.011 |
| mistral | roberta-base+shufflenet_v2_x1_0 | 0.638±0.010 | 0.773±0.013 | 0.431±0.006 | 0.719±0.011 |
| mistral | roberta-base+shufflenet_v2_x1_0 | 0.474±0.007 | 0.586±0.011 | 0.148±0.002 | 0.578±0.008 |
| mistral | roberta-base+vgg16 | 0.000±0.000 | 0.000±0.000 | 0.000±0.000 | 0.000±0.000 |
| mistral | roberta-base+vgg16 | 0.000±0.000 | 0.000±0.000 | 0.000±0.000 | 0.000±0.000 |
| mistral | roberta-base+vgg16 | 0.000±0.000 | 0.000±0.000 | 0.000±0.000 | 0.000±0.000 |
| mistral | roberta-base+vgg16 | 0.000±0.000 | 0.000±0.000 | 0.000±0.000 | 0.000±0.000 |
| kimi | distilbert-base-uncased+resnet34 | 0.619±0.011 | 0.768±0.012 | 0.384±0.007 | 0.703±0.011 |
| mistral | distilbert-base-uncased+resnet18 | 0.596±0.012 | 0.736±0.013 | 0.339±0.007 | 0.672±0.009 |

*(end of table)*

*(continued)*

| Src Text | Model | F1 | AUC | MCC | Balanced Acc |
|---|---|---|---|---|---|
| kimi | distilbert-base-uncased+resnet18 | 0.605±0.010 | 0.752±0.013 | 0.356±0.005 | 0.688±0.012 |
| mistral | distilbert-base-uncased+resnet34 | 0.564±0.011 | 0.650±0.012 | 0.295±0.004 | 0.656±0.012 |
| kimi | roberta-base+vgg16 | 0.000±0.000 | 0.000±0.000 | 0.000±0.000 | 0.000±0.000 |
| mistral | distilbert-base-uncased+resnet50 | 0.500±0.007 | 0.422±0.006 | 0.000±0.000 | 0.500±0.008 |
| mistral | distilbert-base-uncased+resnet50 | 0.000±0.000 | 0.000±0.000 | 0.000±0.000 | 0.000±0.000 |
| kimi | roberta-base+vgg16 | 0.000±0.000 | 0.000±0.000 | 0.000±0.000 | 0.000±0.000 |
| mistral | distilbert-base-uncased+densenet121 | 0.600±0.011 | 0.672±0.011 | 0.354±0.007 | 0.688±0.013 |
| kimi | roberta-base+vgg16 | 0.000±0.000 | 0.000±0.000 | 0.000±0.000 | 0.000±0.000 |
| mistral | distilbert-base-uncased+efficientnet_b0 | 0.000±0.000 | 0.000±0.000 | 0.000±0.000 | 0.000±0.000 |
| mistral | distilbert-base-uncased+efficientnet_b0 | 0.000±0.000 | 0.000±0.000 | 0.000±0.000 | 0.000±0.000 |
| kimi | roberta-base+vgg16 | 0.000±0.000 | 0.000±0.000 | 0.000±0.000 | 0.000±0.000 |
| mistral | distilbert-base-uncased+mobilenet_v3_small | 0.000±0.000 | 0.000±0.000 | 0.000±0.000 | 0.000±0.000 |
| mistral | distilbert-base-uncased+mobilenet_v3_small | 0.000±0.000 | 0.000±0.000 | 0.000±0.000 | 0.000±0.000 |
| mistral | distilbert-base-uncased+shufflenet_v2_x1_0 | 0.000±0.000 | 0.000±0.000 | 0.000±0.000 | 0.000±0.000 |
| mistral | distilbert-base-uncased+shufflenet_v2_x1_0 | 0.000±0.000 | 0.000±0.000 | 0.000±0.000 | 0.000±0.000 |
| mistral | distilbert-base-uncased+shufflenet_v2_x1_0 | 0.000±0.000 | 0.000±0.000 | 0.000±0.000 | 0.000±0.000 |
| mistral | distilbert-base-uncased+shufflenet_v2_x1_0 | 0.000±0.000 | 0.000±0.000 | 0.000±0.000 | 0.000±0.000 |
| qwen | albert-base-v2+vgg16 | 0.000±0.000 | 0.000±0.000 | 0.000±0.000 | 0.000±0.000 |
| qwen | bert-base-uncased+mobilenet_v3_small | 0.211±0.004 | 0.656±0.010 | 0.183±0.003 | 0.547±0.010 |
| qwen | albert-base-v2+mobilenet_v3_small | 0.211±0.003 | 0.635±0.013 | 0.183±0.003 | 0.547±0.007 |
| mistral | distilbert-base-uncased+mobilenet_v3_small | 0.118±0.002 | 0.654±0.010 | 0.206±0.004 | 0.531±0.010 |

*(end of table)*

**Table S4.** Multimodal results using Feature concat

| Model | Init | F1 | AUC | MCC | Balanced Acc |
|---|---|---|---|---|---|
| mistral | albert-base-v2+resnet50 | 0.698±0.011 | 0.850±0.015 | 0.535±0.008 | 0.781±0.014 |
| kimi | roberta-base+resnet34 | 0.732±0.014 | 0.830±0.015 | 0.590±0.011 | 0.812±0.014 |
| kimi | bert-base-uncased+efficientnet_b0 | 0.667±0.012 | 0.779±0.012 | 0.473±0.007 | 0.750±0.013 |
| mistral | roberta-base+efficientnet_b0 | 0.667±0.011 | 0.707±0.010 | 0.473±0.009 | 0.750±0.014 |
| qwen | bert-base-uncased+densenet121 | 0.683±0.010 | 0.768±0.014 | 0.501±0.007 | 0.766±0.011 |
| mistral | bert-base-uncased+efficientnet_b0 | 0.683±0.010 | 0.768±0.014 | 0.501±0.008 | 0.766±0.015 |
| kimi | albert-base-v2+resnet50 | 0.700±0.009 | 0.779±0.016 | 0.530±0.010 | 0.781±0.015 |
| qwen | distilbert-base-uncased+resnet50 | 0.700±0.013 | 0.779±0.012 | 0.530±0.007 | 0.781±0.010 |
| kimi | albert-base-v2+densenet121 | 0.737±0.013 | 0.809±0.013 | 0.591±0.009 | 0.812±0.015 |
| qwen | albert-base-v2+resnet50 | 0.737±0.013 | 0.738±0.014 | 0.591±0.011 | 0.812±0.011 |
| qwen | roberta-base+resnet50 | 0.737±0.010 | 0.811±0.012 | 0.591±0.011 | 0.812±0.012 |
| qwen | roberta-base+resnet34 | 0.737±0.015 | 0.789±0.015 | 0.591±0.010 | 0.812±0.015 |
| mistral | roberta-base+densenet121 | 0.634±0.009 | 0.713±0.011 | 0.413±0.007 | 0.719±0.013 |
| kimi | bert-base-uncased+densenet121 | 0.634±0.012 | 0.738±0.014 | 0.413±0.007 | 0.719±0.010 |
| kimi | albert-base-v2+mobilenet_v3_small | 0.650±0.009 | 0.748±0.014 | 0.442±0.007 | 0.734±0.014 |
| qwen | bert-base-uncased+vgg16 | 0.650±0.009 | 0.754±0.014 | 0.442±0.007 | 0.734±0.012 |
| qwen | bert-base-uncased+efficientnet_b0 | 0.650±0.010 | 0.732±0.010 | 0.442±0.006 | 0.734±0.010 |
| kimi | roberta-base+efficientnet_b0 | 0.650±0.011 | 0.771±0.015 | 0.442±0.009 | 0.734±0.011 |
| mistral | roberta-base+resnet34 | 0.650±0.010 | 0.773±0.013 | 0.442±0.008 | 0.734±0.013 |
| mistral | roberta-base+mobilenet_v3_small | 0.650±0.010 | 0.764±0.014 | 0.442±0.007 | 0.734±0.015 |
| mistral | bert-base-uncased+vgg16 | 0.667±0.012 | 0.787±0.012 | 0.472±0.008 | 0.750±0.013 |
| kimi | distilbert-base-uncased+resnet50 | 0.667±0.009 | 0.828±0.016 | 0.472±0.009 | 0.750±0.011 |
| qwen | distilbert-base-uncased+mobilenet_v3_small | 0.667±0.010 | 0.783±0.012 | 0.472±0.007 | 0.750±0.013 |
| qwen | distilbert-base-uncased+vgg16 | 0.684±0.013 | 0.855±0.012 | 0.503±0.009 | 0.766±0.011 |

*(end of table)*

*(continued)*

| Model | Init | F1 | AUC | MCC | Balanced Acc |
|-------|------|-----|-----|-----|--------------|
| kimi | albert-base-v2+mobilenet_v3_small | 0.684±0.011 | 0.785±0.015 | 0.503±0.008 | 0.766±0.014 |
| mistral | albert-base-v2+densenet121 | 0.684±0.011 | 0.777±0.010 | 0.503±0.009 | 0.766±0.013 |
| qwen | albert-base-v2+mobilenet_v3_small | 0.703±0.010 | 0.748±0.011 | 0.535±0.007 | 0.781±0.014 |
| mistral | roberta-base+resnet34 | 0.703±0.012 | 0.771±0.015 | 0.535±0.010 | 0.781±0.013 |
| mistral | distilbert-base-uncased+resnet50 | 0.703±0.012 | 0.812±0.016 | 0.535±0.010 | 0.781±0.015 |
| mistral | albert-base-v2+resnet50 | 0.722±0.010 | 0.828±0.014 | 0.568±0.010 | 0.797±0.012 |
| mistral | albert-base-v2+resnet34 | 0.722±0.012 | 0.777±0.014 | 0.568±0.010 | 0.797±0.014 |
| kimi | bert-base-uncased+vgg16 | 0.722±0.010 | 0.869±0.014 | 0.568±0.011 | 0.797±0.016 |
| mistral | roberta-base+vgg16 | 0.722±0.010 | 0.838±0.013 | 0.568±0.010 | 0.797±0.011 |
| qwen | roberta-base+resnet34 | 0.722±0.014 | 0.797±0.012 | 0.568±0.008 | 0.797±0.016 |
| qwen | roberta-base+mobilenet_v3_small | 0.722±0.014 | 0.840±0.015 | 0.568±0.010 | 0.797±0.014 |
| mistral | distilbert-base-uncased+mobilenet_v3_small | 0.722±0.013 | 0.789±0.014 | 0.568±0.010 | 0.797±0.011 |
| kimi | albert-base-v2+resnet18 | 0.743±0.011 | 0.852±0.016 | 0.602±0.010 | 0.812±0.012 |
| qwen | distilbert-base-uncased+resnet34 | 0.765±0.010 | 0.855±0.016 | 0.639±0.012 | 0.828±0.015 |
| mistral | distilbert-base-uncased+resnet18 | 0.765±0.014 | 0.859±0.012 | 0.639±0.012 | 0.828±0.011 |
| kimi | albert-base-v2+resnet34 | 0.812±0.013 | 0.881±0.014 | 0.719±0.014 | 0.859±0.015 |
| kimi | albert-base-v2+densenet121 | 0.615±0.009 | 0.742±0.010 | 0.383±0.006 | 0.703±0.011 |
| kimi | albert-base-v2+efficientnet_b0 | 0.615±0.010 | 0.746±0.013 | 0.383±0.005 | 0.703±0.012 |
| mistral | bert-base-uncased+mobilenet_v3_small | 0.615±0.008 | 0.729±0.012 | 0.383±0.008 | 0.703±0.009 |
| mistral | distilbert-base-uncased+densenet121 | 0.615±0.012 | 0.695±0.013 | 0.383±0.007 | 0.703±0.013 |
| kimi | distilbert-base-uncased+efficientnet_b0 | 0.615±0.009 | 0.693±0.011 | 0.383±0.005 | 0.703±0.014 |
| qwen | roberta-base+mobilenet_v3_small | 0.649±0.013 | 0.748±0.010 | 0.445±0.006 | 0.734±0.010 |
| mistral | roberta-base+mobilenet_v3_small | 0.649±0.009 | 0.750±0.010 | 0.445±0.007 | 0.734±0.010 |
| kimi | bert-base-uncased+resnet34 | 0.649±0.011 | 0.773±0.013 | 0.445±0.008 | 0.734±0.013 |
| mistral | bert-base-uncased+efficientnet_b0 | 0.649±0.011 | 0.758±0.014 | 0.445±0.007 | 0.734±0.010 |
| mistral | roberta-base+resnet50 | 0.649±0.010 | 0.775±0.011 | 0.445±0.009 | 0.734±0.013 |
| qwen | bert-base-uncased+densenet121 | 0.649±0.011 | 0.803±0.011 | 0.445±0.009 | 0.734±0.015 |
| mistral | albert-base-v2+densenet121 | 0.667±0.012 | 0.695±0.012 | 0.478±0.007 | 0.750±0.011 |
| mistral | bert-base-uncased+resnet50 | 0.667±0.013 | 0.754±0.014 | 0.478±0.006 | 0.750±0.014 |
| kimi | bert-base-uncased+efficientnet_b0 | 0.667±0.011 | 0.730±0.013 | 0.478±0.007 | 0.750±0.013 |
| kimi | bert-base-uncased+mobilenet_v3_small | 0.667±0.013 | 0.730±0.010 | 0.478±0.008 | 0.750±0.012 |
| mistral | roberta-base+efficientnet_b0 | 0.667±0.009 | 0.787±0.011 | 0.478±0.008 | 0.750±0.011 |
| mistral | distilbert-base-uncased+vgg16 | 0.667±0.010 | 0.777±0.015 | 0.478±0.009 | 0.750±0.015 |
| kimi | distilbert-base-uncased+mobilenet_v3_small | 0.667±0.009 | 0.762±0.012 | 0.478±0.007 | 0.750±0.012 |
| qwen | albert-base-v2+resnet50 | 0.667±0.013 | 0.805±0.011 | 0.478±0.009 | 0.750±0.012 |
| mistral | bert-base-uncased+resnet34 | 0.686±0.011 | 0.771±0.014 | 0.512±0.008 | 0.766±0.011 |
| mistral | distilbert-base-uncased+efficientnet_b0 | 0.686±0.012 | 0.828±0.011 | 0.512±0.007 | 0.766±0.014 |
| qwen | bert-base-uncased+mobilenet_v3_small | 0.686±0.010 | 0.793±0.012 | 0.512±0.007 | 0.766±0.010 |
| mistral | albert-base-v2+resnet18 | 0.686±0.011 | 0.818±0.015 | 0.512±0.009 | 0.766±0.011 |
| qwen | bert-base-uncased+efficientnet_b0 | 0.686±0.013 | 0.797±0.016 | 0.512±0.007 | 0.766±0.014 |
| kimi | bert-base-uncased+densenet121 | 0.686±0.010 | 0.766±0.015 | 0.512±0.008 | 0.766±0.014 |
| qwen | albert-base-v2+densenet121 | 0.686±0.010 | 0.807±0.013 | 0.512±0.008 | 0.766±0.012 |
| qwen | bert-base-uncased+resnet34 | 0.706±0.010 | 0.775±0.013 | 0.548±0.010 | 0.781±0.015 |
| mistral | distilbert-base-uncased+resnet34 | 0.706±0.013 | 0.812±0.011 | 0.548±0.011 | 0.781±0.012 |
| kimi | distilbert-base-uncased+resnet34 | 0.706±0.013 | 0.838±0.011 | 0.548±0.008 | 0.781±0.012 |
| kimi | roberta-base+resnet18 | 0.706±0.013 | 0.842±0.014 | 0.548±0.010 | 0.781±0.014 |
| qwen | roberta-base+shufflenet_v2_x1_0 | 0.706±0.009 | 0.805±0.011 | 0.548±0.007 | 0.781±0.012 |
| qwen | bert-base-uncased+resnet34 | 0.706±0.013 | 0.809±0.011 | 0.548±0.009 | 0.781±0.013 |
| mistral | bert-base-uncased+resnet50 | 0.706±0.010 | 0.795±0.013 | 0.548±0.011 | 0.781±0.015 |
| kimi | bert-base-uncased+resnet50 | 0.706±0.013 | 0.771±0.013 | 0.548±0.009 | 0.781±0.012 |
| mistral | albert-base-v2+resnet18 | 0.706±0.010 | 0.820±0.015 | 0.548±0.009 | 0.781±0.014 |
| qwen | bert-base-uncased+resnet50 | 0.706±0.014 | 0.803±0.016 | 0.548±0.009 | 0.781±0.011 |

*(end of table)*

*(continued)*

| Model | Init | F1 | AUC | MCC | Balanced Acc |
|-------|------|-----|-----|-----|--------------|
| kimi | albert-base-v2+resnet50 | 0.727±0.013 | 0.822±0.011 | 0.585±0.008 | 0.797±0.016 |
| kimi | bert-base-uncased+vgg16 | 0.727±0.010 | 0.822±0.014 | 0.585±0.009 | 0.797±0.013 |
| kimi | roberta-base+resnet50 | 0.727±0.011 | 0.793±0.014 | 0.585±0.008 | 0.797±0.015 |
| mistral | roberta-base+resnet18 | 0.727±0.010 | 0.822±0.014 | 0.585±0.012 | 0.797±0.012 |
| kimi | distilbert-base-uncased+resnet50 | 0.727±0.014 | 0.830±0.016 | 0.585±0.010 | 0.797±0.012 |
| mistral | albert-base-v2+resnet34 | 0.750±0.013 | 0.846±0.013 | 0.625±0.011 | 0.812±0.011 |
| kimi | albert-base-v2+vgg16 | 0.774±0.012 | 0.859±0.013 | 0.667±0.009 | 0.828±0.015 |
| mistral | distilbert-base-uncased+resnet34 | 0.800±0.012 | 0.805±0.012 | 0.713±0.010 | 0.844±0.016 |
| qwen | albert-base-v2+resnet34 | 0.800±0.011 | 0.869±0.017 | 0.713±0.013 | 0.844±0.013 |
| kimi | roberta-base+resnet50 | 0.595±0.009 | 0.645±0.013 | 0.356±0.007 | 0.688±0.011 |
| kimi | bert-base-uncased+resnet34 | 0.595±0.011 | 0.650±0.009 | 0.356±0.005 | 0.688±0.014 |
| mistral | bert-base-uncased+shufflenet_v2_x1_0 | 0.595±0.010 | 0.715±0.014 | 0.356±0.005 | 0.688±0.010 |
| qwen | albert-base-v2+densenet121 | 0.611±0.012 | 0.645±0.009 | 0.388±0.005 | 0.703±0.013 |
| qwen | albert-base-v2+shufflenet_v2_x1_0 | 0.611±0.010 | 0.746±0.014 | 0.388±0.006 | 0.703±0.013 |
| mistral | bert-base-uncased+resnet34 | 0.611±0.008 | 0.717±0.012 | 0.388±0.007 | 0.703±0.013 |
| mistral | roberta-base+resnet18 | 0.611±0.011 | 0.639±0.013 | 0.388±0.005 | 0.703±0.013 |
| mistral | distilbert-base-uncased+shufflenet_v2_x1_0 | 0.611±0.011 | 0.717±0.010 | 0.388±0.007 | 0.703±0.013 |
| qwen | roberta-base+densenet121 | 0.629±0.011 | 0.756±0.013 | 0.422±0.007 | 0.719±0.012 |
| mistral | bert-base-uncased+mobilenet_v3_small | 0.629±0.012 | 0.730±0.014 | 0.422±0.007 | 0.719±0.010 |
| qwen | bert-base-uncased+resnet50 | 0.647±0.011 | 0.771±0.013 | 0.456±0.007 | 0.734±0.014 |
| mistral | albert-base-v2+efficientnet_b0 | 0.647±0.011 | 0.777±0.013 | 0.456±0.007 | 0.734±0.010 |
| kimi | roberta-base+resnet18 | 0.667±0.012 | 0.848±0.016 | 0.493±0.010 | 0.750±0.010 |
| kimi | distilbert-base-uncased+densenet121 | 0.667±0.011 | 0.795±0.012 | 0.493±0.007 | 0.750±0.012 |
| mistral | bert-base-uncased+resnet18 | 0.667±0.010 | 0.795±0.011 | 0.493±0.008 | 0.750±0.013 |
| qwen | albert-base-v2+densenet121 | 0.667±0.011 | 0.787±0.014 | 0.493±0.008 | 0.750±0.013 |
| mistral | roberta-base+densenet121 | 0.667±0.009 | 0.840±0.014 | 0.493±0.009 | 0.750±0.010 |
| qwen | albert-base-v2+efficientnet_b0 | 0.667±0.009 | 0.791±0.013 | 0.493±0.007 | 0.750±0.014 |
| kimi | bert-base-uncased+densenet121 | 0.667±0.009 | 0.791±0.014 | 0.493±0.007 | 0.750±0.012 |
| kimi | albert-base-v2+resnet18 | 0.688±0.011 | 0.861±0.014 | 0.531±0.007 | 0.766±0.010 |
| mistral | bert-base-uncased+resnet34 | 0.688±0.010 | 0.820±0.012 | 0.531±0.010 | 0.766±0.011 |
| kimi | bert-base-uncased+shufflenet_v2_x1_0 | 0.688±0.013 | 0.826±0.014 | 0.531±0.007 | 0.766±0.013 |
| qwen | albert-base-v2+resnet18 | 0.688±0.010 | 0.791±0.015 | 0.531±0.008 | 0.766±0.013 |
| kimi | albert-base-v2+efficientnet_b0 | 0.688±0.013 | 0.770±0.015 | 0.531±0.009 | 0.766±0.011 |
| mistral | distilbert-base-uncased+vgg16 | 0.710±0.011 | 0.854±0.014 | 0.572±0.008 | 0.781±0.012 |
| mistral | albert-base-v2+shufflenet_v2_x1_0 | 0.710±0.010 | 0.740±0.011 | 0.572±0.010 | 0.781±0.015 |
| mistral | albert-base-v2+vgg16 | 0.733±0.010 | 0.875±0.014 | 0.616±0.009 | 0.797±0.013 |
| qwen | distilbert-base-uncased+resnet50 | 0.733±0.014 | 0.820±0.014 | 0.616±0.009 | 0.797±0.010 |
| qwen | distilbert-base-uncased+efficientnet_b0 | 0.733±0.014 | 0.834±0.011 | 0.616±0.011 | 0.797±0.011 |
| qwen | distilbert-base-uncased+resnet34 | 0.733±0.013 | 0.842±0.017 | 0.616±0.008 | 0.797±0.013 |
| qwen | albert-base-v2+resnet18 | 0.759±0.014 | 0.904±0.018 | 0.663±0.009 | 0.812±0.013 |
| qwen | bert-base-uncased+resnet18 | 0.759±0.014 | 0.883±0.016 | 0.663±0.012 | 0.812±0.014 |
| qwen | distilbert-base-uncased+vgg16 | 0.571±0.010 | 0.750±0.014 | 0.331±0.005 | 0.672±0.013 |
| mistral | roberta-base+resnet34 | 0.571±0.011 | 0.756±0.011 | 0.331±0.006 | 0.672±0.012 |
| kimi | bert-base-uncased+mobilenet_v3_small | 0.571±0.008 | 0.750±0.013 | 0.331±0.005 | 0.672±0.010 |
| kimi | distilbert-base-uncased+resnet18 | 0.588±0.011 | 0.670±0.009 | 0.365±0.005 | 0.688±0.009 |
| kimi | distilbert-base-uncased+shufflenet_v2_x1_0 | 0.588±0.011 | 0.727±0.013 | 0.365±0.005 | 0.688±0.013 |
| qwen | albert-base-v2+mobilenet_v3_small | 0.606±0.009 | 0.730±0.014 | 0.400±0.007 | 0.703±0.012 |
| kimi | albert-base-v2+resnet34 | 0.606±0.008 | 0.793±0.015 | 0.400±0.008 | 0.703±0.010 |
| qwen | roberta-base+resnet34 | 0.606±0.009 | 0.670±0.013 | 0.400±0.008 | 0.703±0.010 |
| kimi | distilbert-base-uncased+densenet121 | 0.606±0.010 | 0.748±0.014 | 0.400±0.006 | 0.703±0.014 |
| kimi | roberta-base+mobilenet_v3_small | 0.625±0.010 | 0.711±0.009 | 0.438±0.006 | 0.719±0.012 |
| kimi | roberta-base+resnet18 | 0.625±0.009 | 0.768±0.012 | 0.438±0.006 | 0.719±0.011 |

*(end of table)*

*(continued)*

| Model | Init | F1 | AUC | MCC | Balanced Acc |
|---|---|---|---|---|---|
| mistral | distilbert-base-uncased+mobilenet_v3_small | 0.625±0.010 | 0.777±0.013 | 0.438±0.007 | 0.719±0.010 |
| mistral | distilbert-base-uncased+efficientnet_b0 | 0.625±0.010 | 0.721±0.010 | 0.438±0.007 | 0.719±0.011 |
| qwen | distilbert-base-uncased+densenet121 | 0.625±0.012 | 0.758±0.015 | 0.438±0.006 | 0.719±0.010 |
| mistral | bert-base-uncased+resnet18 | 0.625±0.010 | 0.857±0.016 | 0.438±0.006 | 0.719±0.012 |
| qwen | roberta-base+efficientnet_b0 | 0.625±0.008 | 0.738±0.014 | 0.438±0.007 | 0.719±0.012 |
| mistral | roberta-base+resnet34 | 0.625±0.009 | 0.754±0.010 | 0.438±0.006 | 0.719±0.009 |
| mistral | roberta-base+shufflenet_v2_x1_0 | 0.625±0.010 | 0.682±0.013 | 0.438±0.008 | 0.719±0.013 |
| kimi | roberta-base+resnet34 | 0.625±0.012 | 0.795±0.013 | 0.438±0.006 | 0.719±0.012 |
| kimi | roberta-base+densenet121 | 0.625±0.012 | 0.742±0.013 | 0.438±0.008 | 0.719±0.010 |
| kimi | distilbert-base-uncased+resnet18 | 0.625±0.011 | 0.834±0.013 | 0.438±0.008 | 0.719±0.010 |
| mistral | bert-base-uncased+resnet18 | 0.645±0.011 | 0.832±0.013 | 0.477±0.008 | 0.734±0.014 |
| mistral | distilbert-base-uncased+resnet50 | 0.645±0.009 | 0.803±0.015 | 0.477±0.008 | 0.734±0.010 |
| qwen | roberta-base+resnet18 | 0.645±0.010 | 0.748±0.011 | 0.477±0.007 | 0.734±0.011 |
| kimi | distilbert-base-uncased+densenet121 | 0.645±0.010 | 0.779±0.010 | 0.477±0.008 | 0.734±0.010 |
| kimi | roberta-base+efficientnet_b0 | 0.645±0.010 | 0.760±0.012 | 0.477±0.007 | 0.734±0.013 |
| kimi | roberta-base+densenet121 | 0.645±0.010 | 0.789±0.016 | 0.477±0.009 | 0.734±0.012 |
| kimi | bert-base-uncased+efficientnet_b0 | 0.645±0.013 | 0.826±0.016 | 0.477±0.007 | 0.734±0.012 |
| kimi | bert-base-uncased+resnet50 | 0.645±0.013 | 0.787±0.015 | 0.477±0.007 | 0.734±0.010 |
| qwen | albert-base-v2+resnet34 | 0.667±0.012 | 0.812±0.015 | 0.519±0.009 | 0.750±0.013 |
| kimi | bert-base-uncased+shufflenet_v2_x1_0 | 0.667±0.011 | 0.777±0.016 | 0.519±0.008 | 0.750±0.013 |
| kimi | bert-base-uncased+resnet34 | 0.690±0.009 | 0.793±0.016 | 0.564±0.008 | 0.766±0.010 |
| mistral | roberta-base+resnet18 | 0.690±0.010 | 0.893±0.015 | 0.564±0.011 | 0.766±0.015 |
| kimi | distilbert-base-uncased+resnet34 | 0.690±0.013 | 0.824±0.015 | 0.564±0.010 | 0.766±0.010 |
| mistral | roberta-base+resnet50 | 0.714±0.010 | 0.838±0.016 | 0.612±0.009 | 0.781±0.013 |
| qwen | distilbert-base-uncased+resnet18 | 0.714±0.010 | 0.818±0.016 | 0.612±0.012 | 0.781±0.014 |
| qwen | distilbert-base-uncased+shufflenet_v2_x1_0 | 0.545±0.011 | 0.693±0.011 | 0.308±0.006 | 0.656±0.012 |
| kimi | bert-base-uncased+resnet50 | 0.545±0.009 | 0.719±0.011 | 0.308±0.005 | 0.656±0.013 |
| qwen | bert-base-uncased+densenet121 | 0.562±0.008 | 0.650±0.009 | 0.344±0.006 | 0.672±0.013 |
| qwen | albert-base-v2+resnet34 | 0.581±0.010 | 0.754±0.015 | 0.381±0.006 | 0.688±0.011 |
| kimi | roberta-base+densenet121 | 0.581±0.011 | 0.680±0.012 | 0.381±0.005 | 0.688±0.013 |
| qwen | distilbert-base-uncased+densenet121 | 0.581±0.009 | 0.754±0.011 | 0.381±0.006 | 0.688±0.009 |
| mistral | albert-base-v2+resnet18 | 0.600±0.011 | 0.840±0.015 | 0.421±0.007 | 0.703±0.014 |
| qwen | bert-base-uncased+densenet121 | 0.600±0.009 | 0.697±0.012 | 0.421±0.007 | 0.703±0.012 |
| mistral | bert-base-uncased+densenet121 | 0.600±0.010 | 0.705±0.011 | 0.421±0.006 | 0.703±0.013 |
| mistral | bert-base-uncased+densenet121 | 0.600±0.010 | 0.676±0.009 | 0.421±0.006 | 0.703±0.011 |
| mistral | bert-base-uncased+shufflenet_v2_x1_0 | 0.600±0.010 | 0.699±0.012 | 0.421±0.008 | 0.703±0.012 |
| kimi | bert-base-uncased+resnet18 | 0.621±0.009 | 0.854±0.014 | 0.464±0.007 | 0.719±0.010 |
| mistral | distilbert-base-uncased+resnet18 | 0.621±0.008 | 0.832±0.014 | 0.464±0.006 | 0.719±0.011 |
| qwen | roberta-base+resnet18 | 0.621±0.009 | 0.818±0.016 | 0.464±0.007 | 0.719±0.010 |
| mistral | albert-base-v2+resnet34 | 0.621±0.012 | 0.742±0.014 | 0.464±0.008 | 0.719±0.013 |
| qwen | albert-base-v2+shufflenet_v2_x1_0 | 0.621±0.008 | 0.709±0.012 | 0.464±0.009 | 0.719±0.012 |
| qwen | albert-base-v2+densenet121 | 0.643±0.011 | 0.719±0.012 | 0.510±0.010 | 0.734±0.010 |
| qwen | bert-base-uncased+shufflenet_v2_x1_0 | 0.643±0.011 | 0.818±0.016 | 0.510±0.007 | 0.734±0.014 |
| kimi | roberta-base+shufflenet_v2_x1_0 | 0.643±0.008 | 0.791±0.013 | 0.510±0.009 | 0.734±0.011 |
| mistral | distilbert-base-uncased+shufflenet_v2_x1_0 | 0.643±0.011 | 0.771±0.011 | 0.510±0.008 | 0.734±0.015 |
| kimi | distilbert-base-uncased+mobilenet_v3_small | 0.643±0.009 | 0.799±0.011 | 0.510±0.009 | 0.734±0.011 |
| qwen | bert-base-uncased+resnet18 | 0.667±0.012 | 0.875±0.016 | 0.561±0.010 | 0.750±0.011 |
| mistral | distilbert-base-uncased+resnet50 | 0.667±0.009 | 0.791±0.014 | 0.561±0.011 | 0.750±0.014 |
| qwen | bert-base-uncased+shufflenet_v2_x1_0 | 0.692±0.012 | 0.867±0.016 | 0.617±0.009 | 0.766±0.013 |
| mistral | albert-base-v2+densenet121 | 0.533±0.008 | 0.779±0.011 | 0.324±0.004 | 0.656±0.011 |
| mistral | distilbert-base-uncased+resnet34 | 0.571±0.009 | 0.727±0.011 | 0.408±0.008 | 0.688±0.010 |
| qwen | roberta-base+resnet18 | 0.571±0.008 | 0.711±0.009 | 0.408±0.008 | 0.688±0.012 |

*(end of table)*

*(continued)*

| Model | Init | F1 | AUC | MCC | Balanced Acc |
|-------|------|-----|-----|-----|--------------|
| mistral | albert-base-v2+resnet34 | 0.571±0.011 | 0.662±0.009 | 0.408±0.006 | 0.688±0.012 |
| kimi | albert-base-v2+shufflenet_v2_x1_0 | 0.593±0.008 | 0.781±0.013 | 0.456±0.008 | 0.703±0.014 |
| qwen | roberta-base+shufflenet_v2_x1_0 | 0.593±0.012 | 0.684±0.011 | 0.456±0.008 | 0.703±0.014 |
| kimi | roberta-base+shufflenet_v2_x1_0 | 0.593±0.011 | 0.740±0.012 | 0.456±0.008 | 0.703±0.012 |
| mistral | distilbert-base-uncased+densenet121 | 0.593±0.009 | 0.762±0.015 | 0.456±0.006 | 0.703±0.010 |
| qwen | distilbert-base-uncased+shufflenet_v2_x1_0 | 0.593±0.010 | 0.744±0.011 | 0.456±0.009 | 0.703±0.012 |
| kimi | distilbert-base-uncased+resnet18 | 0.593±0.011 | 0.826±0.014 | 0.456±0.007 | 0.703±0.009 |
| qwen | albert-base-v2+resnet34 | 0.615±0.010 | 0.797±0.015 | 0.508±0.008 | 0.719±0.009 |
| mistral | roberta-base+shufflenet_v2_x1_0 | 0.615±0.010 | 0.799±0.013 | 0.508±0.008 | 0.719±0.011 |
| kimi | bert-base-uncased+resnet18 | 0.615±0.009 | 0.865±0.015 | 0.508±0.007 | 0.719±0.012 |
| qwen | roberta-base+resnet18 | 0.615±0.008 | 0.836±0.016 | 0.508±0.007 | 0.719±0.014 |
| qwen | distilbert-base-uncased+resnet18 | 0.615±0.012 | 0.836±0.016 | 0.508±0.009 | 0.719±0.013 |
| qwen | distilbert-base-uncased+resnet18 | 0.483±0.009 | 0.715±0.010 | 0.265±0.004 | 0.625±0.011 |
| kimi | roberta-base+resnet18 | 0.500±0.009 | 0.715±0.014 | 0.306±0.005 | 0.641±0.010 |
| mistral | albert-base-v2+resnet50 | 0.519±0.008 | 0.584±0.009 | 0.350±0.006 | 0.656±0.010 |
| qwen | albert-base-v2+resnet18 | 0.519±0.009 | 0.697±0.011 | 0.350±0.005 | 0.656±0.013 |
| kimi | albert-base-v2+densenet121 | 0.519±0.008 | 0.695±0.011 | 0.350±0.005 | 0.656±0.010 |
| kimi | albert-base-v2+resnet34 | 0.538±0.010 | 0.820±0.016 | 0.399±0.006 | 0.672±0.009 |
| qwen | bert-base-uncased+resnet50 | 0.538±0.010 | 0.783±0.013 | 0.399±0.007 | 0.672±0.013 |
| mistral | bert-base-uncased+resnet50 | 0.538±0.010 | 0.709±0.011 | 0.399±0.006 | 0.672±0.011 |
| mistral | distilbert-base-uncased+densenet121 | 0.538±0.009 | 0.791±0.014 | 0.399±0.007 | 0.672±0.011 |
| qwen | distilbert-base-uncased+resnet50 | 0.538±0.007 | 0.658±0.010 | 0.399±0.007 | 0.672±0.012 |
| mistral | albert-base-v2+mobilenet_v3_small | 0.560±0.008 | 0.811±0.011 | 0.453±0.008 | 0.688±0.011 |
| qwen | distilbert-base-uncased+shufflenet_v2_x1_0 | 0.560±0.011 | 0.795±0.011 | 0.453±0.009 | 0.688±0.013 |
| mistral | roberta-base+densenet121 | 0.583±0.010 | 0.822±0.011 | 0.514±0.009 | 0.703±0.014 |
| qwen | roberta-base+resnet50 | 0.583±0.008 | 0.613±0.009 | 0.514±0.007 | 0.703±0.011 |
| qwen | roberta-base+shufflenet_v2_x1_0 | 0.462±0.008 | 0.705±0.012 | 0.290±0.004 | 0.625±0.011 |
| kimi | albert-base-v2+shufflenet_v2_x1_0 | 0.462±0.006 | 0.668±0.009 | 0.290±0.004 | 0.625±0.009 |
| qwen | distilbert-base-uncased+resnet34 | 0.480±0.006 | 0.688±0.010 | 0.340±0.006 | 0.641±0.012 |
| kimi | roberta-base+densenet121 | 0.500±0.007 | 0.768±0.012 | 0.395±0.005 | 0.656±0.009 |
| kimi | bert-base-uncased+resnet18 | 0.571±0.008 | 0.787±0.011 | 0.316±0.005 | 0.625±0.012 |
| mistral | roberta-base+mobilenet_v3_small | 0.000±0.000 | 0.000±0.000 | 0.000±0.000 | 0.000±0.000 |
| qwen | roberta-base+mobilenet_v3_small | 0.000±0.000 | 0.000±0.000 | 0.000±0.000 | 0.000±0.000 |
| mistral | roberta-base+mobilenet_v3_small | 0.000±0.000 | 0.000±0.000 | 0.000±0.000 | 0.000±0.000 |
| mistral | bert-base-uncased+resnet18 | 0.491±0.008 | 0.633±0.009 | 0.042±0.001 | 0.516±0.010 |
| kimi | roberta-base+mobilenet_v3_small | 0.000±0.000 | 0.000±0.000 | 0.000±0.000 | 0.000±0.000 |
| kimi | roberta-base+mobilenet_v3_small | 0.622±0.011 | 0.752±0.011 | 0.392±0.008 | 0.703±0.011 |
| kimi | roberta-base+mobilenet_v3_small | 0.000±0.000 | 0.000±0.000 | 0.000±0.000 | 0.000±0.000 |
| qwen | roberta-base+shufflenet_v2_x1_0 | 0.500±0.010 | 0.383±0.006 | 0.000±0.000 | 0.500±0.009 |
| mistral | roberta-base+resnet18 | 0.549±0.007 | 0.717±0.013 | 0.232±0.003 | 0.609±0.009 |
| qwen | bert-base-uncased+resnet18 | 0.385±0.006 | 0.674±0.012 | 0.181±0.003 | 0.578±0.011 |
| kimi | bert-base-uncased+resnet18 | 0.390±0.007 | 0.525±0.009 | -0.029±0.000 | 0.484±0.008 |
| qwen | roberta-base+resnet34 | 0.600±0.010 | 0.715±0.012 | 0.354±0.005 | 0.688±0.013 |
| qwen | roberta-base+mobilenet_v3_small | 0.000±0.000 | 0.000±0.000 | 0.000±0.000 | 0.000±0.000 |
| qwen | bert-base-uncased+resnet18 | 0.500±0.008 | 0.555±0.009 | 0.000±0.000 | 0.500±0.008 |
| qwen | bert-base-uncased+shufflenet_v2_x1_0 | 0.000±0.000 | 0.000±0.000 | 0.000±0.000 | 0.000±0.000 |
| mistral | bert-base-uncased+mobilenet_v3_small | 0.000±0.000 | 0.000±0.000 | 0.000±0.000 | 0.000±0.000 |
| qwen | distilbert-base-uncased+vgg16 | 0.500±0.007 | 0.574±0.011 | 0.000±0.000 | 0.500±0.008 |
| qwen | bert-base-uncased+mobilenet_v3_small | 0.000±0.000 | 0.000±0.000 | 0.000±0.000 | 0.000±0.000 |
| kimi | roberta-base+efficientnet_b0 | 0.000±0.000 | 0.000±0.000 | 0.000±0.000 | 0.000±0.000 |
| mistral | bert-base-uncased+mobilenet_v3_small | 0.000±0.000 | 0.000±0.000 | 0.000±0.000 | 0.000±0.000 |
| qwen | bert-base-uncased+mobilenet_v3_small | 0.000±0.000 | 0.000±0.000 | 0.000±0.000 | 0.000±0.000 |

*(end of table)*

*(continued)*

| Model | Init | F1 | AUC | MCC | Balanced Acc |
|---|---|---|---|---|---|
| qwen | bert-base-uncased+resnet34 | 0.500±0.009 | 0.553±0.010 | 0.250±0.004 | 0.625±0.012 |
| kimi | roberta-base+resnet34 | 0.564±0.008 | 0.664±0.010 | 0.295±0.005 | 0.656±0.012 |
| qwen | bert-base-uncased+resnet34 | 0.508±0.009 | 0.684±0.010 | 0.096±0.001 | 0.531±0.009 |
| kimi | roberta-base+resnet34 | 0.500±0.007 | 0.516±0.009 | 0.000±0.000 | 0.500±0.009 |
| qwen | roberta-base+efficientnet_b0 | 0.652±0.009 | 0.711±0.014 | 0.456±0.007 | 0.734±0.014 |
| qwen | bert-base-uncased+shufflenet_v2_x1_0 | 0.000±0.000 | 0.000±0.000 | 0.000±0.000 | 0.000±0.000 |
| kimi | roberta-base+shufflenet_v2_x1_0 | 0.500±0.008 | 0.500±0.009 | 0.000±0.000 | 0.500±0.007 |
| mistral | roberta-base+shufflenet_v2_x1_0 | 0.414±0.007 | 0.613±0.009 | 0.166±0.002 | 0.578±0.011 |
| kimi | roberta-base+shufflenet_v2_x1_0 | 0.500±0.009 | 0.426±0.008 | 0.000±0.000 | 0.500±0.009 |
| mistral | bert-base-uncased+shufflenet_v2_x1_0 | 0.000±0.000 | 0.000±0.000 | 0.000±0.000 | 0.000±0.000 |
| kimi | distilbert-base-uncased+vgg16 | 0.000±0.000 | 0.000±0.000 | 0.000±0.000 | 0.000±0.000 |
| mistral | roberta-base+efficientnet_b0 | 0.000±0.000 | 0.000±0.000 | 0.000±0.000 | 0.000±0.000 |
| mistral | bert-base-uncased+resnet34 | 0.500±0.009 | 0.588±0.010 | 0.000±0.000 | 0.500±0.008 |
| kimi | bert-base-uncased+mobilenet_v3_small | 0.000±0.000 | 0.000±0.000 | 0.000±0.000 | 0.000±0.000 |
| qwen | roberta-base+efficientnet_b0 | 0.000±0.000 | 0.000±0.000 | 0.000±0.000 | 0.000±0.000 |
| mistral | bert-base-uncased+shufflenet_v2_x1_0 | 0.500±0.007 | 0.516±0.009 | 0.000±0.000 | 0.500±0.009 |
| qwen | roberta-base+efficientnet_b0 | 0.000±0.000 | 0.000±0.000 | 0.000±0.000 | 0.000±0.000 |
| qwen | distilbert-base-uncased+vgg16 | 0.000±0.000 | 0.000±0.000 | 0.000±0.000 | 0.000±0.000 |
| qwen | distilbert-base-uncased+densenet121 | 0.571±0.009 | 0.689±0.013 | 0.296±0.004 | 0.656±0.012 |
| kimi | distilbert-base-uncased+vgg16 | 0.000±0.000 | 0.000±0.000 | 0.000±0.000 | 0.000±0.000 |
| kimi | distilbert-base-uncased+shufflenet_v2_x1_0 | 0.000±0.000 | 0.000±0.000 | 0.000±0.000 | 0.000±0.000 |
| kimi | distilbert-base-uncased+mobilenet_v3_small | 0.000±0.000 | 0.000±0.000 | 0.000±0.000 | 0.000±0.000 |
| mistral | distilbert-base-uncased+shufflenet_v2_x1_0 | 0.000±0.000 | 0.000±0.000 | 0.000±0.000 | 0.000±0.000 |
| mistral | distilbert-base-uncased+shufflenet_v2_x1_0 | 0.000±0.000 | 0.000±0.000 | 0.000±0.000 | 0.000±0.000 |
| kimi | distilbert-base-uncased+resnet34 | 0.564±0.009 | 0.703±0.011 | 0.295±0.005 | 0.656±0.009 |
| kimi | distilbert-base-uncased+mobilenet_v3_small | 0.000±0.000 | 0.000±0.000 | 0.000±0.000 | 0.000±0.000 |
| qwen | distilbert-base-uncased+shufflenet_v2_x1_0 | 0.486±0.010 | 0.654±0.013 | 0.178±0.002 | 0.594±0.010 |
| qwen | distilbert-base-uncased+resnet34 | 0.456±0.007 | 0.531±0.009 | -0.083±0.001 | 0.469±0.009 |
| mistral | distilbert-base-uncased+resnet34 | 0.582±0.011 | 0.695±0.010 | 0.340±0.006 | 0.641±0.009 |
| kimi | distilbert-base-uncased+resnet34 | 0.605±0.009 | 0.721±0.013 | 0.356±0.007 | 0.688±0.009 |
| mistral | distilbert-base-uncased+vgg16 | 0.000±0.000 | 0.000±0.000 | 0.000±0.000 | 0.000±0.000 |
| qwen | distilbert-base-uncased+mobilenet_v3_small | 0.000±0.000 | 0.000±0.000 | 0.000±0.000 | 0.000±0.000 |
| mistral | distilbert-base-uncased+mobilenet_v3_small | 0.500±0.007 | 0.607±0.008 | 0.000±0.000 | 0.500±0.008 |
| mistral | distilbert-base-uncased+mobilenet_v3_small | 0.000±0.000 | 0.000±0.000 | 0.000±0.000 | 0.000±0.000 |
| qwen | distilbert-base-uncased+mobilenet_v3_small | 0.000±0.000 | 0.000±0.000 | 0.000±0.000 | 0.000±0.000 |
| kimi | distilbert-base-uncased+resnet18 | 0.571±0.010 | 0.729±0.012 | 0.316±0.005 | 0.625±0.009 |
| mistral | distilbert-base-uncased+resnet18 | 0.429±0.006 | 0.680±0.013 | 0.204±0.004 | 0.594±0.009 |
| qwen | distilbert-base-uncased+resnet18 | 0.520±0.008 | 0.643±0.012 | 0.162±0.003 | 0.578±0.009 |
| mistral | distilbert-base-uncased+resnet18 | 0.636±0.012 | 0.721±0.010 | 0.418±0.006 | 0.719±0.013 |
| kimi | distilbert-base-uncased+efficientnet_b0 | 0.636±0.012 | 0.689±0.013 | 0.418±0.006 | 0.719±0.013 |
| kimi | distilbert-base-uncased+shufflenet_v2_x1_0 | 0.500±0.009 | 0.574±0.009 | 0.000±0.000 | 0.500±0.009 |
| qwen | distilbert-base-uncased+efficientnet_b0 | 0.000±0.000 | 0.000±0.000 | 0.000±0.000 | 0.000±0.000 |
| kimi | distilbert-base-uncased+shufflenet_v2_x1_0 | 0.564±0.010 | 0.605±0.009 | 0.295±0.006 | 0.656±0.011 |
| kimi | distilbert-base-uncased+vgg16 | 0.000±0.000 | 0.000±0.000 | 0.000±0.000 | 0.000±0.000 |
| kimi | distilbert-base-uncased+vgg16 | 0.000±0.000 | 0.000±0.000 | 0.000±0.000 | 0.000±0.000 |
| mistral | distilbert-base-uncased+vgg16 | 0.000±0.000 | 0.000±0.000 | 0.000±0.000 | 0.000±0.000 |
| mistral | distilbert-base-uncased+densenet121 | 0.564±0.009 | 0.676±0.013 | 0.295±0.004 | 0.656±0.009 |
| kimi | distilbert-base-uncased+densenet121 | 0.545±0.008 | 0.588±0.008 | 0.226±0.004 | 0.594±0.010 |
| kimi | roberta-base+efficientnet_b0 | 0.000±0.000 | 0.000±0.000 | 0.000±0.000 | 0.000±0.000 |
| qwen | distilbert-base-uncased+densenet121 | 0.588±0.009 | 0.732±0.011 | 0.331±0.005 | 0.656±0.011 |
| qwen | distilbert-base-uncased+resnet50 | 0.500±0.009 | 0.611±0.009 | 0.000±0.000 | 0.500±0.008 |
| kimi | distilbert-base-uncased+resnet50 | 0.545±0.010 | 0.611±0.009 | 0.239±0.004 | 0.625±0.010 |

*(end of table)*

*(continued)*

| Model | Init | F1 | AUC | MCC | Balanced Acc |
|---|---|---|---|---|---|
| mistral | distilbert-base-uncased+resnet50 | 0.500±0.007 | 0.549±0.010 | 0.149±0.003 | 0.578±0.011 |
| kimi | distilbert-base-uncased+resnet50 | 0.517±0.008 | 0.631±0.011 | 0.134±0.002 | 0.547±0.009 |
| mistral | distilbert-base-uncased+efficientnet_b0 | 0.000±0.000 | 0.000±0.000 | 0.000±0.000 | 0.000±0.000 |
| kimi | distilbert-base-uncased+efficientnet_b0 | 0.000±0.000 | 0.000±0.000 | 0.000±0.000 | 0.000±0.000 |
| kimi | distilbert-base-uncased+efficientnet_b0 | 0.000±0.000 | 0.000±0.000 | 0.000±0.000 | 0.000±0.000 |
| qwen | distilbert-base-uncased+efficientnet_b0 | 0.000±0.000 | 0.000±0.000 | 0.000±0.000 | 0.000±0.000 |
| qwen | distilbert-base-uncased+efficientnet_b0 | 0.566±0.008 | 0.742±0.014 | 0.280±0.006 | 0.625±0.010 |
| mistral | distilbert-base-uncased+efficientnet_b0 | 0.000±0.000 | 0.000±0.000 | 0.000±0.000 | 0.000±0.000 |
| mistral | roberta-base+efficientnet_b0 | 0.000±0.000 | 0.000±0.000 | 0.000±0.000 | 0.000±0.000 |
| kimi | bert-base-uncased+shufflenet_v2_x1_0 | 0.500±0.008 | 0.629±0.009 | 0.177±0.003 | 0.594±0.012 |
| kimi | bert-base-uncased+resnet34 | 0.564±0.009 | 0.680±0.011 | 0.295±0.006 | 0.656±0.009 |
| mistral | albert-base-v2+shufflenet_v2_x1_0 | 0.438±0.006 | 0.551±0.009 | 0.156±0.003 | 0.578±0.008 |
| mistral | albert-base-v2+efficientnet_b0 | 0.000±0.000 | 0.000±0.000 | 0.000±0.000 | 0.000±0.000 |
| kimi | albert-base-v2+efficientnet_b0 | 0.000±0.000 | 0.000±0.000 | 0.000±0.000 | 0.000±0.000 |
| qwen | albert-base-v2+efficientnet_b0 | 0.000±0.000 | 0.000±0.000 | 0.000±0.000 | 0.000±0.000 |
| mistral | albert-base-v2+efficientnet_b0 | 0.000±0.000 | 0.000±0.000 | 0.000±0.000 | 0.000±0.000 |
| kimi | albert-base-v2+mobilenet_v3_small | 0.000±0.000 | 0.000±0.000 | 0.000±0.000 | 0.000±0.000 |
| kimi | albert-base-v2+shufflenet_v2_x1_0 | 0.000±0.000 | 0.000±0.000 | 0.000±0.000 | 0.000±0.000 |
| qwen | albert-base-v2+shufflenet_v2_x1_0 | 0.526±0.007 | 0.578±0.008 | 0.237±0.004 | 0.625±0.010 |
| kimi | albert-base-v2+shufflenet_v2_x1_0 | 0.500±0.009 | 0.357±0.006 | 0.000±0.000 | 0.500±0.007 |
| mistral | albert-base-v2+shufflenet_v2_x1_0 | 0.556±0.011 | 0.639±0.011 | 0.299±0.004 | 0.656±0.009 |
| kimi | albert-base-v2+resnet18 | 0.524±0.008 | 0.598±0.008 | 0.207±0.004 | 0.609±0.008 |
| qwen | albert-base-v2+shufflenet_v2_x1_0 | 0.500±0.010 | 0.570±0.009 | 0.000±0.000 | 0.500±0.008 |
| mistral | albert-base-v2+shufflenet_v2_x1_0 | 0.513±0.010 | 0.646±0.010 | 0.206±0.003 | 0.609±0.009 |
| kimi | albert-base-v2+mobilenet_v3_small | 0.000±0.000 | 0.000±0.000 | 0.000±0.000 | 0.000±0.000 |
| qwen | albert-base-v2+mobilenet_v3_small | 0.000±0.000 | 0.000±0.000 | 0.000±0.000 | 0.000±0.000 |
| mistral | albert-base-v2+mobilenet_v3_small | 0.000±0.000 | 0.000±0.000 | 0.000±0.000 | 0.000±0.000 |
| kimi | albert-base-v2+resnet34 | 0.596±0.010 | 0.650±0.012 | 0.339±0.006 | 0.672±0.012 |
| qwen | bert-base-uncased+efficientnet_b0 | 0.000±0.000 | 0.000±0.000 | 0.000±0.000 | 0.000±0.000 |
| mistral | albert-base-v2+mobilenet_v3_small | 0.000±0.000 | 0.000±0.000 | 0.000±0.000 | 0.000±0.000 |
| qwen | albert-base-v2+efficientnet_b0 | 0.000±0.000 | 0.000±0.000 | 0.000±0.000 | 0.000±0.000 |
| qwen | albert-base-v2+efficientnet_b0 | 0.652±0.011 | 0.721±0.010 | 0.456±0.006 | 0.734±0.012 |
| kimi | albert-base-v2+efficientnet_b0 | 0.000±0.000 | 0.000±0.000 | 0.000±0.000 | 0.000±0.000 |
| mistral | albert-base-v2+efficientnet_b0 | 0.609±0.010 | 0.674±0.013 | 0.365±0.005 | 0.688±0.012 |
| kimi | albert-base-v2+vgg16 | 0.000±0.000 | 0.000±0.000 | 0.000±0.000 | 0.000±0.000 |
| kimi | albert-base-v2+vgg16 | 0.000±0.000 | 0.000±0.000 | 0.000±0.000 | 0.000±0.000 |
| qwen | albert-base-v2+vgg16 | 0.000±0.000 | 0.000±0.000 | 0.000±0.000 | 0.000±0.000 |
| kimi | albert-base-v2+vgg16 | 0.000±0.000 | 0.000±0.000 | 0.000±0.000 | 0.000±0.000 |
| qwen | albert-base-v2+vgg16 | 0.000±0.000 | 0.000±0.000 | 0.000±0.000 | 0.000±0.000 |
| mistral | albert-base-v2+vgg16 | 0.000±0.000 | 0.000±0.000 | 0.000±0.000 | 0.000±0.000 |
| mistral | albert-base-v2+vgg16 | 0.000±0.000 | 0.000±0.000 | 0.000±0.000 | 0.000±0.000 |
| mistral | albert-base-v2+vgg16 | 0.000±0.000 | 0.000±0.000 | 0.000±0.000 | 0.000±0.000 |
| qwen | albert-base-v2+vgg16 | 0.000±0.000 | 0.000±0.000 | 0.000±0.000 | 0.000±0.000 |
| qwen | albert-base-v2+vgg16 | 0.000±0.000 | 0.000±0.000 | 0.000±0.000 | 0.000±0.000 |
| kimi | albert-base-v2+densenet121 | 0.600±0.011 | 0.777±0.011 | 0.357±0.005 | 0.672±0.013 |
| mistral | albert-base-v2+densenet121 | 0.549±0.009 | 0.709±0.012 | 0.232±0.004 | 0.609±0.010 |
| qwen | albert-base-v2+resnet50 | 0.491±0.007 | 0.619±0.009 | 0.042±0.001 | 0.516±0.008 |
| kimi | albert-base-v2+resnet50 | 0.500±0.008 | 0.549±0.009 | 0.000±0.000 | 0.500±0.010 |
| mistral | albert-base-v2+resnet50 | 0.625±0.009 | 0.729±0.012 | 0.406±0.006 | 0.703±0.011 |
| kimi | albert-base-v2+resnet50 | 0.524±0.008 | 0.672±0.012 | 0.207±0.004 | 0.609±0.010 |
| qwen | albert-base-v2+resnet50 | 0.537±0.011 | 0.684±0.009 | 0.236±0.004 | 0.625±0.010 |
| qwen | albert-base-v2+resnet18 | 0.612±0.012 | 0.693±0.012 | 0.381±0.006 | 0.688±0.010 |

*(end of table)*

*(continued)*

| Model | Init | F1 | AUC | MCC | Balanced Acc |
|---|---|---|---|---|---|
| qwen | albert-base-v2+mobilenet_v3_small | 0.000±0.000 | 0.000±0.000 | 0.000±0.000 | 0.000±0.000 |
| mistral | bert-base-uncased+vgg16 | 0.500±0.008 | 0.562±0.009 | 0.000±0.000 | 0.500±0.009 |
| mistral | bert-base-uncased+densenet121 | 0.533±0.010 | 0.627±0.009 | 0.213±0.004 | 0.562±0.011 |
| kimi | bert-base-uncased+vgg16 | 0.000±0.000 | 0.000±0.000 | 0.000±0.000 | 0.000±0.000 |
| qwen | roberta-base+densenet121 | 0.571±0.009 | 0.697±0.012 | 0.286±0.005 | 0.641±0.013 |
| kimi | bert-base-uncased+resnet50 | 0.500±0.009 | 0.686±0.011 | 0.000±0.000 | 0.500±0.009 |
| kimi | roberta-base+resnet50 | 0.600±0.011 | 0.779±0.016 | 0.354±0.006 | 0.688±0.013 |
| qwen | bert-base-uncased+resnet50 | 0.571±0.010 | 0.688±0.012 | 0.296±0.004 | 0.656±0.011 |
| kimi | bert-base-uncased+shufflenet_v2_x1_0 | 0.000±0.000 | 0.000±0.000 | 0.000±0.000 | 0.000±0.000 |
| kimi | roberta-base+resnet50 | 0.500±0.010 | 0.479±0.007 | 0.053±0.001 | 0.516±0.008 |
| mistral | bert-base-uncased+resnet50 | 0.500±0.008 | 0.445±0.006 | 0.000±0.000 | 0.500±0.009 |
| kimi | bert-base-uncased+efficientnet_b0 | 0.000±0.000 | 0.000±0.000 | 0.000±0.000 | 0.000±0.000 |
| mistral | roberta-base+resnet50 | 0.500±0.007 | 0.516±0.007 | 0.102±0.002 | 0.547±0.007 |
| mistral | roberta-base+resnet50 | 0.579±0.010 | 0.725±0.010 | 0.325±0.006 | 0.672±0.010 |
| kimi | bert-base-uncased+mobilenet_v3_small | 0.000±0.000 | 0.000±0.000 | 0.000±0.000 | 0.000±0.000 |
| qwen | roberta-base+resnet50 | 0.578±0.009 | 0.750±0.011 | 0.301±0.006 | 0.656±0.009 |
| qwen | roberta-base+resnet50 | 0.500±0.008 | 0.561±0.011 | 0.000±0.000 | 0.500±0.009 |
| mistral | bert-base-uncased+efficientnet_b0 | 0.000±0.000 | 0.000±0.000 | 0.000±0.000 | 0.000±0.000 |
| mistral | bert-base-uncased+efficientnet_b0 | 0.000±0.000 | 0.000±0.000 | 0.000±0.000 | 0.000±0.000 |
| qwen | bert-base-uncased+efficientnet_b0 | 0.000±0.000 | 0.000±0.000 | 0.000±0.000 | 0.000±0.000 |
| qwen | roberta-base+densenet121 | 0.542±0.011 | 0.709±0.011 | 0.219±0.004 | 0.609±0.012 |
| mistral | roberta-base+densenet121 | 0.564±0.009 | 0.701±0.012 | 0.295±0.004 | 0.656±0.009 |
| mistral | bert-base-uncased+densenet121 | 0.467±0.009 | 0.762±0.011 | 0.227±0.003 | 0.609±0.009 |
| mistral | bert-base-uncased+vgg16 | 0.000±0.000 | 0.000±0.000 | 0.000±0.000 | 0.000±0.000 |
| kimi | roberta-base+vgg16 | 0.000±0.000 | 0.000±0.000 | 0.000±0.000 | 0.000±0.000 |
| qwen | bert-base-uncased+vgg16 | 0.000±0.000 | 0.000±0.000 | 0.000±0.000 | 0.000±0.000 |
| qwen | bert-base-uncased+vgg16 | 0.000±0.000 | 0.000±0.000 | 0.000±0.000 | 0.000±0.000 |
| mistral | bert-base-uncased+vgg16 | 0.000±0.000 | 0.000±0.000 | 0.000±0.000 | 0.000±0.000 |
| qwen | bert-base-uncased+vgg16 | 0.000±0.000 | 0.000±0.000 | 0.000±0.000 | 0.000±0.000 |
| kimi | bert-base-uncased+densenet121 | 0.549±0.010 | 0.666±0.011 | 0.232±0.003 | 0.609±0.008 |
| kimi | bert-base-uncased+vgg16 | 0.000±0.000 | 0.000±0.000 | 0.000±0.000 | 0.000±0.000 |
| kimi | roberta-base+vgg16 | 0.000±0.000 | 0.000±0.000 | 0.000±0.000 | 0.000±0.000 |
| kimi | roberta-base+vgg16 | 0.000±0.000 | 0.000±0.000 | 0.000±0.000 | 0.000±0.000 |
| kimi | roberta-base+vgg16 | 0.000±0.000 | 0.000±0.000 | 0.000±0.000 | 0.000±0.000 |
| qwen | roberta-base+vgg16 | 0.000±0.000 | 0.000±0.000 | 0.000±0.000 | 0.000±0.000 |
| mistral | roberta-base+vgg16 | 0.000±0.000 | 0.000±0.000 | 0.000±0.000 | 0.000±0.000 |
| qwen | roberta-base+vgg16 | 0.000±0.000 | 0.000±0.000 | 0.000±0.000 | 0.000±0.000 |
| mistral | roberta-base+vgg16 | 0.000±0.000 | 0.000±0.000 | 0.000±0.000 | 0.000±0.000 |
| qwen | roberta-base+vgg16 | 0.000±0.000 | 0.000±0.000 | 0.000±0.000 | 0.000±0.000 |
| qwen | roberta-base+vgg16 | 0.541±0.008 | 0.605±0.010 | 0.267±0.004 | 0.641±0.010 |
| mistral | roberta-base+vgg16 | 0.000±0.000 | 0.000±0.000 | 0.000±0.000 | 0.000±0.000 |
| mistral | roberta-base+shufflenet_v2_x1_0 | 0.417±0.005 | 0.619±0.009 | 0.277±0.005 | 0.609±0.010 |
| kimi | albert-base-v2+resnet18 | 0.435±0.008 | 0.789±0.012 | 0.334±0.006 | 0.625±0.009 |
| qwen | roberta-base+densenet121 | 0.435±0.007 | 0.779±0.014 | 0.334±0.006 | 0.625±0.011 |
| mistral | albert-base-v2+mobilenet_v3_small | 0.300±0.004 | 0.678±0.009 | 0.267±0.005 | 0.578±0.010 |
| mistral | albert-base-v2+resnet18 | 0.222±0.004 | 0.707±0.012 | 0.295±0.006 | 0.562±0.008 |
| qwen | bert-base-uncased+mobilenet_v3_small | 0.118±0.002 | 0.773±0.014 | 0.206±0.004 | 0.531±0.008 |
| qwen | distilbert-base-uncased+mobilenet_v3_small | 0.118±0.002 | 0.762±0.015 | 0.206±0.004 | 0.531±0.007 |

*(end of table)*

**Table S5.** Multimodal results using Cross Attention

| Model | Init | F1 | AUC | MCC | Balanced Acc |
|---|---|---|---|---|---|
| qwen | albert-base-v2+mobilenet_v3_small | 0.667±0.009 | 0.785±0.011 | 0.482±0.008 | 0.750±0.011 |
| kimi | albert-base-v2+resnet18 | 0.732±0.010 | 0.797±0.012 | 0.590±0.009 | 0.812±0.013 |
| qwen | roberta-base+resnet34 | 0.750±0.011 | 0.846±0.014 | 0.619±0.012 | 0.828±0.013 |
| kimi | bert-base-uncased+resnet18 | 0.750±0.015 | 0.848±0.014 | 0.619±0.009 | 0.828±0.014 |
| mistral | albert-base-v2+resnet50 | 0.651±0.012 | 0.752±0.011 | 0.445±0.006 | 0.734±0.010 |
| qwen | bert-base-uncased+densenet121 | 0.651±0.011 | 0.758±0.014 | 0.445±0.006 | 0.734±0.010 |
| mistral | albert-base-v2+mobilenet_v3_small | 0.651±0.010 | 0.797±0.011 | 0.445±0.007 | 0.734±0.013 |
| kimi | albert-base-v2+mobilenet_v3_small | 0.651±0.009 | 0.783±0.013 | 0.445±0.008 | 0.734±0.014 |
| kimi | distilbert-base-uncased+vgg16 | 0.667±0.012 | 0.699±0.014 | 0.473±0.009 | 0.750±0.014 |
| mistral | distilbert-base-uncased+densenet121 | 0.667±0.013 | 0.725±0.013 | 0.473±0.008 | 0.750±0.012 |
| kimi | distilbert-base-uncased+resnet34 | 0.683±0.012 | 0.795±0.013 | 0.501±0.008 | 0.766±0.011 |
| qwen | albert-base-v2+resnet50 | 0.683±0.012 | 0.760±0.012 | 0.501±0.009 | 0.766±0.012 |
| mistral | roberta-base+mobilenet_v3_small | 0.683±0.014 | 0.793±0.012 | 0.501±0.007 | 0.766±0.010 |
| qwen | bert-base-uncased+resnet50 | 0.700±0.013 | 0.822±0.015 | 0.530±0.010 | 0.781±0.012 |
| qwen | albert-base-v2+resnet34 | 0.700±0.013 | 0.795±0.014 | 0.530±0.009 | 0.781±0.014 |
| qwen | distilbert-base-uncased+resnet34 | 0.718±0.012 | 0.793±0.012 | 0.560±0.011 | 0.797±0.013 |
| qwen | roberta-base+densenet121 | 0.718±0.009 | 0.783±0.014 | 0.560±0.008 | 0.797±0.011 |
| qwen | distilbert-base-uncased+resnet18 | 0.718±0.014 | 0.785±0.010 | 0.560±0.009 | 0.797±0.014 |
| qwen | roberta-base+efficientnet_b0 | 0.718±0.010 | 0.793±0.013 | 0.560±0.008 | 0.797±0.014 |
| kimi | roberta-base+resnet18 | 0.737±0.012 | 0.789±0.015 | 0.591±0.009 | 0.812±0.011 |
| qwen | albert-base-v2+resnet34 | 0.737±0.012 | 0.826±0.015 | 0.591±0.011 | 0.812±0.012 |
| kimi | albert-base-v2+vgg16 | 0.757±0.012 | 0.838±0.015 | 0.624±0.012 | 0.828±0.014 |
| qwen | distilbert-base-uncased+vgg16 | 0.757±0.010 | 0.840±0.017 | 0.624±0.011 | 0.828±0.016 |
| mistral | albert-base-v2+densenet121 | 0.757±0.014 | 0.830±0.016 | 0.624±0.010 | 0.828±0.012 |
| mistral | distilbert-base-uncased+mobilenet_v3_small | 0.757±0.012 | 0.811±0.015 | 0.624±0.008 | 0.828±0.013 |
| kimi | roberta-base+resnet34 | 0.778±0.014 | 0.852±0.013 | 0.657±0.011 | 0.844±0.016 |
| kimi | bert-base-uncased+vgg16 | 0.800±0.014 | 0.822±0.014 | 0.693±0.009 | 0.859±0.015 |
| kimi | roberta-base+resnet50 | 0.634±0.011 | 0.740±0.012 | 0.413±0.006 | 0.719±0.014 |
| kimi | distilbert-base-uncased+resnet18 | 0.634±0.011 | 0.771±0.015 | 0.413±0.007 | 0.719±0.011 |
| qwen | distilbert-base-uncased+densenet121 | 0.634±0.011 | 0.721±0.011 | 0.413±0.008 | 0.719±0.010 |
| kimi | albert-base-v2+shufflenet_v2_x1_0 | 0.650±0.013 | 0.764±0.014 | 0.442±0.007 | 0.734±0.012 |
| mistral | roberta-base+resnet18 | 0.650±0.009 | 0.818±0.012 | 0.442±0.006 | 0.734±0.010 |
| qwen | albert-base-v2+mobilenet_v3_small | 0.650±0.010 | 0.811±0.012 | 0.442±0.009 | 0.734±0.014 |
| kimi | albert-base-v2+efficientnet_b0 | 0.650±0.009 | 0.766±0.015 | 0.442±0.007 | 0.734±0.014 |
| kimi | roberta-base+vgg16 | 0.667±0.011 | 0.752±0.014 | 0.472±0.009 | 0.750±0.010 |
| mistral | bert-base-uncased+densenet121 | 0.667±0.010 | 0.805±0.011 | 0.472±0.008 | 0.750±0.013 |
| kimi | albert-base-v2+mobilenet_v3_small | 0.667±0.009 | 0.762±0.015 | 0.472±0.009 | 0.750±0.015 |
| kimi | distilbert-base-uncased+mobilenet_v3_small | 0.667±0.012 | 0.789±0.013 | 0.472±0.006 | 0.750±0.014 |
| qwen | bert-base-uncased+shufflenet_v2_x1_0 | 0.667±0.011 | 0.742±0.015 | 0.472±0.009 | 0.750±0.013 |
| mistral | roberta-base+resnet18 | 0.667±0.010 | 0.818±0.013 | 0.472±0.006 | 0.750±0.014 |
| kimi | albert-base-v2+resnet34 | 0.684±0.012 | 0.781±0.015 | 0.503±0.007 | 0.766±0.011 |
| qwen | roberta-base+shufflenet_v2_x1_0 | 0.684±0.011 | 0.799±0.014 | 0.503±0.007 | 0.766±0.011 |
| kimi | bert-base-uncased+densenet121 | 0.684±0.011 | 0.789±0.016 | 0.503±0.009 | 0.766±0.011 |
| mistral | roberta-base+resnet34 | 0.684±0.011 | 0.799±0.011 | 0.503±0.007 | 0.766±0.014 |
| mistral | distilbert-base-uncased+resnet50 | 0.684±0.010 | 0.834±0.016 | 0.503±0.009 | 0.766±0.012 |
| kimi | distilbert-base-uncased+densenet121 | 0.684±0.013 | 0.779±0.011 | 0.503±0.009 | 0.766±0.012 |
| mistral | bert-base-uncased+shufflenet_v2_x1_0 | 0.684±0.011 | 0.791±0.012 | 0.503±0.007 | 0.766±0.015 |
| kimi | distilbert-base-uncased+resnet50 | 0.703±0.011 | 0.809±0.011 | 0.535±0.008 | 0.781±0.015 |
| kimi | albert-base-v2+resnet50 | 0.703±0.011 | 0.816±0.012 | 0.535±0.010 | 0.781±0.015 |
| kimi | distilbert-base-uncased+densenet121 | 0.703±0.013 | 0.791±0.016 | 0.535±0.009 | 0.781±0.012 |
| kimi | roberta-base+resnet18 | 0.703±0.011 | 0.826±0.015 | 0.535±0.009 | 0.781±0.012 |

*(end of table)*

*(continued)*

| Model | Init | F1 | AUC | MCC | Balanced Acc |
|---|---|---|---|---|---|
| qwen | distilbert-base-uncased+densenet121 | 0.722±0.013 | 0.801±0.013 | 0.568±0.011 | 0.797±0.016 |
| qwen | albert-base-v2+efficientnet_b0 | 0.722±0.013 | 0.783±0.013 | 0.568±0.009 | 0.797±0.012 |
| qwen | roberta-base+resnet50 | 0.722±0.013 | 0.842±0.014 | 0.568±0.008 | 0.797±0.012 |
| mistral | roberta-base+mobilenet_v3_small | 0.743±0.012 | 0.822±0.012 | 0.602±0.010 | 0.812±0.016 |
| qwen | roberta-base+vgg16 | 0.743±0.014 | 0.777±0.015 | 0.602±0.011 | 0.812±0.015 |
| mistral | bert-base-uncased+mobilenet_v3_small | 0.743±0.011 | 0.785±0.010 | 0.602±0.009 | 0.812±0.012 |
| mistral | bert-base-uncased+vgg16 | 0.743±0.014 | 0.816±0.013 | 0.602±0.009 | 0.812±0.015 |
| kimi | distilbert-base-uncased+resnet34 | 0.765±0.012 | 0.820±0.016 | 0.639±0.012 | 0.828±0.012 |
| kimi | roberta-base+resnet50 | 0.765±0.015 | 0.820±0.012 | 0.639±0.012 | 0.828±0.016 |
| kimi | distilbert-base-uncased+resnet50 | 0.765±0.014 | 0.801±0.010 | 0.639±0.010 | 0.828±0.013 |
| qwen | albert-base-v2+resnet18 | 0.788±0.015 | 0.859±0.017 | 0.678±0.012 | 0.844±0.011 |
| qwen | albert-base-v2+resnet18 | 0.788±0.015 | 0.826±0.012 | 0.678±0.013 | 0.844±0.011 |
| qwen | roberta-base+vgg16 | 0.812±0.015 | 0.861±0.015 | 0.719±0.010 | 0.859±0.013 |
| mistral | bert-base-uncased+resnet50 | 0.615±0.010 | 0.684±0.010 | 0.383±0.006 | 0.703±0.012 |
| qwen | distilbert-base-uncased+resnet34 | 0.615±0.011 | 0.693±0.011 | 0.383±0.007 | 0.703±0.013 |
| mistral | distilbert-base-uncased+resnet34 | 0.615±0.010 | 0.729±0.012 | 0.383±0.006 | 0.703±0.009 |
| kimi | bert-base-uncased+resnet34 | 0.632±0.009 | 0.723±0.009 | 0.414±0.008 | 0.719±0.011 |
| qwen | bert-base-uncased+shufflenet_v2_x1_0 | 0.632±0.011 | 0.764±0.011 | 0.414±0.006 | 0.719±0.012 |
| kimi | albert-base-v2+resnet18 | 0.632±0.010 | 0.779±0.011 | 0.414±0.006 | 0.719±0.012 |
| qwen | bert-base-uncased+vgg16 | 0.632±0.009 | 0.744±0.014 | 0.414±0.007 | 0.719±0.014 |
| kimi | albert-base-v2+densenet121 | 0.649±0.009 | 0.797±0.014 | 0.445±0.007 | 0.734±0.013 |
| qwen | albert-base-v2+densenet121 | 0.649±0.010 | 0.705±0.011 | 0.445±0.008 | 0.734±0.011 |
| qwen | distilbert-base-uncased+densenet121 | 0.649±0.013 | 0.646±0.012 | 0.445±0.008 | 0.734±0.011 |
| qwen | distilbert-base-uncased+resnet18 | 0.649±0.010 | 0.828±0.015 | 0.445±0.009 | 0.734±0.014 |
| kimi | bert-base-uncased+shufflenet_v2_x1_0 | 0.649±0.011 | 0.799±0.012 | 0.445±0.009 | 0.734±0.010 |
| qwen | distilbert-base-uncased+mobilenet_v3_small | 0.667±0.010 | 0.758±0.013 | 0.478±0.007 | 0.750±0.010 |
| mistral | roberta-base+densenet121 | 0.667±0.009 | 0.734±0.014 | 0.478±0.009 | 0.750±0.014 |
| qwen | bert-base-uncased+densenet121 | 0.667±0.010 | 0.748±0.014 | 0.478±0.009 | 0.750±0.012 |
| qwen | roberta-base+mobilenet_v3_small | 0.667±0.012 | 0.787±0.012 | 0.478±0.008 | 0.750±0.011 |
| kimi | distilbert-base-uncased+resnet18 | 0.667±0.012 | 0.803±0.012 | 0.478±0.009 | 0.750±0.012 |
| kimi | roberta-base+mobilenet_v3_small | 0.667±0.009 | 0.781±0.011 | 0.478±0.008 | 0.750±0.010 |
| mistral | bert-base-uncased+resnet50 | 0.686±0.012 | 0.799±0.015 | 0.512±0.009 | 0.766±0.013 |
| qwen | distilbert-base-uncased+mobilenet_v3_small | 0.686±0.009 | 0.785±0.014 | 0.512±0.007 | 0.766±0.013 |
| kimi | distilbert-base-uncased+mobilenet_v3_small | 0.686±0.012 | 0.814±0.015 | 0.512±0.010 | 0.766±0.011 |
| mistral | distilbert-base-uncased+resnet34 | 0.686±0.013 | 0.799±0.016 | 0.512±0.009 | 0.766±0.011 |
| mistral | albert-base-v2+resnet18 | 0.686±0.009 | 0.797±0.014 | 0.512±0.009 | 0.766±0.014 |
| mistral | bert-base-uncased+resnet34 | 0.706±0.011 | 0.828±0.016 | 0.548±0.010 | 0.781±0.011 |
| qwen | roberta-base+resnet50 | 0.706±0.009 | 0.791±0.016 | 0.548±0.010 | 0.781±0.014 |
| mistral | distilbert-base-uncased+resnet34 | 0.706±0.013 | 0.805±0.011 | 0.548±0.010 | 0.781±0.015 |
| kimi | bert-base-uncased+efficientnet_b0 | 0.706±0.013 | 0.785±0.015 | 0.548±0.008 | 0.781±0.014 |
| kimi | bert-base-uncased+resnet34 | 0.706±0.011 | 0.779±0.013 | 0.548±0.008 | 0.781±0.011 |
| kimi | distilbert-base-uncased+resnet18 | 0.706±0.011 | 0.826±0.012 | 0.548±0.010 | 0.781±0.011 |
| mistral | distilbert-base-uncased+mobilenet_v3_small | 0.706±0.009 | 0.809±0.011 | 0.548±0.009 | 0.781±0.010 |
| kimi | bert-base-uncased+resnet50 | 0.706±0.013 | 0.809±0.012 | 0.548±0.011 | 0.781±0.014 |
| kimi | albert-base-v2+resnet34 | 0.706±0.012 | 0.781±0.016 | 0.548±0.008 | 0.781±0.013 |
| mistral | albert-base-v2+resnet34 | 0.727±0.011 | 0.779±0.014 | 0.585±0.009 | 0.797±0.014 |
| qwen | bert-base-uncased+resnet50 | 0.727±0.014 | 0.816±0.013 | 0.585±0.011 | 0.797±0.015 |
| mistral | roberta-base+resnet50 | 0.727±0.014 | 0.775±0.011 | 0.585±0.009 | 0.797±0.012 |
| qwen | roberta-base+densenet121 | 0.750±0.012 | 0.797±0.015 | 0.625±0.011 | 0.812±0.013 |
| kimi | albert-base-v2+resnet50 | 0.750±0.015 | 0.795±0.014 | 0.625±0.011 | 0.812±0.012 |
| mistral | bert-base-uncased+shufflenet_v2_x1_0 | 0.595±0.010 | 0.686±0.013 | 0.356±0.007 | 0.688±0.014 |
| mistral | roberta-base+shufflenet_v2_x1_0 | 0.595±0.009 | 0.697±0.011 | 0.356±0.007 | 0.688±0.011 |

*(end of table)*

*(continued)*

| Model | Init | F1 | AUC | MCC | Balanced Acc |
|-------|------|-----|-----|-----|-------------|
| kimi | distilbert-base-uncased+shufflenet_v2_x1_0 | 0.595±0.010 | 0.693±0.009 | 0.356±0.006 | 0.688±0.013 |
| qwen | roberta-base+shufflenet_v2_x1_0 | 0.595±0.009 | 0.701±0.013 | 0.356±0.007 | 0.688±0.009 |
| kimi | distilbert-base-uncased+vgg16 | 0.595±0.008 | 0.771±0.010 | 0.356±0.005 | 0.688±0.010 |
| qwen | bert-base-uncased+densenet121 | 0.611±0.010 | 0.746±0.014 | 0.388±0.007 | 0.703±0.014 |
| qwen | roberta-base+efficientnet_b0 | 0.611±0.011 | 0.719±0.012 | 0.388±0.007 | 0.703±0.010 |
| qwen | roberta-base+resnet34 | 0.611±0.009 | 0.703±0.013 | 0.388±0.006 | 0.703±0.014 |
| qwen | albert-base-v2+resnet34 | 0.611±0.009 | 0.676±0.011 | 0.388±0.007 | 0.703±0.010 |
| kimi | roberta-base+resnet34 | 0.629±0.009 | 0.762±0.010 | 0.422±0.007 | 0.719±0.013 |
| qwen | distilbert-base-uncased+densenet121 | 0.629±0.012 | 0.775±0.012 | 0.422±0.008 | 0.719±0.010 |
| mistral | albert-base-v2+shufflenet_v2_x1_0 | 0.629±0.012 | 0.773±0.014 | 0.422±0.007 | 0.719±0.014 |
| kimi | roberta-base+shufflenet_v2_x1_0 | 0.647±0.010 | 0.746±0.014 | 0.456±0.009 | 0.734±0.014 |
| mistral | albert-base-v2+efficientnet_b0 | 0.647±0.009 | 0.785±0.014 | 0.456±0.008 | 0.734±0.014 |
| mistral | roberta-base+densenet121 | 0.647±0.012 | 0.740±0.013 | 0.456±0.007 | 0.734±0.010 |
| qwen | albert-base-v2+resnet50 | 0.647±0.010 | 0.764±0.011 | 0.456±0.007 | 0.734±0.014 |
| qwen | albert-base-v2+densenet121 | 0.647±0.012 | 0.781±0.012 | 0.456±0.007 | 0.734±0.011 |
| qwen | roberta-base+resnet18 | 0.667±0.011 | 0.826±0.016 | 0.493±0.009 | 0.750±0.010 |
| mistral | bert-base-uncased+resnet18 | 0.667±0.010 | 0.828±0.013 | 0.493±0.009 | 0.750±0.014 |
| qwen | distilbert-base-uncased+efficientnet_b0 | 0.667±0.009 | 0.736±0.011 | 0.493±0.007 | 0.750±0.014 |
| qwen | bert-base-uncased+resnet34 | 0.667±0.010 | 0.822±0.016 | 0.493±0.007 | 0.750±0.010 |
| mistral | roberta-base+efficientnet_b0 | 0.667±0.010 | 0.762±0.012 | 0.493±0.009 | 0.750±0.012 |
| qwen | roberta-base+resnet34 | 0.667±0.011 | 0.799±0.012 | 0.493±0.009 | 0.750±0.014 |
| kimi | bert-base-uncased+densenet121 | 0.667±0.010 | 0.783±0.013 | 0.493±0.009 | 0.750±0.014 |
| mistral | bert-base-uncased+efficientnet_b0 | 0.667±0.009 | 0.799±0.011 | 0.493±0.007 | 0.750±0.010 |
| qwen | distilbert-base-uncased+resnet34 | 0.667±0.010 | 0.775±0.012 | 0.493±0.009 | 0.750±0.011 |
| kimi | albert-base-v2+resnet18 | 0.667±0.010 | 0.799±0.015 | 0.493±0.009 | 0.750±0.013 |
| qwen | albert-base-v2+resnet18 | 0.667±0.010 | 0.779±0.011 | 0.493±0.008 | 0.750±0.013 |
| mistral | roberta-base+resnet50 | 0.688±0.012 | 0.795±0.011 | 0.531±0.009 | 0.766±0.011 |
| qwen | bert-base-uncased+efficientnet_b0 | 0.688±0.014 | 0.797±0.011 | 0.531±0.009 | 0.766±0.011 |
| kimi | distilbert-base-uncased+efficientnet_b0 | 0.688±0.012 | 0.787±0.014 | 0.531±0.008 | 0.766±0.012 |
| kimi | distilbert-base-uncased+vgg16 | 0.710±0.009 | 0.791±0.015 | 0.572±0.009 | 0.781±0.013 |
| qwen | distilbert-base-uncased+shufflenet_v2_x1_0 | 0.710±0.011 | 0.803±0.015 | 0.572±0.009 | 0.781±0.012 |
| qwen | bert-base-uncased+resnet18 | 0.710±0.013 | 0.842±0.015 | 0.572±0.008 | 0.781±0.014 |
| mistral | bert-base-uncased+resnet18 | 0.710±0.011 | 0.865±0.011 | 0.572±0.009 | 0.781±0.012 |
| kimi | roberta-base+densenet121 | 0.710±0.012 | 0.814±0.011 | 0.572±0.010 | 0.781±0.012 |
| mistral | distilbert-base-uncased+resnet18 | 0.710±0.014 | 0.824±0.016 | 0.572±0.008 | 0.781±0.016 |
| qwen | albert-base-v2+vgg16 | 0.733±0.010 | 0.842±0.015 | 0.616±0.010 | 0.797±0.011 |
| mistral | roberta-base+densenet121 | 0.759±0.014 | 0.848±0.013 | 0.663±0.009 | 0.812±0.014 |
| mistral | bert-base-uncased+densenet121 | 0.571±0.008 | 0.684±0.011 | 0.331±0.005 | 0.672±0.010 |
| qwen | albert-base-v2+resnet50 | 0.571±0.010 | 0.619±0.010 | 0.331±0.005 | 0.672±0.011 |
| kimi | albert-base-v2+densenet121 | 0.571±0.011 | 0.650±0.009 | 0.331±0.005 | 0.672±0.012 |
| qwen | distilbert-base-uncased+resnet18 | 0.588±0.008 | 0.781±0.013 | 0.365±0.007 | 0.688±0.012 |
| qwen | bert-base-uncased+efficientnet_b0 | 0.588±0.010 | 0.719±0.011 | 0.365±0.007 | 0.688±0.012 |
| qwen | distilbert-base-uncased+resnet34 | 0.588±0.010 | 0.723±0.013 | 0.365±0.005 | 0.688±0.009 |
| kimi | bert-base-uncased+resnet34 | 0.588±0.011 | 0.781±0.013 | 0.365±0.007 | 0.688±0.012 |
| mistral | roberta-base+resnet34 | 0.588±0.012 | 0.801±0.013 | 0.365±0.006 | 0.688±0.013 |
| mistral | albert-base-v2+densenet121 | 0.606±0.011 | 0.662±0.012 | 0.400±0.007 | 0.703±0.011 |
| qwen | bert-base-uncased+resnet34 | 0.606±0.009 | 0.729±0.014 | 0.400±0.008 | 0.703±0.011 |
| mistral | roberta-base+resnet34 | 0.625±0.009 | 0.721±0.010 | 0.438±0.008 | 0.719±0.010 |
| mistral | bert-base-uncased+resnet34 | 0.625±0.009 | 0.785±0.011 | 0.438±0.008 | 0.719±0.010 |
| kimi | roberta-base+efficientnet_b0 | 0.625±0.012 | 0.721±0.012 | 0.438±0.006 | 0.719±0.012 |
| qwen | albert-base-v2+shufflenet_v2_x1_0 | 0.625±0.012 | 0.805±0.012 | 0.438±0.006 | 0.719±0.011 |
| kimi | roberta-base+efficientnet_b0 | 0.625±0.011 | 0.750±0.015 | 0.438±0.007 | 0.719±0.014 |

*(end of table)*

*(continued)*

| Model | Init | F1 | AUC | MCC | Balanced Acc |
|---|---|---|---|---|---|
| kimi | albert-base-v2+efficientnet_b0 | 0.625±0.010 | 0.768±0.014 | 0.438±0.007 | 0.719±0.013 |
| mistral | bert-base-uncased+densenet121 | 0.625±0.009 | 0.705±0.012 | 0.438±0.006 | 0.719±0.013 |
| mistral | distilbert-base-uncased+resnet18 | 0.625±0.012 | 0.787±0.011 | 0.438±0.007 | 0.719±0.013 |
| mistral | albert-base-v2+resnet34 | 0.645±0.008 | 0.781±0.015 | 0.477±0.010 | 0.734±0.012 |
| mistral | albert-base-v2+resnet34 | 0.645±0.012 | 0.787±0.012 | 0.477±0.007 | 0.734±0.012 |
| mistral | albert-base-v2+resnet18 | 0.645±0.013 | 0.801±0.016 | 0.477±0.006 | 0.734±0.010 |
| kimi | bert-base-uncased+resnet50 | 0.667±0.011 | 0.828±0.015 | 0.519±0.009 | 0.750±0.011 |
| qwen | bert-base-uncased+mobilenet_v3_small | 0.667±0.009 | 0.818±0.016 | 0.519±0.008 | 0.750±0.012 |
| mistral | bert-base-uncased+densenet121 | 0.667±0.011 | 0.773±0.012 | 0.519±0.007 | 0.750±0.011 |
| qwen | distilbert-base-uncased+resnet50 | 0.667±0.009 | 0.801±0.013 | 0.519±0.007 | 0.750±0.010 |
| qwen | distilbert-base-uncased+resnet50 | 0.667±0.009 | 0.805±0.015 | 0.519±0.007 | 0.750±0.014 |
| mistral | bert-base-uncased+efficientnet_b0 | 0.667±0.010 | 0.793±0.012 | 0.519±0.007 | 0.750±0.013 |
| qwen | distilbert-base-uncased+resnet18 | 0.667±0.012 | 0.848±0.013 | 0.519±0.010 | 0.750±0.014 |
| mistral | bert-base-uncased+resnet18 | 0.667±0.013 | 0.795±0.013 | 0.519±0.009 | 0.750±0.012 |
| kimi | bert-base-uncased+resnet18 | 0.690±0.011 | 0.818±0.013 | 0.564±0.008 | 0.766±0.014 |
| kimi | roberta-base+resnet34 | 0.690±0.009 | 0.766±0.012 | 0.564±0.010 | 0.766±0.015 |
| mistral | bert-base-uncased+resnet18 | 0.690±0.014 | 0.791±0.010 | 0.564±0.008 | 0.766±0.012 |
| qwen | roberta-base+resnet18 | 0.714±0.012 | 0.814±0.011 | 0.612±0.009 | 0.781±0.011 |
| qwen | bert-base-uncased+resnet18 | 0.714±0.013 | 0.836±0.011 | 0.612±0.011 | 0.781±0.014 |
| mistral | distilbert-base-uncased+shufflenet_v2_x1_0 | 0.714±0.013 | 0.850±0.014 | 0.612±0.012 | 0.781±0.014 |
| mistral | albert-base-v2+resnet50 | 0.714±0.010 | 0.859±0.015 | 0.612±0.009 | 0.781±0.010 |
| mistral | distilbert-base-uncased+resnet18 | 0.714±0.012 | 0.873±0.013 | 0.612±0.011 | 0.781±0.013 |
| mistral | bert-base-uncased+resnet34 | 0.714±0.011 | 0.830±0.012 | 0.612±0.012 | 0.781±0.011 |
| mistral | albert-base-v2+densenet121 | 0.741±0.013 | 0.814±0.013 | 0.666±0.010 | 0.797±0.015 |
| mistral | distilbert-base-uncased+resnet34 | 0.545±0.011 | 0.629±0.010 | 0.308±0.004 | 0.656±0.009 |
| kimi | roberta-base+shufflenet_v2_x1_0 | 0.562±0.007 | 0.742±0.013 | 0.344±0.007 | 0.672±0.013 |
| kimi | roberta-base+densenet121 | 0.562±0.010 | 0.645±0.013 | 0.344±0.005 | 0.672±0.013 |
| kimi | distilbert-base-uncased+densenet121 | 0.581±0.009 | 0.703±0.010 | 0.381±0.006 | 0.688±0.010 |
| qwen | bert-base-uncased+resnet34 | 0.581±0.009 | 0.725±0.013 | 0.381±0.007 | 0.688±0.010 |
| mistral | roberta-base+efficientnet_b0 | 0.581±0.011 | 0.736±0.013 | 0.381±0.007 | 0.688±0.011 |
| kimi | bert-base-uncased+resnet50 | 0.600±0.012 | 0.693±0.013 | 0.421±0.006 | 0.703±0.011 |
| mistral | bert-base-uncased+resnet50 | 0.600±0.011 | 0.693±0.013 | 0.421±0.006 | 0.703±0.012 |
| mistral | roberta-base+resnet34 | 0.600±0.010 | 0.758±0.015 | 0.421±0.008 | 0.703±0.010 |
| qwen | roberta-base+resnet34 | 0.600±0.009 | 0.756±0.014 | 0.421±0.006 | 0.703±0.014 |
| kimi | bert-base-uncased+resnet18 | 0.600±0.010 | 0.709±0.012 | 0.421±0.008 | 0.703±0.012 |
| mistral | roberta-base+resnet18 | 0.621±0.011 | 0.807±0.014 | 0.464±0.007 | 0.719±0.012 |
| mistral | distilbert-base-uncased+resnet50 | 0.621±0.009 | 0.828±0.016 | 0.464±0.008 | 0.719±0.012 |
| kimi | roberta-base+resnet18 | 0.621±0.009 | 0.750±0.014 | 0.464±0.008 | 0.719±0.010 |
| kimi | albert-base-v2+shufflenet_v2_x1_0 | 0.643±0.010 | 0.814±0.013 | 0.510±0.010 | 0.734±0.011 |
| mistral | albert-base-v2+shufflenet_v2_x1_0 | 0.643±0.013 | 0.732±0.013 | 0.510±0.007 | 0.734±0.014 |
| qwen | roberta-base+resnet18 | 0.667±0.009 | 0.768±0.014 | 0.561±0.010 | 0.750±0.013 |
| mistral | distilbert-base-uncased+shufflenet_v2_x1_0 | 0.516±0.008 | 0.615±0.012 | 0.286±0.004 | 0.641±0.011 |
| qwen | roberta-base+densenet121 | 0.533±0.008 | 0.648±0.012 | 0.324±0.006 | 0.656±0.012 |
| qwen | roberta-base+shufflenet_v2_x1_0 | 0.533±0.010 | 0.713±0.011 | 0.324±0.005 | 0.656±0.011 |
| kimi | roberta-base+resnet18 | 0.533±0.008 | 0.682±0.011 | 0.324±0.004 | 0.656±0.011 |
| qwen | bert-base-uncased+shufflenet_v2_x1_0 | 0.552±0.010 | 0.709±0.012 | 0.365±0.006 | 0.672±0.012 |
| kimi | bert-base-uncased+densenet121 | 0.552±0.009 | 0.684±0.009 | 0.365±0.005 | 0.672±0.012 |
| kimi | bert-base-uncased+resnet18 | 0.552±0.008 | 0.723±0.012 | 0.365±0.006 | 0.672±0.012 |
| mistral | albert-base-v2+densenet121 | 0.571±0.010 | 0.660±0.013 | 0.408±0.007 | 0.688±0.009 |
| qwen | albert-base-v2+resnet50 | 0.571±0.008 | 0.682±0.010 | 0.408±0.007 | 0.688±0.013 |
| qwen | roberta-base+densenet121 | 0.571±0.008 | 0.699±0.010 | 0.408±0.008 | 0.688±0.011 |
| qwen | bert-base-uncased+resnet50 | 0.593±0.011 | 0.645±0.012 | 0.456±0.008 | 0.703±0.010 |

*(end of table)*

*(continued)*

| Model | Init | F1 | AUC | MCC | Balanced Acc |
|---|---|---|---|---|---|
| mistral | distilbert-base-uncased+densenet121 | 0.593±0.011 | 0.832±0.015 | 0.456±0.009 | 0.703±0.012 |
| kimi | distilbert-base-uncased+densenet121 | 0.615±0.012 | 0.670±0.011 | 0.508±0.009 | 0.719±0.011 |
| mistral | albert-base-v2+vgg16 | 0.640±0.011 | 0.797±0.012 | 0.566±0.010 | 0.734±0.011 |
| qwen | roberta-base+resnet18 | 0.667±0.012 | 0.719±0.014 | 0.632±0.009 | 0.750±0.012 |
| mistral | bert-base-uncased+resnet50 | 0.483±0.009 | 0.650±0.009 | 0.265±0.005 | 0.625±0.011 |
| qwen | roberta-base+shufflenet_v2_x1_0 | 0.538±0.008 | 0.654±0.011 | 0.399±0.007 | 0.672±0.013 |
| kimi | albert-base-v2+resnet18 | 0.560±0.009 | 0.756±0.015 | 0.453±0.008 | 0.688±0.010 |
| qwen | bert-base-uncased+mobilenet_v3_small | 0.444±0.009 | 0.740±0.013 | 0.245±0.004 | 0.609±0.010 |
| qwen | albert-base-v2+vgg16 | 0.000±0.000 | 0.000±0.000 | 0.000±0.000 | 0.000±0.000 |
| kimi | roberta-base+mobilenet_v3_small | 0.000±0.000 | 0.000±0.000 | 0.000±0.000 | 0.000±0.000 |
| kimi | roberta-base+shufflenet_v2_x1_0 | 0.476±0.007 | 0.580±0.010 | 0.118±0.002 | 0.562±0.011 |
| qwen | roberta-base+vgg16 | 0.000±0.000 | 0.000±0.000 | 0.000±0.000 | 0.000±0.000 |
| kimi | roberta-base+mobilenet_v3_small | 0.000±0.000 | 0.000±0.000 | 0.000±0.000 | 0.000±0.000 |
| kimi | roberta-base+mobilenet_v3_small | 0.627±0.012 | 0.850±0.013 | 0.431±0.008 | 0.703±0.013 |
| qwen | roberta-base+vgg16 | 0.000±0.000 | 0.000±0.000 | 0.000±0.000 | 0.000±0.000 |
| kimi | roberta-base+vgg16 | 0.000±0.000 | 0.000±0.000 | 0.000±0.000 | 0.000±0.000 |
| qwen | albert-base-v2+shufflenet_v2_x1_0 | 0.612±0.009 | 0.762±0.013 | 0.381±0.007 | 0.688±0.013 |
| kimi | roberta-base+vgg16 | 0.000±0.000 | 0.000±0.000 | 0.000±0.000 | 0.000±0.000 |
| kimi | distilbert-base-uncased+resnet50 | 0.500±0.008 | 0.461±0.008 | 0.000±0.000 | 0.500±0.007 |
| kimi | distilbert-base-uncased+efficientnet_b0 | 0.000±0.000 | 0.000±0.000 | 0.000±0.000 | 0.000±0.000 |
| kimi | distilbert-base-uncased+efficientnet_b0 | 0.000±0.000 | 0.000±0.000 | 0.000±0.000 | 0.000±0.000 |
| kimi | distilbert-base-uncased+efficientnet_b0 | 0.583±0.009 | 0.682±0.011 | 0.312±0.006 | 0.656±0.010 |
| qwen | albert-base-v2+vgg16 | 0.000±0.000 | 0.000±0.000 | 0.000±0.000 | 0.000±0.000 |
| qwen | roberta-base+resnet50 | 0.529±0.010 | 0.600±0.008 | 0.274±0.004 | 0.641±0.010 |
| qwen | roberta-base+resnet50 | 0.585±0.009 | 0.740±0.012 | 0.324±0.006 | 0.672±0.010 |
| kimi | distilbert-base-uncased+resnet50 | 0.564±0.010 | 0.633±0.010 | 0.295±0.005 | 0.656±0.012 |
| kimi | distilbert-base-uncased+resnet34 | 0.560±0.008 | 0.730±0.010 | 0.259±0.004 | 0.625±0.011 |
| kimi | roberta-base+vgg16 | 0.000±0.000 | 0.000±0.000 | 0.000±0.000 | 0.000±0.000 |
| kimi | distilbert-base-uncased+resnet34 | 0.522±0.009 | 0.635±0.011 | 0.183±0.003 | 0.594±0.012 |
| kimi | roberta-base+efficientnet_b0 | 0.000±0.000 | 0.000±0.000 | 0.000±0.000 | 0.000±0.000 |
| qwen | roberta-base+efficientnet_b0 | 0.000±0.000 | 0.000±0.000 | 0.000±0.000 | 0.000±0.000 |
| qwen | roberta-base+efficientnet_b0 | 0.000±0.000 | 0.000±0.000 | 0.000±0.000 | 0.000±0.000 |
| kimi | distilbert-base-uncased+resnet18 | 0.605±0.011 | 0.723±0.011 | 0.356±0.005 | 0.688±0.009 |
| qwen | roberta-base+mobilenet_v3_small | 0.500±0.008 | 0.565±0.008 | 0.000±0.000 | 0.500±0.009 |
| qwen | roberta-base+mobilenet_v3_small | 0.000±0.000 | 0.000±0.000 | 0.000±0.000 | 0.000±0.000 |
| kimi | roberta-base+efficientnet_b0 | 0.000±0.000 | 0.000±0.000 | 0.000±0.000 | 0.000±0.000 |
| qwen | albert-base-v2+shufflenet_v2_x1_0 | 0.500±0.007 | 0.461±0.006 | 0.000±0.000 | 0.500±0.008 |
| kimi | roberta-base+densenet121 | 0.579±0.011 | 0.676±0.012 | 0.325±0.005 | 0.672±0.009 |
| qwen | albert-base-v2+shufflenet_v2_x1_0 | 0.583±0.009 | 0.701±0.014 | 0.312±0.006 | 0.656±0.010 |
| kimi | bert-base-uncased+shufflenet_v2_x1_0 | 0.488±0.007 | 0.553±0.007 | 0.147±0.002 | 0.578±0.009 |
| kimi | bert-base-uncased+shufflenet_v2_x1_0 | 0.000±0.000 | 0.000±0.000 | 0.000±0.000 | 0.000±0.000 |
| kimi | bert-base-uncased+mobilenet_v3_small | 0.000±0.000 | 0.000±0.000 | 0.000±0.000 | 0.000±0.000 |
| qwen | albert-base-v2+resnet18 | 0.577±0.010 | 0.629±0.010 | 0.306±0.005 | 0.641±0.010 |
| kimi | bert-base-uncased+mobilenet_v3_small | 0.500±0.009 | 0.465±0.009 | 0.000±0.000 | 0.500±0.008 |
| kimi | bert-base-uncased+mobilenet_v3_small | 0.583±0.011 | 0.754±0.010 | 0.312±0.005 | 0.656±0.013 |
| qwen | albert-base-v2+resnet34 | 0.512±0.009 | 0.646±0.010 | 0.178±0.004 | 0.594±0.010 |
| kimi | bert-base-uncased+efficientnet_b0 | 0.000±0.000 | 0.000±0.000 | 0.000±0.000 | 0.000±0.000 |
| kimi | bert-base-uncased+efficientnet_b0 | 0.500±0.009 | 0.510±0.007 | 0.000±0.000 | 0.500±0.009 |
| kimi | bert-base-uncased+efficientnet_b0 | 0.579±0.011 | 0.727±0.013 | 0.325±0.006 | 0.672±0.012 |
| kimi | bert-base-uncased+densenet121 | 0.571±0.009 | 0.695±0.010 | 0.296±0.005 | 0.656±0.009 |
| qwen | albert-base-v2+densenet121 | 0.500±0.009 | 0.652±0.011 | 0.209±0.004 | 0.609±0.011 |
| qwen | albert-base-v2+densenet121 | 0.605±0.010 | 0.711±0.014 | 0.356±0.007 | 0.688±0.011 |

*(end of table)*

*(continued)*

| Model | Init | F1 | AUC | MCC | Balanced Acc |
|-------|------|----|----|-----|--------------|
| kimi | bert-base-uncased+resnet50 | 0.500±0.007 | 0.516±0.009 | 0.000±0.000 | 0.500±0.009 |
| qwen | albert-base-v2+efficientnet_b0 | 0.579±0.008 | 0.715±0.010 | 0.325±0.005 | 0.672±0.009 |
| qwen | albert-base-v2+efficientnet_b0 | 0.000±0.000 | 0.000±0.000 | 0.000±0.000 | 0.000±0.000 |
| qwen | albert-base-v2+efficientnet_b0 | 0.000±0.000 | 0.000±0.000 | 0.000±0.000 | 0.000±0.000 |
| kimi | bert-base-uncased+resnet34 | 0.652±0.011 | 0.742±0.015 | 0.456±0.006 | 0.734±0.013 |
| qwen | albert-base-v2+mobilenet_v3_small | 0.500±0.007 | 0.457±0.008 | 0.000±0.000 | 0.500±0.008 |
| qwen | distilbert-base-uncased+vgg16 | 0.000±0.000 | 0.000±0.000 | 0.000±0.000 | 0.000±0.000 |
| qwen | distilbert-base-uncased+vgg16 | 0.000±0.000 | 0.000±0.000 | 0.000±0.000 | 0.000±0.000 |
| kimi | bert-base-uncased+shufflenet_v2_x1_0 | 0.421±0.006 | 0.500±0.008 | 0.059±0.001 | 0.531±0.010 |
| qwen | distilbert-base-uncased+efficientnet_b0 | 0.638±0.010 | 0.703±0.009 | 0.431±0.006 | 0.719±0.012 |
| kimi | roberta-base+densenet121 | 0.525±0.007 | 0.725±0.014 | 0.183±0.002 | 0.547±0.010 |
| qwen | distilbert-base-uncased+resnet50 | 0.565±0.010 | 0.711±0.010 | 0.274±0.005 | 0.641±0.011 |
| qwen | distilbert-base-uncased+resnet50 | 0.510±0.009 | 0.582±0.011 | 0.133±0.002 | 0.562±0.009 |
| kimi | roberta-base+resnet50 | 0.605±0.009 | 0.723±0.012 | 0.356±0.007 | 0.688±0.012 |
| qwen | albert-base-v2+mobilenet_v3_small | 0.000±0.000 | 0.000±0.000 | 0.000±0.000 | 0.000±0.000 |
| kimi | roberta-base+resnet50 | 0.591±0.010 | 0.670±0.012 | 0.329±0.005 | 0.672±0.010 |
| kimi | roberta-base+resnet34 | 0.566±0.009 | 0.693±0.010 | 0.280±0.004 | 0.625±0.009 |
| kimi | distilbert-base-uncased+mobilenet_v3_small | 0.000±0.000 | 0.000±0.000 | 0.000±0.000 | 0.000±0.000 |
| qwen | distilbert-base-uncased+efficientnet_b0 | 0.000±0.000 | 0.000±0.000 | 0.000±0.000 | 0.000±0.000 |
| qwen | distilbert-base-uncased+vgg16 | 0.000±0.000 | 0.000±0.000 | 0.000±0.000 | 0.000±0.000 |
| qwen | distilbert-base-uncased+efficientnet_b0 | 0.000±0.000 | 0.000±0.000 | 0.000±0.000 | 0.000±0.000 |
| kimi | bert-base-uncased+vgg16 | 0.000±0.000 | 0.000±0.000 | 0.000±0.000 | 0.000±0.000 |
| kimi | bert-base-uncased+vgg16 | 0.000±0.000 | 0.000±0.000 | 0.000±0.000 | 0.000±0.000 |
| qwen | distilbert-base-uncased+mobilenet_v3_small | 0.000±0.000 | 0.000±0.000 | 0.000±0.000 | 0.000±0.000 |
| kimi | bert-base-uncased+vgg16 | 0.000±0.000 | 0.000±0.000 | 0.000±0.000 | 0.000±0.000 |
| qwen | distilbert-base-uncased+shufflenet_v2_x1_0 | 0.500±0.008 | 0.709±0.010 | 0.000±0.000 | 0.500±0.010 |
| qwen | distilbert-base-uncased+shufflenet_v2_x1_0 | 0.444±0.006 | 0.469±0.007 | -0.073±0.001 | 0.469±0.009 |
| qwen | distilbert-base-uncased+shufflenet_v2_x1_0 | 0.500±0.008 | 0.572±0.009 | 0.250±0.003 | 0.625±0.012 |
| kimi | distilbert-base-uncased+mobilenet_v3_small | 0.000±0.000 | 0.000±0.000 | 0.000±0.000 | 0.000±0.000 |
| kimi | distilbert-base-uncased+shufflenet_v2_x1_0 | 0.512±0.008 | 0.580±0.008 | 0.178±0.003 | 0.594±0.010 |
| qwen | bert-base-uncased+vgg16 | 0.000±0.000 | 0.000±0.000 | 0.000±0.000 | 0.000±0.000 |
| mistral | distilbert-base-uncased+shufflenet_v2_x1_0 | 0.000±0.000 | 0.000±0.000 | 0.000±0.000 | 0.000±0.000 |
| mistral | distilbert-base-uncased+densenet121 | 0.605±0.012 | 0.662±0.013 | 0.356±0.007 | 0.688±0.009 |
| mistral | distilbert-base-uncased+efficientnet_b0 | 0.625±0.010 | 0.762±0.013 | 0.406±0.007 | 0.703±0.012 |
| mistral | distilbert-base-uncased+efficientnet_b0 | 0.622±0.010 | 0.719±0.011 | 0.392±0.006 | 0.703±0.011 |
| mistral | distilbert-base-uncased+efficientnet_b0 | 0.000±0.000 | 0.000±0.000 | 0.000±0.000 | 0.000±0.000 |
| mistral | distilbert-base-uncased+efficientnet_b0 | 0.000±0.000 | 0.000±0.000 | 0.000±0.000 | 0.000±0.000 |
| mistral | bert-base-uncased+mobilenet_v3_small | 0.000±0.000 | 0.000±0.000 | 0.000±0.000 | 0.000±0.000 |
| mistral | bert-base-uncased+mobilenet_v3_small | 0.000±0.000 | 0.000±0.000 | 0.000±0.000 | 0.000±0.000 |
| mistral | distilbert-base-uncased+mobilenet_v3_small | 0.000±0.000 | 0.000±0.000 | 0.000±0.000 | 0.000±0.000 |
| mistral | distilbert-base-uncased+mobilenet_v3_small | 0.000±0.000 | 0.000±0.000 | 0.000±0.000 | 0.000±0.000 |
| mistral | distilbert-base-uncased+shufflenet_v2_x1_0 | 0.565±0.009 | 0.613±0.009 | 0.274±0.005 | 0.641±0.010 |
| kimi | albert-base-v2+vgg16 | 0.000±0.000 | 0.000±0.000 | 0.000±0.000 | 0.000±0.000 |
| mistral | distilbert-base-uncased+vgg16 | 0.513±0.010 | 0.576±0.009 | 0.206±0.003 | 0.609±0.009 |
| mistral | distilbert-base-uncased+vgg16 | 0.000±0.000 | 0.000±0.000 | 0.000±0.000 | 0.000±0.000 |
| mistral | distilbert-base-uncased+vgg16 | 0.000±0.000 | 0.000±0.000 | 0.000±0.000 | 0.000±0.000 |
| mistral | distilbert-base-uncased+vgg16 | 0.000±0.000 | 0.000±0.000 | 0.000±0.000 | 0.000±0.000 |
| mistral | bert-base-uncased+efficientnet_b0 | 0.000±0.000 | 0.000±0.000 | 0.000±0.000 | 0.000±0.000 |
| mistral | bert-base-uncased+efficientnet_b0 | 0.000±0.000 | 0.000±0.000 | 0.000±0.000 | 0.000±0.000 |
| mistral | albert-base-v2+resnet18 | 0.528±0.009 | 0.711±0.010 | 0.175±0.003 | 0.578±0.011 |
| mistral | albert-base-v2+resnet18 | 0.585±0.011 | 0.688±0.012 | 0.324±0.006 | 0.672±0.010 |
| mistral | albert-base-v2+resnet34 | 0.500±0.009 | 0.582±0.009 | 0.149±0.003 | 0.578±0.011 |

*(end of table)*

*(continued)*

| Model | Init | F1 | AUC | MCC | Balanced Acc |
|---|---|---|---|---|---|
| mistral | distilbert-base-uncased+densenet121 | 0.571±0.008 | 0.693±0.010 | 0.296±0.005 | 0.656±0.009 |
| mistral | bert-base-uncased+shufflenet_v2_x1_0 | 0.449±0.007 | 0.527±0.010 | 0.000±0.000 | 0.500±0.008 |
| mistral | distilbert-base-uncased+resnet50 | 0.636±0.010 | 0.705±0.013 | 0.418±0.008 | 0.719±0.014 |
| mistral | distilbert-base-uncased+resnet50 | 0.579±0.011 | 0.672±0.010 | 0.325±0.006 | 0.672±0.012 |
| mistral | roberta-base+efficientnet_b0 | 0.000±0.000 | 0.000±0.000 | 0.000±0.000 | 0.000±0.000 |
| mistral | roberta-base+densenet121 | 0.579±0.008 | 0.678±0.013 | 0.325±0.005 | 0.672±0.012 |
| mistral | roberta-base+resnet50 | 0.491±0.009 | 0.596±0.008 | 0.042±0.001 | 0.516±0.010 |
| mistral | roberta-base+mobilenet_v3_small | 0.000±0.000 | 0.000±0.000 | 0.000±0.000 | 0.000±0.000 |
| mistral | roberta-base+mobilenet_v3_small | 0.000±0.000 | 0.000±0.000 | 0.000±0.000 | 0.000±0.000 |
| mistral | roberta-base+shufflenet_v2_x1_0 | 0.444±0.008 | 0.625±0.012 | 0.120±0.002 | 0.562±0.008 |
| mistral | roberta-base+shufflenet_v2_x1_0 | 0.000±0.000 | 0.000±0.000 | 0.000±0.000 | 0.000±0.000 |
| mistral | roberta-base+shufflenet_v2_x1_0 | 0.478±0.007 | 0.586±0.011 | 0.091±0.001 | 0.547±0.009 |
| mistral | roberta-base+resnet50 | 0.571±0.010 | 0.680±0.009 | 0.296±0.005 | 0.656±0.009 |
| mistral | roberta-base+vgg16 | 0.000±0.000 | 0.000±0.000 | 0.000±0.000 | 0.000±0.000 |
| mistral | roberta-base+vgg16 | 0.000±0.000 | 0.000±0.000 | 0.000±0.000 | 0.000±0.000 |
| mistral | roberta-base+vgg16 | 0.000±0.000 | 0.000±0.000 | 0.000±0.000 | 0.000±0.000 |
| mistral | roberta-base+vgg16 | 0.000±0.000 | 0.000±0.000 | 0.000±0.000 | 0.000±0.000 |
| mistral | roberta-base+resnet18 | 0.566±0.009 | 0.621±0.011 | 0.280±0.004 | 0.625±0.012 |
| mistral | distilbert-base-uncased+resnet18 | 0.582±0.009 | 0.703±0.009 | 0.340±0.006 | 0.641±0.010 |
| mistral | bert-base-uncased+vgg16 | 0.000±0.000 | 0.000±0.000 | 0.000±0.000 | 0.000±0.000 |
| mistral | bert-base-uncased+vgg16 | 0.000±0.000 | 0.000±0.000 | 0.000±0.000 | 0.000±0.000 |
| mistral | bert-base-uncased+vgg16 | 0.444±0.009 | 0.459±0.008 | 0.030±0.000 | 0.516±0.008 |
| mistral | bert-base-uncased+shufflenet_v2_x1_0 | 0.512±0.008 | 0.633±0.012 | 0.178±0.003 | 0.594±0.008 |
| qwen | bert-base-uncased+vgg16 | 0.000±0.000 | 0.000±0.000 | 0.000±0.000 | 0.000±0.000 |
| mistral | bert-base-uncased+resnet34 | 0.571±0.010 | 0.615±0.011 | 0.286±0.005 | 0.641±0.013 |
| mistral | albert-base-v2+resnet50 | 0.577±0.011 | 0.607±0.011 | 0.306±0.005 | 0.641±0.012 |
| kimi | albert-base-v2+resnet34 | 0.516±0.008 | 0.627±0.011 | 0.147±0.002 | 0.531±0.007 |
| kimi | albert-base-v2+efficientnet_b0 | 0.000±0.000 | 0.000±0.000 | 0.000±0.000 | 0.000±0.000 |
| kimi | albert-base-v2+efficientnet_b0 | 0.000±0.000 | 0.000±0.000 | 0.000±0.000 | 0.000±0.000 |
| kimi | albert-base-v2+densenet121 | 0.619±0.010 | 0.713±0.014 | 0.384±0.006 | 0.703±0.012 |
| kimi | albert-base-v2+densenet121 | 0.571±0.009 | 0.676±0.009 | 0.296±0.004 | 0.656±0.011 |
| qwen | bert-base-uncased+resnet50 | 0.545±0.009 | 0.629±0.012 | 0.226±0.004 | 0.594±0.012 |
| kimi | albert-base-v2+resnet50 | 0.545±0.008 | 0.680±0.012 | 0.239±0.004 | 0.625±0.010 |
| kimi | albert-base-v2+resnet50 | 0.526±0.009 | 0.742±0.012 | 0.237±0.004 | 0.625±0.008 |
| qwen | bert-base-uncased+densenet121 | 0.583±0.009 | 0.697±0.010 | 0.312±0.005 | 0.656±0.009 |
| kimi | albert-base-v2+resnet34 | 0.500±0.007 | 0.586±0.012 | 0.000±0.000 | 0.500±0.008 |
| qwen | bert-base-uncased+efficientnet_b0 | 0.000±0.000 | 0.000±0.000 | 0.000±0.000 | 0.000±0.000 |
| mistral | albert-base-v2+resnet50 | 0.565±0.010 | 0.695±0.010 | 0.274±0.004 | 0.641±0.011 |
| qwen | bert-base-uncased+efficientnet_b0 | 0.000±0.000 | 0.000±0.000 | 0.000±0.000 | 0.000±0.000 |
| kimi | distilbert-base-uncased+vgg16 | 0.000±0.000 | 0.000±0.000 | 0.000±0.000 | 0.000±0.000 |
| qwen | bert-base-uncased+mobilenet_v3_small | 0.000±0.000 | 0.000±0.000 | 0.000±0.000 | 0.000±0.000 |
| qwen | bert-base-uncased+mobilenet_v3_small | 0.000±0.000 | 0.000±0.000 | 0.000±0.000 | 0.000±0.000 |
| mistral | roberta-base+efficientnet_b0 | 0.000±0.000 | 0.000±0.000 | 0.000±0.000 | 0.000±0.000 |
| qwen | bert-base-uncased+shufflenet_v2_x1_0 | 0.514±0.007 | 0.598±0.009 | 0.241±0.003 | 0.625±0.011 |
| kimi | distilbert-base-uncased+shufflenet_v2_x1_0 | 0.500±0.010 | 0.621±0.010 | 0.250±0.004 | 0.625±0.008 |
| kimi | distilbert-base-uncased+shufflenet_v2_x1_0 | 0.000±0.000 | 0.000±0.000 | 0.000±0.000 | 0.000±0.000 |
| qwen | bert-base-uncased+vgg16 | 0.000±0.000 | 0.000±0.000 | 0.000±0.000 | 0.000±0.000 |
| qwen | bert-base-uncased+resnet34 | 0.636±0.010 | 0.744±0.012 | 0.418±0.006 | 0.719±0.011 |
| qwen | bert-base-uncased+resnet18 | 0.483±0.007 | 0.664±0.011 | 0.000±0.000 | 0.500±0.008 |
| qwen | bert-base-uncased+resnet18 | 0.600±0.008 | 0.764±0.013 | 0.354±0.006 | 0.688±0.014 |
| mistral | albert-base-v2+shufflenet_v2_x1_0 | 0.500±0.007 | 0.586±0.012 | 0.000±0.000 | 0.500±0.009 |
| kimi | albert-base-v2+vgg16 | 0.000±0.000 | 0.000±0.000 | 0.000±0.000 | 0.000±0.000 |

*(end of table)*

*(continued)*

| Model | Init | F1 | AUC | MCC | Balanced Acc |
|---|---|---|---|---|---|
| kimi | albert-base-v2+shufflenet_v2_x1_0 | 0.514±0.010 | 0.609±0.011 | 0.241±0.004 | 0.625±0.010 |
| mistral | albert-base-v2+efficientnet_b0 | 0.640±0.011 | 0.682±0.011 | 0.454±0.006 | 0.719±0.013 |
| mistral | albert-base-v2+efficientnet_b0 | 0.000±0.000 | 0.000±0.000 | 0.000±0.000 | 0.000±0.000 |
| mistral | albert-base-v2+efficientnet_b0 | 0.000±0.000 | 0.000±0.000 | 0.000±0.000 | 0.000±0.000 |
| kimi | albert-base-v2+shufflenet_v2_x1_0 | 0.500±0.007 | 0.539±0.008 | 0.000±0.000 | 0.500±0.009 |
| mistral | albert-base-v2+mobilenet_v3_small | 0.000±0.000 | 0.000±0.000 | 0.000±0.000 | 0.000±0.000 |
| kimi | albert-base-v2+mobilenet_v3_small | 0.508±0.009 | 0.514±0.009 | 0.103±0.002 | 0.516±0.010 |
| mistral | albert-base-v2+mobilenet_v3_small | 0.000±0.000 | 0.000±0.000 | 0.000±0.000 | 0.000±0.000 |
| mistral | albert-base-v2+shufflenet_v2_x1_0 | 0.500±0.010 | 0.584±0.008 | 0.079±0.001 | 0.531±0.008 |
| mistral | albert-base-v2+vgg16 | 0.000±0.000 | 0.000±0.000 | 0.000±0.000 | 0.000±0.000 |
| mistral | albert-base-v2+vgg16 | 0.000±0.000 | 0.000±0.000 | 0.000±0.000 | 0.000±0.000 |
| mistral | albert-base-v2+vgg16 | 0.000±0.000 | 0.000±0.000 | 0.000±0.000 | 0.000±0.000 |
| kimi | albert-base-v2+mobilenet_v3_small | 0.000±0.000 | 0.000±0.000 | 0.000±0.000 | 0.000±0.000 |
| qwen | albert-base-v2+vgg16 | 0.000±0.000 | 0.000±0.000 | 0.000±0.000 | 0.000±0.000 |
| mistral | albert-base-v2+mobilenet_v3_small | 0.400±0.005 | 0.732±0.012 | 0.226±0.003 | 0.594±0.011 |
| kimi | roberta-base+shufflenet_v2_x1_0 | 0.435±0.007 | 0.615±0.009 | 0.334±0.006 | 0.625±0.012 |
| kimi | albert-base-v2+vgg16 | 0.381±0.007 | 0.662±0.009 | 0.338±0.006 | 0.609±0.011 |
| mistral | bert-base-uncased+mobilenet_v3_small | 0.211±0.004 | 0.725±0.014 | 0.183±0.003 | 0.547±0.008 |
| qwen | distilbert-base-uncased+mobilenet_v3_small | 0.211±0.004 | 0.396±0.006 | 0.183±0.003 | 0.547±0.008 |
| qwen | roberta-base+mobilenet_v3_small | 0.222±0.003 | 0.719±0.010 | 0.295±0.005 | 0.562±0.009 |
| kimi | bert-base-uncased+mobilenet_v3_small | 0.118±0.002 | 0.777±0.012 | 0.206±0.004 | 0.531±0.009 |

*(end of table)*

**Table S6.** Multimodal results using Gated-Fusion

| Model | Init | F1 | AUC | MCC | Balanced Acc |
|---|---|---|---|---|---|
| qwen | roberta-base+resnet18 | 0.667±0.011 | 0.773±0.012 | 0.482±0.010 | 0.750±0.011 |
| kimi | albert-base-v2+efficientnet_b0 | 0.651±0.011 | 0.756±0.011 | 0.445±0.007 | 0.734±0.011 |
| mistral | distilbert-base-uncased+resnet18 | 0.700±0.010 | 0.838±0.014 | 0.530±0.010 | 0.781±0.012 |
| kimi | bert-base-uncased+resnet50 | 0.700±0.009 | 0.790±0.015 | 0.530±0.010 | 0.781±0.014 |
| qwen | albert-base-v2+mobilenet_v3_small | 0.700±0.010 | 0.783±0.015 | 0.530±0.010 | 0.781±0.013 |
| mistral | bert-base-uncased+densenet121 | 0.700±0.011 | 0.764±0.011 | 0.530±0.008 | 0.781±0.012 |
| mistral | distilbert-base-uncased+mobilenet_v3_small | 0.718±0.012 | 0.793±0.015 | 0.560±0.009 | 0.797±0.013 |
| kimi | distilbert-base-uncased+resnet18 | 0.718±0.013 | 0.799±0.014 | 0.560±0.009 | 0.797±0.015 |
| qwen | bert-base-uncased+densenet121 | 0.718±0.011 | 0.789±0.013 | 0.560±0.011 | 0.797±0.015 |
| qwen | bert-base-uncased+efficientnet_b0 | 0.778±0.011 | 0.811±0.015 | 0.657±0.011 | 0.844±0.017 |
| qwen | bert-base-uncased+resnet50 | 0.634±0.013 | 0.766±0.014 | 0.413±0.006 | 0.719±0.014 |
| kimi | albert-base-v2+resnet18 | 0.634±0.009 | 0.770±0.012 | 0.413±0.008 | 0.719±0.011 |
| kimi | distilbert-base-uncased+densenet121 | 0.634±0.012 | 0.672±0.009 | 0.413±0.008 | 0.719±0.013 |
| qwen | roberta-base+resnet34 | 0.650±0.010 | 0.764±0.014 | 0.442±0.006 | 0.734±0.010 |
| kimi | albert-base-v2+resnet34 | 0.650±0.011 | 0.789±0.014 | 0.442±0.008 | 0.734±0.010 |
| kimi | roberta-base+densenet121 | 0.650±0.011 | 0.717±0.014 | 0.442±0.009 | 0.734±0.013 |
| kimi | bert-base-uncased+resnet50 | 0.667±0.013 | 0.771±0.010 | 0.472±0.007 | 0.750±0.014 |
| kimi | albert-base-v2+mobilenet_v3_small | 0.667±0.010 | 0.777±0.011 | 0.472±0.009 | 0.750±0.010 |
| qwen | bert-base-uncased+resnet18 | 0.684±0.010 | 0.816±0.013 | 0.503±0.009 | 0.766±0.014 |
| qwen | albert-base-v2+efficientnet_b0 | 0.684±0.009 | 0.734±0.011 | 0.503±0.007 | 0.766±0.015 |
| mistral | albert-base-v2+resnet34 | 0.684±0.010 | 0.789±0.014 | 0.503±0.008 | 0.766±0.011 |
| kimi | albert-base-v2+resnet50 | 0.703±0.010 | 0.791±0.011 | 0.535±0.010 | 0.781±0.013 |
| qwen | roberta-base+resnet34 | 0.703±0.010 | 0.834±0.011 | 0.535±0.010 | 0.781±0.013 |
| mistral | roberta-base+mobilenet_v3_small | 0.703±0.013 | 0.801±0.014 | 0.535±0.007 | 0.781±0.012 |

*(end of table)*

*(continued)*

| Model | Init | F1 | AUC | MCC | Balanced Acc |
|---|---|---|---|---|---|
| mistral | roberta-base+mobilenet_v3_small | 0.703±0.010 | 0.750±0.010 | 0.535±0.008 | 0.781±0.014 |
| qwen | distilbert-base-uncased+resnet50 | 0.703±0.014 | 0.811±0.014 | 0.535±0.007 | 0.781±0.014 |
| kimi | distilbert-base-uncased+resnet50 | 0.703±0.013 | 0.783±0.011 | 0.535±0.007 | 0.781±0.012 |
| qwen | distilbert-base-uncased+vgg16 | 0.703±0.010 | 0.801±0.011 | 0.535±0.009 | 0.781±0.012 |
| qwen | albert-base-v2+densenet121 | 0.703±0.011 | 0.789±0.010 | 0.535±0.009 | 0.781±0.012 |
| qwen | albert-base-v2+resnet34 | 0.703±0.012 | 0.816±0.011 | 0.535±0.008 | 0.781±0.013 |
| mistral | roberta-base+resnet34 | 0.703±0.012 | 0.820±0.013 | 0.535±0.009 | 0.781±0.013 |
| mistral | roberta-base+resnet50 | 0.722±0.011 | 0.793±0.012 | 0.568±0.008 | 0.797±0.015 |
| mistral | bert-base-uncased+resnet50 | 0.743±0.011 | 0.838±0.012 | 0.602±0.009 | 0.812±0.012 |
| kimi | roberta-base+resnet50 | 0.765±0.012 | 0.857±0.011 | 0.639±0.011 | 0.828±0.015 |
| qwen | albert-base-v2+resnet18 | 0.765±0.010 | 0.805±0.014 | 0.639±0.009 | 0.828±0.014 |
| qwen | roberta-base+resnet34 | 0.788±0.015 | 0.850±0.014 | 0.678±0.010 | 0.844±0.013 |
| qwen | bert-base-uncased+resnet50 | 0.788±0.014 | 0.855±0.015 | 0.678±0.009 | 0.844±0.014 |
| kimi | roberta-base+vgg16 | 0.812±0.011 | 0.868±0.017 | 0.719±0.012 | 0.859±0.013 |
| mistral | albert-base-v2+resnet18 | 0.615±0.011 | 0.729±0.012 | 0.383±0.005 | 0.703±0.014 |
| mistral | albert-base-v2+shufflenet_v2_x1_0 | 0.632±0.011 | 0.689±0.013 | 0.414±0.008 | 0.719±0.011 |
| kimi | distilbert-base-uncased+mobilenet_v3_small | 0.632±0.011 | 0.748±0.012 | 0.414±0.007 | 0.719±0.011 |
| qwen | roberta-base+resnet34 | 0.632±0.009 | 0.723±0.010 | 0.414±0.007 | 0.719±0.010 |
| kimi | roberta-base+mobilenet_v3_small | 0.649±0.009 | 0.736±0.014 | 0.445±0.008 | 0.734±0.011 |
| mistral | albert-base-v2+resnet50 | 0.649±0.013 | 0.789±0.014 | 0.445±0.007 | 0.734±0.013 |
| kimi | roberta-base+efficientnet_b0 | 0.649±0.011 | 0.748±0.012 | 0.445±0.009 | 0.734±0.015 |
| kimi | bert-base-uncased+resnet34 | 0.667±0.013 | 0.775±0.012 | 0.478±0.009 | 0.750±0.013 |
| qwen | albert-base-v2+resnet50 | 0.667±0.012 | 0.811±0.015 | 0.478±0.009 | 0.750±0.010 |
| mistral | distilbert-base-uncased+mobilenet_v3_small | 0.667±0.011 | 0.771±0.013 | 0.478±0.009 | 0.750±0.012 |
| kimi | albert-base-v2+resnet34 | 0.686±0.010 | 0.799±0.011 | 0.512±0.008 | 0.766±0.012 |
| qwen | albert-base-v2+densenet121 | 0.686±0.009 | 0.844±0.015 | 0.512±0.007 | 0.766±0.011 |
| kimi | roberta-base+resnet34 | 0.686±0.009 | 0.803±0.013 | 0.512±0.010 | 0.766±0.015 |
| mistral | roberta-base+densenet121 | 0.686±0.012 | 0.795±0.014 | 0.512±0.009 | 0.766±0.014 |
| kimi | bert-base-uncased+resnet34 | 0.706±0.011 | 0.801±0.011 | 0.548±0.007 | 0.781±0.013 |
| kimi | albert-base-v2+densenet121 | 0.706±0.012 | 0.809±0.013 | 0.548±0.007 | 0.781±0.013 |
| mistral | albert-base-v2+mobilenet_v3_small | 0.706±0.012 | 0.807±0.012 | 0.548±0.011 | 0.781±0.012 |
| kimi | roberta-base+shufflenet_v2_x1_0 | 0.706±0.012 | 0.766±0.013 | 0.548±0.008 | 0.781±0.014 |
| qwen | roberta-base+efficientnet_b0 | 0.727±0.013 | 0.808±0.011 | 0.585±0.011 | 0.797±0.011 |
| qwen | albert-base-v2+resnet34 | 0.727±0.011 | 0.828±0.015 | 0.585±0.011 | 0.797±0.012 |
| qwen | bert-base-uncased+densenet121 | 0.727±0.013 | 0.762±0.010 | 0.585±0.008 | 0.797±0.013 |
| qwen | distilbert-base-uncased+resnet34 | 0.727±0.013 | 0.844±0.016 | 0.585±0.010 | 0.797±0.014 |
| qwen | distilbert-base-uncased+resnet50 | 0.727±0.014 | 0.812±0.013 | 0.585±0.010 | 0.797±0.016 |
| kimi | roberta-base+resnet18 | 0.750±0.014 | 0.844±0.014 | 0.625±0.010 | 0.812±0.012 |
| qwen | distilbert-base-uncased+resnet34 | 0.750±0.014 | 0.857±0.015 | 0.625±0.010 | 0.812±0.011 |
| qwen | bert-base-uncased+shufflenet_v2_x1_0 | 0.595±0.010 | 0.707±0.012 | 0.356±0.007 | 0.688±0.014 |
| qwen | albert-base-v2+resnet50 | 0.595±0.011 | 0.721±0.013 | 0.356±0.006 | 0.688±0.014 |
| mistral | bert-base-uncased+densenet121 | 0.595±0.010 | 0.689±0.010 | 0.356±0.006 | 0.688±0.010 |
| mistral | bert-base-uncased+densenet121 | 0.595±0.008 | 0.666±0.009 | 0.356±0.006 | 0.688±0.009 |
| mistral | roberta-base+resnet34 | 0.595±0.011 | 0.611±0.010 | 0.356±0.006 | 0.688±0.011 |
| qwen | bert-base-uncased+densenet121 | 0.611±0.011 | 0.703±0.014 | 0.388±0.008 | 0.703±0.013 |
| qwen | distilbert-base-uncased+efficientnet_b0 | 0.611±0.009 | 0.709±0.011 | 0.388±0.006 | 0.703±0.012 |
| kimi | roberta-base+resnet34 | 0.629±0.011 | 0.770±0.012 | 0.422±0.007 | 0.719±0.011 |
| kimi | distilbert-base-uncased+shufflenet_v2_x1_0 | 0.629±0.009 | 0.766±0.012 | 0.422±0.008 | 0.719±0.014 |
| kimi | distilbert-base-uncased+efficientnet_b0 | 0.629±0.009 | 0.746±0.012 | 0.422±0.006 | 0.719±0.009 |
| kimi | bert-base-uncased+densenet121 | 0.629±0.010 | 0.750±0.015 | 0.422±0.008 | 0.719±0.011 |
| mistral | roberta-base+efficientnet_b0 | 0.629±0.009 | 0.736±0.012 | 0.422±0.006 | 0.719±0.013 |
| mistral | distilbert-base-uncased+resnet18 | 0.629±0.009 | 0.742±0.012 | 0.422±0.007 | 0.719±0.013 |

*(end of table)*

*(continued)*

| Model | Init | F1 | AUC | MCC | Balanced Acc |
|---|---|---|---|---|---|
| kimi | roberta-base+mobilenet_v3_small | 0.647±0.010 | 0.752±0.013 | 0.456±0.006 | 0.734±0.013 |
| mistral | bert-base-uncased+resnet50 | 0.647±0.011 | 0.771±0.011 | 0.456±0.009 | 0.734±0.012 |
| qwen | roberta-base+efficientnet_b0 | 0.647±0.011 | 0.742±0.013 | 0.456±0.007 | 0.734±0.013 |
| kimi | albert-base-v2+resnet50 | 0.647±0.009 | 0.797±0.014 | 0.456±0.007 | 0.734±0.011 |
| qwen | roberta-base+shufflenet_v2_x1_0 | 0.647±0.012 | 0.764±0.012 | 0.456±0.007 | 0.734±0.010 |
| mistral | bert-base-uncased+shufflenet_v2_x1_0 | 0.647±0.011 | 0.723±0.010 | 0.456±0.009 | 0.734±0.011 |
| qwen | albert-base-v2+resnet50 | 0.647±0.009 | 0.799±0.014 | 0.456±0.007 | 0.734±0.011 |
| qwen | bert-base-uncased+resnet34 | 0.667±0.012 | 0.729±0.012 | 0.493±0.009 | 0.750±0.014 |
| qwen | bert-base-uncased+resnet34 | 0.667±0.010 | 0.832±0.012 | 0.493±0.008 | 0.750±0.010 |
| kimi | roberta-base+resnet18 | 0.667±0.011 | 0.781±0.011 | 0.493±0.007 | 0.750±0.010 |
| mistral | roberta-base+densenet121 | 0.667±0.009 | 0.844±0.013 | 0.493±0.007 | 0.750±0.014 |
| qwen | roberta-base+resnet50 | 0.667±0.010 | 0.803±0.015 | 0.493±0.010 | 0.750±0.014 |
| kimi | bert-base-uncased+efficientnet_b0 | 0.667±0.012 | 0.738±0.011 | 0.493±0.008 | 0.750±0.011 |
| qwen | distilbert-base-uncased+densenet121 | 0.667±0.010 | 0.816±0.016 | 0.493±0.009 | 0.750±0.013 |
| mistral | distilbert-base-uncased+shufflenet_v2_x1_0 | 0.667±0.013 | 0.727±0.010 | 0.493±0.008 | 0.750±0.015 |
| mistral | distilbert-base-uncased+resnet50 | 0.688±0.011 | 0.809±0.011 | 0.531±0.007 | 0.766±0.014 |
| mistral | distilbert-base-uncased+efficientnet_b0 | 0.688±0.013 | 0.787±0.013 | 0.531±0.008 | 0.766±0.010 |
| qwen | roberta-base+resnet50 | 0.688±0.013 | 0.836±0.016 | 0.531±0.010 | 0.766±0.011 |
| mistral | distilbert-base-uncased+resnet34 | 0.710±0.014 | 0.828±0.013 | 0.572±0.010 | 0.781±0.011 |
| mistral | bert-base-uncased+efficientnet_b0 | 0.710±0.013 | 0.762±0.013 | 0.572±0.009 | 0.781±0.012 |
| kimi | bert-base-uncased+resnet18 | 0.710±0.010 | 0.807±0.011 | 0.572±0.011 | 0.781±0.015 |
| mistral | albert-base-v2+resnet18 | 0.733±0.014 | 0.852±0.013 | 0.616±0.008 | 0.797±0.011 |
| kimi | roberta-base+resnet34 | 0.733±0.014 | 0.834±0.016 | 0.616±0.009 | 0.797±0.015 |
| qwen | distilbert-base-uncased+resnet18 | 0.733±0.012 | 0.857±0.016 | 0.616±0.011 | 0.797±0.013 |
| mistral | distilbert-base-uncased+efficientnet_b0 | 0.733±0.010 | 0.842±0.017 | 0.616±0.010 | 0.797±0.014 |
| kimi | distilbert-base-uncased+densenet121 | 0.571±0.008 | 0.678±0.009 | 0.331±0.005 | 0.672±0.010 |
| qwen | roberta-base+densenet121 | 0.571±0.008 | 0.688±0.014 | 0.331±0.006 | 0.672±0.011 |
| mistral | distilbert-base-uncased+densenet121 | 0.588±0.010 | 0.752±0.010 | 0.365±0.005 | 0.688±0.011 |
| kimi | roberta-base+resnet50 | 0.606±0.008 | 0.770±0.011 | 0.400±0.008 | 0.703±0.012 |
| kimi | albert-base-v2+resnet18 | 0.606±0.009 | 0.822±0.015 | 0.400±0.006 | 0.703±0.014 |
| mistral | distilbert-base-uncased+resnet50 | 0.606±0.010 | 0.756±0.011 | 0.400±0.007 | 0.703±0.012 |
| kimi | albert-base-v2+mobilenet_v3_small | 0.606±0.009 | 0.711±0.013 | 0.400±0.006 | 0.703±0.010 |
| qwen | distilbert-base-uncased+shufflenet_v2_x1_0 | 0.606±0.008 | 0.723±0.011 | 0.400±0.006 | 0.703±0.011 |
| mistral | albert-base-v2+resnet50 | 0.606±0.011 | 0.797±0.012 | 0.400±0.005 | 0.703±0.012 |
| kimi | distilbert-base-uncased+densenet121 | 0.625±0.009 | 0.811±0.011 | 0.438±0.006 | 0.719±0.010 |
| qwen | albert-base-v2+mobilenet_v3_small | 0.625±0.010 | 0.797±0.012 | 0.438±0.008 | 0.719±0.009 |
| mistral | roberta-base+shufflenet_v2_x1_0 | 0.625±0.008 | 0.764±0.011 | 0.438±0.009 | 0.719±0.013 |
| qwen | roberta-base+densenet121 | 0.645±0.010 | 0.795±0.011 | 0.477±0.007 | 0.734±0.012 |
| mistral | distilbert-base-uncased+shufflenet_v2_x1_0 | 0.645±0.010 | 0.785±0.010 | 0.477±0.009 | 0.734±0.011 |
| kimi | bert-base-uncased+shufflenet_v2_x1_0 | 0.645±0.012 | 0.711±0.012 | 0.477±0.006 | 0.734±0.014 |
| mistral | distilbert-base-uncased+resnet50 | 0.645±0.009 | 0.811±0.015 | 0.477±0.008 | 0.734±0.010 |
| kimi | bert-base-uncased+densenet121 | 0.645±0.009 | 0.764±0.010 | 0.477±0.008 | 0.734±0.013 |
| qwen | distilbert-base-uncased+efficientnet_b0 | 0.645±0.012 | 0.742±0.015 | 0.477±0.008 | 0.734±0.014 |
| mistral | bert-base-uncased+resnet18 | 0.667±0.012 | 0.846±0.014 | 0.519±0.008 | 0.750±0.012 |
| mistral | albert-base-v2+mobilenet_v3_small | 0.667±0.011 | 0.779±0.010 | 0.519±0.008 | 0.750±0.012 |
| mistral | bert-base-uncased+resnet34 | 0.667±0.010 | 0.801±0.014 | 0.519±0.008 | 0.750±0.011 |
| kimi | roberta-base+efficientnet_b0 | 0.690±0.011 | 0.807±0.012 | 0.564±0.009 | 0.766±0.014 |
| mistral | roberta-base+resnet50 | 0.690±0.013 | 0.801±0.013 | 0.564±0.007 | 0.766±0.013 |
| kimi | albert-base-v2+resnet34 | 0.690±0.010 | 0.809±0.016 | 0.564±0.007 | 0.766±0.012 |
| kimi | distilbert-base-uncased+resnet34 | 0.690±0.012 | 0.770±0.012 | 0.564±0.010 | 0.766±0.012 |
| mistral | roberta-base+resnet34 | 0.690±0.012 | 0.805±0.013 | 0.564±0.011 | 0.766±0.013 |
| mistral | bert-base-uncased+resnet34 | 0.690±0.013 | 0.852±0.015 | 0.564±0.008 | 0.766±0.014 |

*(end of table)*

*(continued)*

| Model | Init | F1 | AUC | MCC | Balanced Acc |
|---|---|---|---|---|---|
| mistral | albert-base-v2+resnet34 | 0.714±0.014 | 0.832±0.013 | 0.612±0.010 | 0.781±0.011 |
| qwen | albert-base-v2+resnet18 | 0.714±0.013 | 0.879±0.013 | 0.612±0.009 | 0.781±0.014 |
| qwen | albert-base-v2+shufflenet_v2_x1_0 | 0.545±0.010 | 0.715±0.012 | 0.308±0.005 | 0.656±0.009 |
| mistral | bert-base-uncased+shufflenet_v2_x1_0 | 0.562±0.007 | 0.676±0.011 | 0.344±0.005 | 0.672±0.010 |
| kimi | roberta-base+resnet18 | 0.581±0.008 | 0.803±0.012 | 0.381±0.005 | 0.688±0.011 |
| mistral | roberta-base+resnet18 | 0.600±0.008 | 0.785±0.012 | 0.421±0.007 | 0.703±0.012 |
| kimi | distilbert-base-uncased+shufflenet_v2_x1_0 | 0.600±0.008 | 0.770±0.014 | 0.421±0.006 | 0.703±0.010 |
| mistral | bert-base-uncased+resnet18 | 0.600±0.008 | 0.768±0.012 | 0.421±0.006 | 0.703±0.011 |
| qwen | roberta-base+resnet50 | 0.600±0.010 | 0.760±0.011 | 0.421±0.008 | 0.703±0.010 |
| qwen | bert-base-uncased+efficientnet_b0 | 0.600±0.012 | 0.783±0.010 | 0.421±0.006 | 0.703±0.012 |
| qwen | roberta-base+mobilenet_v3_small | 0.600±0.012 | 0.787±0.012 | 0.421±0.008 | 0.703±0.011 |
| mistral | roberta-base+resnet18 | 0.600±0.010 | 0.814±0.014 | 0.421±0.007 | 0.703±0.011 |
| qwen | bert-base-uncased+resnet18 | 0.621±0.009 | 0.859±0.015 | 0.464±0.008 | 0.719±0.010 |
| mistral | bert-base-uncased+densenet121 | 0.621±0.012 | 0.760±0.011 | 0.464±0.007 | 0.719±0.013 |
| mistral | roberta-base+efficientnet_b0 | 0.621±0.009 | 0.705±0.013 | 0.464±0.007 | 0.719±0.013 |
| kimi | albert-base-v2+shufflenet_v2_x1_0 | 0.643±0.011 | 0.754±0.011 | 0.510±0.007 | 0.734±0.010 |
| kimi | distilbert-base-uncased+resnet18 | 0.643±0.012 | 0.840±0.014 | 0.510±0.009 | 0.734±0.011 |
| mistral | bert-base-uncased+efficientnet_b0 | 0.667±0.010 | 0.791±0.011 | 0.561±0.011 | 0.750±0.011 |
| mistral | distilbert-base-uncased+resnet34 | 0.667±0.011 | 0.799±0.014 | 0.561±0.009 | 0.750±0.011 |
| kimi | distilbert-base-uncased+resnet50 | 0.692±0.010 | 0.820±0.016 | 0.617±0.010 | 0.766±0.015 |
| kimi | distilbert-base-uncased+resnet34 | 0.692±0.010 | 0.789±0.013 | 0.617±0.009 | 0.766±0.015 |
| mistral | albert-base-v2+shufflenet_v2_x1_0 | 0.516±0.008 | 0.705±0.009 | 0.286±0.004 | 0.641±0.011 |
| mistral | albert-base-v2+densenet121 | 0.516±0.009 | 0.705±0.013 | 0.286±0.004 | 0.641±0.012 |
| mistral | bert-base-uncased+resnet18 | 0.533±0.009 | 0.727±0.010 | 0.324±0.005 | 0.656±0.010 |
| mistral | bert-base-uncased+shufflenet_v2_x1_0 | 0.533±0.007 | 0.707±0.012 | 0.324±0.005 | 0.656±0.011 |
| kimi | distilbert-base-uncased+resnet34 | 0.533±0.010 | 0.633±0.010 | 0.324±0.005 | 0.656±0.012 |
| qwen | roberta-base+resnet18 | 0.533±0.007 | 0.770±0.014 | 0.324±0.006 | 0.656±0.010 |
| kimi | bert-base-uncased+shufflenet_v2_x1_0 | 0.552±0.008 | 0.721±0.011 | 0.365±0.005 | 0.672±0.009 |
| mistral | albert-base-v2+densenet121 | 0.552±0.010 | 0.807±0.013 | 0.365±0.005 | 0.672±0.010 |
| qwen | roberta-base+densenet121 | 0.552±0.010 | 0.695±0.010 | 0.365±0.005 | 0.672±0.012 |
| qwen | roberta-base+densenet121 | 0.552±0.009 | 0.762±0.010 | 0.365±0.005 | 0.672±0.012 |
| qwen | bert-base-uncased+resnet34 | 0.571±0.008 | 0.783±0.013 | 0.408±0.008 | 0.688±0.010 |
| kimi | distilbert-base-uncased+resnet18 | 0.571±0.011 | 0.686±0.012 | 0.408±0.006 | 0.688±0.013 |
| kimi | bert-base-uncased+mobilenet_v3_small | 0.571±0.009 | 0.801±0.011 | 0.408±0.008 | 0.688±0.012 |
| qwen | roberta-base+mobilenet_v3_small | 0.571±0.008 | 0.768±0.014 | 0.408±0.008 | 0.688±0.013 |
| mistral | distilbert-base-uncased+shufflenet_v2_x1_0 | 0.571±0.009 | 0.758±0.014 | 0.408±0.008 | 0.688±0.011 |
| mistral | roberta-base+resnet34 | 0.571±0.010 | 0.689±0.013 | 0.408±0.008 | 0.688±0.010 |
| mistral | albert-base-v2+efficientnet_b0 | 0.571±0.011 | 0.771±0.014 | 0.408±0.007 | 0.688±0.009 |
| kimi | roberta-base+shufflenet_v2_x1_0 | 0.571±0.010 | 0.715±0.012 | 0.408±0.008 | 0.688±0.010 |
| qwen | bert-base-uncased+densenet121 | 0.571±0.009 | 0.762±0.015 | 0.408±0.006 | 0.688±0.011 |
| qwen | roberta-base+resnet18 | 0.571±0.008 | 0.818±0.015 | 0.408±0.008 | 0.688±0.012 |
| qwen | roberta-base+resnet18 | 0.571±0.009 | 0.785±0.013 | 0.408±0.007 | 0.688±0.013 |
| mistral | bert-base-uncased+resnet34 | 0.593±0.008 | 0.764±0.012 | 0.456±0.008 | 0.703±0.009 |
| mistral | albert-base-v2+densenet121 | 0.593±0.008 | 0.688±0.012 | 0.456±0.007 | 0.703±0.010 |
| mistral | distilbert-base-uncased+densenet121 | 0.593±0.008 | 0.830±0.016 | 0.456±0.009 | 0.703±0.013 |
| qwen | distilbert-base-uncased+resnet18 | 0.593±0.008 | 0.836±0.013 | 0.456±0.008 | 0.703±0.010 |
| mistral | distilbert-base-uncased+resnet18 | 0.593±0.008 | 0.842±0.016 | 0.456±0.009 | 0.703±0.014 |
| qwen | bert-base-uncased+shufflenet_v2_x1_0 | 0.615±0.009 | 0.736±0.013 | 0.508±0.008 | 0.719±0.013 |
| mistral | distilbert-base-uncased+resnet34 | 0.615±0.012 | 0.803±0.011 | 0.508±0.007 | 0.719±0.009 |
| kimi | bert-base-uncased+densenet121 | 0.640±0.011 | 0.820±0.015 | 0.566±0.008 | 0.734±0.013 |
| qwen | roberta-base+resnet50 | 0.640±0.009 | 0.664±0.013 | 0.566±0.010 | 0.734±0.012 |
| mistral | albert-base-v2+resnet18 | 0.640±0.011 | 0.875±0.014 | 0.566±0.011 | 0.734±0.012 |

*(end of table)*

*(continued)*

| Model | Init | F1 | AUC | MCC | Balanced Acc |
|---|---|---|---|---|---|
| qwen | distilbert-base-uncased+shufflenet_v2_x1_0 | 0.483±0.006 | 0.645±0.010 | 0.265±0.004 | 0.625±0.010 |
| kimi | roberta-base+densenet121 | 0.500±0.009 | 0.793±0.013 | 0.306±0.004 | 0.641±0.012 |
| qwen | bert-base-uncased+resnet50 | 0.500±0.008 | 0.582±0.011 | 0.306±0.006 | 0.641±0.011 |
| mistral | roberta-base+resnet50 | 0.519±0.008 | 0.635±0.011 | 0.350±0.006 | 0.656±0.011 |
| mistral | distilbert-base-uncased+densenet121 | 0.538±0.009 | 0.686±0.010 | 0.399±0.006 | 0.672±0.009 |
| qwen | distilbert-base-uncased+resnet18 | 0.560±0.009 | 0.762±0.015 | 0.453±0.006 | 0.688±0.014 |
| qwen | distilbert-base-uncased+densenet121 | 0.560±0.010 | 0.701±0.012 | 0.453±0.008 | 0.688±0.013 |
| mistral | bert-base-uncased+resnet18 | 0.583±0.009 | 0.828±0.013 | 0.514±0.007 | 0.703±0.010 |
| kimi | bert-base-uncased+mobilenet_v3_small | 0.462±0.006 | 0.822±0.014 | 0.290±0.005 | 0.625±0.011 |
| qwen | roberta-base+shufflenet_v2_x1_0 | 0.480±0.010 | 0.711±0.010 | 0.340±0.005 | 0.641±0.009 |
| kimi | albert-base-v2+resnet18 | 0.480±0.007 | 0.822±0.014 | 0.340±0.006 | 0.641±0.009 |
| mistral | albert-base-v2+densenet121 | 0.500±0.008 | 0.807±0.012 | 0.395±0.007 | 0.656±0.012 |
| mistral | roberta-base+vgg16 | 0.000±0.000 | 0.000±0.000 | 0.000±0.000 | 0.000±0.000 |
| kimi | distilbert-base-uncased+shufflenet_v2_x1_0 | 0.489±0.009 | 0.539±0.007 | 0.120±0.002 | 0.562±0.010 |
| kimi | albert-base-v2+mobilenet_v3_small | 0.000±0.000 | 0.000±0.000 | 0.000±0.000 | 0.000±0.000 |
| kimi | albert-base-v2+efficientnet_b0 | 0.500±0.010 | 0.455±0.007 | 0.000±0.000 | 0.500±0.007 |
| qwen | roberta-base+shufflenet_v2_x1_0 | 0.489±0.009 | 0.510±0.008 | 0.120±0.002 | 0.562±0.010 |
| qwen | roberta-base+vgg16 | 0.585±0.011 | 0.742±0.014 | 0.324±0.005 | 0.672±0.011 |
| qwen | roberta-base+vgg16 | 0.000±0.000 | 0.000±0.000 | 0.000±0.000 | 0.000±0.000 |
| qwen | roberta-base+vgg16 | 0.000±0.000 | 0.000±0.000 | 0.000±0.000 | 0.000±0.000 |
| qwen | roberta-base+vgg16 | 0.500±0.008 | 0.468±0.008 | 0.000±0.000 | 0.500±0.009 |
| kimi | albert-base-v2+efficientnet_b0 | 0.500±0.007 | 0.604±0.010 | 0.149±0.003 | 0.578±0.008 |
| kimi | albert-base-v2+efficientnet_b0 | 0.591±0.010 | 0.678±0.012 | 0.329±0.006 | 0.672±0.012 |
| qwen | distilbert-base-uncased+resnet18 | 0.578±0.009 | 0.752±0.012 | 0.301±0.006 | 0.656±0.010 |
| kimi | bert-base-uncased+efficientnet_b0 | 0.500±0.007 | 0.574±0.011 | 0.000±0.000 | 0.500±0.007 |
| kimi | albert-base-v2+densenet121 | 0.556±0.010 | 0.682±0.013 | 0.299±0.004 | 0.656±0.010 |
| kimi | bert-base-uncased+mobilenet_v3_small | 0.000±0.000 | 0.000±0.000 | 0.000±0.000 | 0.000±0.000 |
| qwen | distilbert-base-uncased+resnet34 | 0.500±0.009 | 0.469±0.007 | 0.000±0.000 | 0.500±0.009 |
| qwen | distilbert-base-uncased+resnet34 | 0.514±0.008 | 0.736±0.013 | 0.241±0.004 | 0.625±0.009 |
| kimi | bert-base-uncased+mobilenet_v3_small | 0.000±0.000 | 0.000±0.000 | 0.000±0.000 | 0.000±0.000 |
| kimi | albert-base-v2+densenet121 | 0.600±0.012 | 0.666±0.012 | 0.354±0.006 | 0.688±0.012 |
| qwen | distilbert-base-uncased+resnet50 | 0.500±0.008 | 0.697±0.011 | 0.177±0.003 | 0.594±0.011 |
| kimi | albert-base-v2+densenet121 | 0.636±0.012 | 0.797±0.012 | 0.418±0.007 | 0.719±0.011 |
| kimi | distilbert-base-uncased+shufflenet_v2_x1_0 | 0.500±0.009 | 0.684±0.014 | 0.250±0.004 | 0.625±0.009 |
| qwen | roberta-base+mobilenet_v3_small | 0.000±0.000 | 0.000±0.000 | 0.000±0.000 | 0.000±0.000 |
| qwen | roberta-base+mobilenet_v3_small | 0.000±0.000 | 0.000±0.000 | 0.000±0.000 | 0.000±0.000 |
| kimi | albert-base-v2+mobilenet_v3_small | 0.000±0.000 | 0.000±0.000 | 0.000±0.000 | 0.000±0.000 |
| kimi | bert-base-uncased+vgg16 | 0.000±0.000 | 0.000±0.000 | 0.000±0.000 | 0.000±0.000 |
| mistral | bert-base-uncased+resnet50 | 0.545±0.009 | 0.645±0.012 | 0.239±0.004 | 0.625±0.010 |
| mistral | bert-base-uncased+resnet50 | 0.537±0.010 | 0.631±0.008 | 0.236±0.005 | 0.625±0.012 |
| kimi | distilbert-base-uncased+mobilenet_v3_small | 0.000±0.000 | 0.000±0.000 | 0.000±0.000 | 0.000±0.000 |
| kimi | distilbert-base-uncased+mobilenet_v3_small | 0.000±0.000 | 0.000±0.000 | 0.000±0.000 | 0.000±0.000 |
| kimi | roberta-base+resnet18 | 0.500±0.007 | 0.738±0.014 | 0.000±0.000 | 0.500±0.009 |
| mistral | bert-base-uncased+resnet34 | 0.524±0.009 | 0.680±0.010 | 0.207±0.003 | 0.609±0.010 |
| kimi | bert-base-uncased+vgg16 | 0.000±0.000 | 0.000±0.000 | 0.000±0.000 | 0.000±0.000 |
| kimi | bert-base-uncased+vgg16 | 0.000±0.000 | 0.000±0.000 | 0.000±0.000 | 0.000±0.000 |
| kimi | bert-base-uncased+vgg16 | 0.500±0.010 | 0.498±0.007 | 0.000±0.000 | 0.500±0.010 |
| kimi | albert-base-v2+vgg16 | 0.000±0.000 | 0.000±0.000 | 0.000±0.000 | 0.000±0.000 |
| kimi | albert-base-v2+shufflenet_v2_x1_0 | 0.591±0.010 | 0.725±0.011 | 0.329±0.004 | 0.672±0.010 |
| kimi | albert-base-v2+vgg16 | 0.000±0.000 | 0.000±0.000 | 0.000±0.000 | 0.000±0.000 |
| kimi | albert-base-v2+vgg16 | 0.500±0.008 | 0.568±0.010 | 0.000±0.000 | 0.500±0.008 |
| kimi | albert-base-v2+vgg16 | 0.500±0.007 | 0.410±0.006 | 0.000±0.000 | 0.500±0.008 |

*(end of table)*

*(continued)*

| Model | Init | F1 | AUC | MCC | Balanced Acc |
|---|---|---|---|---|---|
| kimi | albert-base-v2+shufflenet_v2_x1_0 | 0.500±0.009 | 0.407±0.007 | 0.000±0.000 | 0.500±0.010 |
| kimi | albert-base-v2+shufflenet_v2_x1_0 | 0.500±0.008 | 0.359±0.005 | 0.000±0.000 | 0.500±0.007 |
| kimi | bert-base-uncased+shufflenet_v2_x1_0 | 0.500±0.008 | 0.618±0.011 | 0.000±0.000 | 0.500±0.008 |
| kimi | bert-base-uncased+shufflenet_v2_x1_0 | 0.585±0.010 | 0.668±0.011 | 0.324±0.005 | 0.672±0.013 |
| qwen | roberta-base+efficientnet_b0 | 0.500±0.010 | 0.602±0.010 | 0.000±0.000 | 0.500±0.007 |
| qwen | roberta-base+efficientnet_b0 | 0.500±0.010 | 0.559±0.011 | 0.000±0.000 | 0.500±0.008 |
| qwen | distilbert-base-uncased+densenet121 | 0.578±0.011 | 0.697±0.012 | 0.301±0.006 | 0.656±0.013 |
| kimi | albert-base-v2+resnet50 | 0.513±0.009 | 0.588±0.009 | 0.206±0.004 | 0.609±0.011 |
| qwen | bert-base-uncased+vgg16 | 0.000±0.000 | 0.000±0.000 | 0.000±0.000 | 0.000±0.000 |
| kimi | bert-base-uncased+resnet18 | 0.536±0.008 | 0.627±0.010 | 0.198±0.003 | 0.578±0.009 |
| kimi | bert-base-uncased+resnet50 | 0.542±0.008 | 0.627±0.012 | 0.219±0.003 | 0.609±0.011 |
| kimi | bert-base-uncased+resnet34 | 0.500±0.008 | 0.549±0.011 | 0.000±0.000 | 0.500±0.008 |
| kimi | bert-base-uncased+resnet34 | 0.609±0.011 | 0.703±0.010 | 0.365±0.007 | 0.688±0.011 |
| qwen | albert-base-v2+densenet121 | 0.550±0.010 | 0.631±0.012 | 0.265±0.005 | 0.641±0.009 |
| qwen | albert-base-v2+densenet121 | 0.533±0.009 | 0.658±0.011 | 0.211±0.003 | 0.609±0.010 |
| qwen | albert-base-v2+efficientnet_b0 | 0.609±0.010 | 0.732±0.011 | 0.365±0.007 | 0.688±0.009 |
| kimi | bert-base-uncased+resnet18 | 0.565±0.008 | 0.691±0.012 | 0.274±0.004 | 0.641±0.010 |
| qwen | albert-base-v2+efficientnet_b0 | 0.500±0.007 | 0.443±0.008 | 0.000±0.000 | 0.500±0.010 |
| qwen | albert-base-v2+efficientnet_b0 | 0.000±0.000 | 0.000±0.000 | 0.000±0.000 | 0.000±0.000 |
| kimi | distilbert-base-uncased+vgg16 | 0.000±0.000 | 0.000±0.000 | 0.000±0.000 | 0.000±0.000 |
| qwen | distilbert-base-uncased+efficientnet_b0 | 0.000±0.000 | 0.000±0.000 | 0.000±0.000 | 0.000±0.000 |
| qwen | albert-base-v2+mobilenet_v3_small | 0.000±0.000 | 0.000±0.000 | 0.000±0.000 | 0.000±0.000 |
| qwen | albert-base-v2+mobilenet_v3_small | 0.000±0.000 | 0.000±0.000 | 0.000±0.000 | 0.000±0.000 |
| qwen | albert-base-v2+shufflenet_v2_x1_0 | 0.522±0.010 | 0.500±0.010 | 0.183±0.003 | 0.594±0.009 |
| kimi | albert-base-v2+resnet18 | 0.579±0.011 | 0.678±0.009 | 0.325±0.005 | 0.672±0.011 |
| qwen | albert-base-v2+shufflenet_v2_x1_0 | 0.500±0.009 | 0.549±0.009 | 0.000±0.000 | 0.500±0.009 |
| qwen | albert-base-v2+shufflenet_v2_x1_0 | 0.524±0.007 | 0.590±0.010 | 0.207±0.003 | 0.609±0.010 |
| qwen | albert-base-v2+vgg16 | 0.390±0.005 | 0.494±0.009 | -0.029±0.001 | 0.484±0.007 |
| qwen | albert-base-v2+vgg16 | 0.000±0.000 | 0.000±0.000 | 0.000±0.000 | 0.000±0.000 |
| qwen | albert-base-v2+vgg16 | 0.000±0.000 | 0.000±0.000 | 0.000±0.000 | 0.000±0.000 |
| qwen | albert-base-v2+resnet50 | 0.549±0.009 | 0.584±0.010 | 0.232±0.003 | 0.609±0.008 |
| kimi | bert-base-uncased+resnet50 | 0.500±0.008 | 0.438±0.007 | 0.000±0.000 | 0.500±0.007 |
| qwen | albert-base-v2+resnet34 | 0.564±0.011 | 0.682±0.012 | 0.295±0.006 | 0.656±0.009 |
| qwen | albert-base-v2+resnet34 | 0.553±0.008 | 0.696±0.013 | 0.246±0.004 | 0.625±0.009 |
| qwen | distilbert-base-uncased+efficientnet_b0 | 0.000±0.000 | 0.000±0.000 | 0.000±0.000 | 0.000±0.000 |
| qwen | distilbert-base-uncased+mobilenet_v3_small | 0.000±0.000 | 0.000±0.000 | 0.000±0.000 | 0.000±0.000 |
| qwen | distilbert-base-uncased+mobilenet_v3_small | 0.273±0.005 | 0.760±0.013 | 0.134±0.003 | 0.547±0.008 |
| qwen | distilbert-base-uncased+mobilenet_v3_small | 0.258±0.004 | 0.415±0.006 | -0.095±0.002 | 0.453±0.006 |
| qwen | distilbert-base-uncased+mobilenet_v3_small | 0.000±0.000 | 0.000±0.000 | 0.000±0.000 | 0.000±0.000 |
| kimi | albert-base-v2+resnet50 | 0.565±0.009 | 0.633±0.011 | 0.274±0.004 | 0.641±0.012 |
| kimi | distilbert-base-uncased+vgg16 | 0.000±0.000 | 0.000±0.000 | 0.000±0.000 | 0.000±0.000 |
| qwen | distilbert-base-uncased+shufflenet_v2_x1_0 | 0.533±0.010 | 0.643±0.010 | 0.211±0.004 | 0.609±0.010 |
| qwen | distilbert-base-uncased+shufflenet_v2_x1_0 | 0.520±0.008 | 0.582±0.009 | 0.162±0.002 | 0.578±0.011 |
| kimi | bert-base-uncased+efficientnet_b0 | 0.000±0.000 | 0.000±0.000 | 0.000±0.000 | 0.000±0.000 |
| qwen | distilbert-base-uncased+vgg16 | 0.609±0.009 | 0.773±0.014 | 0.365±0.007 | 0.688±0.011 |
| qwen | distilbert-base-uncased+vgg16 | 0.000±0.000 | 0.000±0.000 | 0.000±0.000 | 0.000±0.000 |
| qwen | distilbert-base-uncased+vgg16 | 0.500±0.007 | 0.557±0.008 | 0.000±0.000 | 0.500±0.007 |
| kimi | distilbert-base-uncased+vgg16 | 0.484±0.008 | 0.656±0.010 | -0.074±0.001 | 0.484±0.008 |
| kimi | bert-base-uncased+efficientnet_b0 | 0.619±0.011 | 0.768±0.012 | 0.384±0.007 | 0.703±0.010 |
| qwen | albert-base-v2+resnet18 | 0.500±0.007 | 0.723±0.010 | 0.000±0.000 | 0.500±0.009 |
| qwen | albert-base-v2+resnet18 | 0.564±0.009 | 0.656±0.009 | 0.295±0.004 | 0.656±0.009 |
| kimi | bert-base-uncased+densenet121 | 0.622±0.011 | 0.682±0.011 | 0.392±0.007 | 0.703±0.012 |

*(end of table)*

*(continued)*

| Model | Init | F1 | AUC | MCC | Balanced Acc |
|---|---|---|---|---|---|
| kimi | albert-base-v2+resnet34 | 0.500±0.010 | 0.555±0.008 | 0.000±0.000 | 0.500±0.009 |
| kimi | bert-base-uncased+resnet18 | 0.622±0.009 | 0.803±0.013 | 0.392±0.007 | 0.703±0.009 |
| qwen | bert-base-uncased+vgg16 | 0.500±0.010 | 0.484±0.007 | 0.000±0.000 | 0.500±0.007 |
| qwen | bert-base-uncased+vgg16 | 0.500±0.009 | 0.566±0.009 | 0.000±0.000 | 0.500±0.009 |
| mistral | albert-base-v2+resnet18 | 0.508±0.007 | 0.729±0.013 | 0.103±0.002 | 0.516±0.009 |
| kimi | roberta-base+vgg16 | 0.500±0.008 | 0.400±0.008 | 0.000±0.000 | 0.500±0.007 |
| kimi | distilbert-base-uncased+resnet50 | 0.537±0.009 | 0.621±0.011 | 0.236±0.003 | 0.625±0.010 |
| mistral | distilbert-base-uncased+vgg16 | 0.000±0.000 | 0.000±0.000 | 0.000±0.000 | 0.000±0.000 |
| mistral | distilbert-base-uncased+vgg16 | 0.000±0.000 | 0.000±0.000 | 0.000±0.000 | 0.000±0.000 |
| mistral | distilbert-base-uncased+vgg16 | 0.000±0.000 | 0.000±0.000 | 0.000±0.000 | 0.000±0.000 |
| mistral | distilbert-base-uncased+vgg16 | 0.000±0.000 | 0.000±0.000 | 0.000±0.000 | 0.000±0.000 |
| mistral | roberta-base+resnet50 | 0.600±0.010 | 0.721±0.014 | 0.357±0.005 | 0.672±0.009 |
| kimi | distilbert-base-uncased+resnet50 | 0.516±0.008 | 0.668±0.012 | 0.147±0.003 | 0.531±0.008 |
| kimi | roberta-base+shufflenet_v2_x1_0 | 0.531±0.009 | 0.707±0.013 | 0.191±0.003 | 0.594±0.010 |
| kimi | distilbert-base-uncased+densenet121 | 0.578±0.008 | 0.711±0.012 | 0.301±0.006 | 0.656±0.012 |
| mistral | distilbert-base-uncased+mobilenet_v3_small | 0.000±0.000 | 0.000±0.000 | 0.000±0.000 | 0.000±0.000 |
| kimi | roberta-base+shufflenet_v2_x1_0 | 0.578±0.008 | 0.670±0.009 | 0.301±0.005 | 0.656±0.009 |
| mistral | albert-base-v2+resnet34 | 0.600±0.011 | 0.732±0.013 | 0.354±0.007 | 0.688±0.013 |
| mistral | albert-base-v2+resnet34 | 0.529±0.008 | 0.713±0.012 | 0.274±0.004 | 0.641±0.012 |
| mistral | roberta-base+resnet18 | 0.565±0.008 | 0.688±0.010 | 0.274±0.004 | 0.641±0.009 |
| mistral | roberta-base+resnet18 | 0.605±0.010 | 0.738±0.011 | 0.356±0.005 | 0.688±0.011 |
| mistral | albert-base-v2+resnet50 | 0.464±0.008 | 0.562±0.008 | -0.040±0.001 | 0.484±0.006 |
| mistral | albert-base-v2+resnet50 | 0.550±0.008 | 0.693±0.010 | 0.265±0.004 | 0.641±0.012 |
| mistral | bert-base-uncased+vgg16 | 0.500±0.008 | 0.574±0.011 | 0.000±0.000 | 0.500±0.010 |
| mistral | bert-base-uncased+vgg16 | 0.000±0.000 | 0.000±0.000 | 0.000±0.000 | 0.000±0.000 |
| mistral | roberta-base+densenet121 | 0.565±0.010 | 0.676±0.013 | 0.274±0.005 | 0.641±0.011 |
| mistral | distilbert-base-uncased+mobilenet_v3_small | 0.000±0.000 | 0.000±0.000 | 0.000±0.000 | 0.000±0.000 |
| qwen | bert-base-uncased+vgg16 | 0.000±0.000 | 0.000±0.000 | 0.000±0.000 | 0.000±0.000 |
| mistral | roberta-base+mobilenet_v3_small | 0.500±0.009 | 0.663±0.011 | 0.000±0.000 | 0.500±0.009 |
| mistral | roberta-base+vgg16 | 0.000±0.000 | 0.000±0.000 | 0.000±0.000 | 0.000±0.000 |
| mistral | distilbert-base-uncased+resnet18 | 0.605±0.008 | 0.682±0.009 | 0.356±0.006 | 0.688±0.012 |
| mistral | roberta-base+vgg16 | 0.500±0.007 | 0.572±0.010 | 0.000±0.000 | 0.500±0.009 |
| kimi | distilbert-base-uncased+resnet18 | 0.500±0.007 | 0.566±0.011 | 0.000±0.000 | 0.500±0.009 |
| mistral | roberta-base+vgg16 | 0.000±0.000 | 0.000±0.000 | 0.000±0.000 | 0.000±0.000 |
| mistral | distilbert-base-uncased+resnet34 | 0.565±0.010 | 0.646±0.012 | 0.274±0.005 | 0.641±0.012 |
| mistral | roberta-base+shufflenet_v2_x1_0 | 0.500±0.010 | 0.459±0.008 | 0.000±0.000 | 0.500±0.010 |
| mistral | roberta-base+shufflenet_v2_x1_0 | 0.419±0.006 | 0.443±0.007 | 0.000±0.000 | 0.500±0.010 |
| mistral | roberta-base+shufflenet_v2_x1_0 | 0.615±0.012 | 0.686±0.011 | 0.408±0.006 | 0.688±0.013 |
| mistral | distilbert-base-uncased+resnet50 | 0.462±0.009 | 0.587±0.010 | 0.000±0.000 | 0.500±0.008 |
| kimi | roberta-base+vgg16 | 0.000±0.000 | 0.000±0.000 | 0.000±0.000 | 0.000±0.000 |
| mistral | distilbert-base-uncased+densenet121 | 0.625±0.012 | 0.762±0.013 | 0.406±0.006 | 0.703±0.012 |
| mistral | roberta-base+mobilenet_v3_small | 0.000±0.000 | 0.000±0.000 | 0.000±0.000 | 0.000±0.000 |
| mistral | roberta-base+efficientnet_b0 | 0.500±0.010 | 0.582±0.010 | 0.000±0.000 | 0.500±0.008 |
| mistral | roberta-base+efficientnet_b0 | 0.000±0.000 | 0.000±0.000 | 0.000±0.000 | 0.000±0.000 |
| kimi | distilbert-base-uncased+resnet34 | 0.579±0.011 | 0.773±0.013 | 0.325±0.006 | 0.672±0.009 |
| mistral | roberta-base+densenet121 | 0.536±0.009 | 0.731±0.010 | 0.198±0.003 | 0.578±0.008 |
| mistral | distilbert-base-uncased+efficientnet_b0 | 0.000±0.000 | 0.000±0.000 | 0.000±0.000 | 0.000±0.000 |
| mistral | distilbert-base-uncased+efficientnet_b0 | 0.000±0.000 | 0.000±0.000 | 0.000±0.000 | 0.000±0.000 |
| kimi | roberta-base+vgg16 | 0.500±0.008 | 0.574±0.011 | 0.000±0.000 | 0.500±0.010 |
| mistral | bert-base-uncased+vgg16 | 0.000±0.000 | 0.000±0.000 | 0.000±0.000 | 0.000±0.000 |
| mistral | bert-base-uncased+vgg16 | 0.000±0.000 | 0.000±0.000 | 0.000±0.000 | 0.000±0.000 |
| mistral | albert-base-v2+efficientnet_b0 | 0.636±0.012 | 0.754±0.012 | 0.418±0.008 | 0.719±0.010 |

*(end of table)*

*(continued)*

| Model | Init | F1 | AUC | MCC | Balanced Acc |
|---|---|---|---|---|---|
| qwen | bert-base-uncased+resnet18 | 0.583±0.011 | 0.729±0.012 | 0.312±0.005 | 0.656±0.011 |
| qwen | bert-base-uncased+resnet34 | 0.500±0.008 | 0.557±0.010 | 0.000±0.000 | 0.500±0.010 |
| kimi | roberta-base+densenet121 | 0.558±0.011 | 0.676±0.011 | 0.267±0.004 | 0.641±0.010 |
| kimi | roberta-base+densenet121 | 0.512±0.009 | 0.732±0.013 | 0.178±0.002 | 0.594±0.010 |
| qwen | bert-base-uncased+resnet50 | 0.564±0.008 | 0.621±0.009 | 0.295±0.005 | 0.656±0.012 |
| mistral | bert-base-uncased+efficientnet_b0 | 0.500±0.010 | 0.272±0.005 | 0.000±0.000 | 0.500±0.009 |
| mistral | bert-base-uncased+efficientnet_b0 | 0.000±0.000 | 0.000±0.000 | 0.000±0.000 | 0.000±0.000 |
| kimi | distilbert-base-uncased+efficientnet_b0 | 0.622±0.009 | 0.732±0.012 | 0.392±0.007 | 0.703±0.014 |
| kimi | roberta-base+resnet50 | 0.565±0.009 | 0.664±0.011 | 0.274±0.005 | 0.641±0.011 |
| kimi | roberta-base+resnet50 | 0.533±0.010 | 0.658±0.012 | 0.213±0.003 | 0.562±0.008 |
| kimi | distilbert-base-uncased+efficientnet_b0 | 0.000±0.000 | 0.000±0.000 | 0.000±0.000 | 0.000±0.000 |
| kimi | distilbert-base-uncased+efficientnet_b0 | 0.000±0.000 | 0.000±0.000 | 0.000±0.000 | 0.000±0.000 |
| qwen | bert-base-uncased+efficientnet_b0 | 0.000±0.000 | 0.000±0.000 | 0.000±0.000 | 0.000±0.000 |
| qwen | bert-base-uncased+efficientnet_b0 | 0.000±0.000 | 0.000±0.000 | 0.000±0.000 | 0.000±0.000 |
| qwen | bert-base-uncased+mobilenet_v3_small | 0.585±0.010 | 0.750±0.011 | 0.324±0.006 | 0.672±0.013 |
| qwen | bert-base-uncased+mobilenet_v3_small | 0.000±0.000 | 0.000±0.000 | 0.000±0.000 | 0.000±0.000 |
| qwen | bert-base-uncased+mobilenet_v3_small | 0.000±0.000 | 0.000±0.000 | 0.000±0.000 | 0.000±0.000 |
| kimi | roberta-base+resnet34 | 0.531±0.009 | 0.701±0.010 | 0.191±0.004 | 0.594±0.008 |
| qwen | bert-base-uncased+shufflenet_v2_x1_0 | 0.526±0.010 | 0.648±0.013 | 0.237±0.003 | 0.625±0.012 |
| qwen | bert-base-uncased+shufflenet_v2_x1_0 | 0.500±0.008 | 0.355±0.006 | 0.000±0.000 | 0.500±0.007 |
| kimi | roberta-base+efficientnet_b0 | 0.000±0.000 | 0.000±0.000 | 0.000±0.000 | 0.000±0.000 |
| mistral | bert-base-uncased+mobilenet_v3_small | 0.560±0.009 | 0.787±0.014 | 0.259±0.004 | 0.625±0.011 |
| kimi | roberta-base+mobilenet_v3_small | 0.000±0.000 | 0.000±0.000 | 0.000±0.000 | 0.000±0.000 |
| mistral | albert-base-v2+vgg16 | 0.500±0.009 | 0.525±0.010 | 0.000±0.000 | 0.500±0.010 |
| mistral | albert-base-v2+efficientnet_b0 | 0.000±0.000 | 0.000±0.000 | 0.000±0.000 | 0.000±0.000 |
| mistral | albert-base-v2+efficientnet_b0 | 0.000±0.000 | 0.000±0.000 | 0.000±0.000 | 0.000±0.000 |
| kimi | roberta-base+mobilenet_v3_small | 0.000±0.000 | 0.000±0.000 | 0.000±0.000 | 0.000±0.000 |
| mistral | bert-base-uncased+shufflenet_v2_x1_0 | 0.516±0.009 | 0.611±0.010 | 0.147±0.002 | 0.531±0.010 |
| mistral | albert-base-v2+mobilenet_v3_small | 0.000±0.000 | 0.000±0.000 | 0.000±0.000 | 0.000±0.000 |
| mistral | albert-base-v2+mobilenet_v3_small | 0.000±0.000 | 0.000±0.000 | 0.000±0.000 | 0.000±0.000 |
| mistral | albert-base-v2+shufflenet_v2_x1_0 | 0.565±0.010 | 0.699±0.010 | 0.274±0.005 | 0.641±0.008 |
| mistral | bert-base-uncased+mobilenet_v3_small | 0.000±0.000 | 0.000±0.000 | 0.000±0.000 | 0.000±0.000 |
| qwen | albert-base-v2+vgg16 | 0.000±0.000 | 0.000±0.000 | 0.000±0.000 | 0.000±0.000 |
| mistral | albert-base-v2+vgg16 | 0.520±0.007 | 0.637±0.011 | 0.162±0.002 | 0.578±0.011 |
| mistral | albert-base-v2+vgg16 | 0.000±0.000 | 0.000±0.000 | 0.000±0.000 | 0.000±0.000 |
| mistral | albert-base-v2+vgg16 | 0.000±0.000 | 0.000±0.000 | 0.000±0.000 | 0.000±0.000 |
| mistral | bert-base-uncased+mobilenet_v3_small | 0.000±0.000 | 0.000±0.000 | 0.000±0.000 | 0.000±0.000 |
| kimi | roberta-base+efficientnet_b0 | 0.000±0.000 | 0.000±0.000 | 0.000±0.000 | 0.000±0.000 |
| qwen | bert-base-uncased+resnet18 | 0.579±0.009 | 0.740±0.013 | 0.325±0.005 | 0.672±0.010 |
| qwen | distilbert-base-uncased+densenet121 | 0.400±0.005 | 0.738±0.013 | 0.226±0.003 | 0.594±0.010 |
| qwen | distilbert-base-uncased+resnet50 | 0.400±0.006 | 0.672±0.013 | 0.226±0.004 | 0.594±0.011 |
| mistral | distilbert-base-uncased+shufflenet_v2_x1_0 | 0.400±0.006 | 0.521±0.009 | 0.226±0.003 | 0.594±0.011 |
| mistral | albert-base-v2+shufflenet_v2_x1_0 | 0.455±0.008 | 0.570±0.008 | 0.401±0.007 | 0.641±0.010 |
| qwen | roberta-base+shufflenet_v2_x1_0 | 0.348±0.005 | 0.602±0.009 | 0.209±0.003 | 0.578±0.009 |
| kimi | distilbert-base-uncased+mobilenet_v3_small | 0.364±0.007 | 0.762±0.011 | 0.267±0.005 | 0.594±0.009 |
| mistral | bert-base-uncased+mobilenet_v3_small | 0.364±0.007 | 0.777±0.012 | 0.267±0.005 | 0.594±0.008 |
| qwen | bert-base-uncased+mobilenet_v3_small | 0.381±0.007 | 0.779±0.014 | 0.338±0.006 | 0.609±0.010 |
| kimi | distilbert-base-uncased+vgg16 | 0.300±0.005 | 0.600±0.011 | 0.267±0.004 | 0.578±0.009 |

*(end of table)*

# B   REPRODUCIBILITY CHECKLIST

## REPRODUCIBILITY CHECKLIST

---

**Instructions for Authors:**

This document outlines key aspects for assessing reproducibility. Please provide your input by editing this `.tex` file directly.

For each question (that applies), replace the "Type your response here" text with your answer.

**Example:** If a question appears as

```
\question{Proofs of all novel claims are included}
{(yes/partial/no)}
Type your response here
```

you would change it to:

```
\question{Proofs of all novel claims are included}
{(yes/partial/no)}
yes
```

Please make sure to:

- Replace ONLY the "Type your response here" text and nothing else.

- Use one of the options listed for that question (e.g., **yes**, **no**, **partial**, or **NA**).

- **Not** modify any other part of the `\question` command or any other lines in this document.

You can `\input` this .tex file right before `\end{document}` of your main file or compile it as a stand-alone document. Check the instructions on your conference's website to see if you will be asked to provide this checklist with your paper or separately.

---

### 1. General Paper Structure

1.1. Includes a conceptual outline and/or pseudocode description of AI methods introduced (yes/partial/no/NA) Yes.

1.2. Clearly delineates statements that are opinions, hypothesis, and speculation from objective facts and results (yes/no) Yes.

1.3. Provides well-marked pedagogical references for less-familiar readers to gain background necessary to replicate the paper (yes/no) Yes.

### 2. Theoretical Contributions

2.1. Does this paper make theoretical contributions? (yes/no) No.

 If yes, please address the following points:

2.2. All assumptions and restrictions are stated clearly and formally (yes/partial/no) NA.

2.3. All novel claims are stated formally (e.g., in theorem statements) (yes/partial/no) NA.

2.4. Proofs of all novel claims are included (yes/partial/no) NA.

2.5. Proof sketches or intuitions are given for complex and/or novel results (yes/partial/no) NA.

2.6. Appropriate citations to theoretical tools used are given (yes/partial/no) NA.

2.7. All theoretical claims are demonstrated empirically to hold (yes/partial/no/NA) NA.

2.8. All experimental code used to eliminate or disprove claims is included (yes/no/NA) NA.

## 3. Dataset Usage

3.1. Does this paper rely on one or more datasets? (yes/no) Yes.

If yes, please address the following points:

3.2. A motivation is given for why the experiments are conducted on the selected datasets (yes/partial/no/NA) Yes.

3.3. All novel datasets introduced in this paper are included in a data appendix (yes/partial/no/NA) Yes.

3.4. All novel datasets introduced in this paper will be made publicly available upon publication of the paper with a license that allows free usage for research purposes (yes/partial/no/NA) Yes.

3.5. All datasets drawn from the existing literature (potentially including authors' own previously published work) are accompanied by appropriate citations (yes/no/NA) NA.

3.6. All datasets drawn from the existing literature (potentially including authors' own previously published work) are publicly available (yes/partial/no/NA) NA.

3.7. All datasets that are not publicly available are described in detail, with explanation why publicly available alternatives are not scientifically satisficing (yes/partial/no/NA) Yes.

## 4. Computational Experiments

4.1. Does this paper include computational experiments? (yes/no) Yes.

If yes, please address the following points:

4.2. This paper states the number and range of values tried per (hyper-) parameter during development of the paper, along with the criterion used for selecting the final parameter setting (yes/partial/no/NA) Yes.

4.3. Any code required for pre-processing data is included in the appendix (yes/partial/no) Yes.

4.4. All source code required for conducting and analyzing the experiments is included in a code appendix (yes/partial/no) Yes.

4.5. All source code required for conducting and analyzing the experiments will be made publicly available upon publication of the paper with a license that allows free usage for research purposes (yes/partial/no) Yes.

4.6. All source code implementing new methods have comments detailing the implementation, with references to the paper where each step comes from (yes/partial/no) Yes.

4.7. If an algorithm depends on randomness, then the method used for setting seeds is described in a way sufficient to allow replication of results (yes/partial/no/NA) NA.

4.8. This paper specifies the computing infrastructure used for running experiments (hardware and software), including GPU/CPU models; amount of memory; operating system; names and versions of relevant software libraries and frameworks (yes/partial/no) Yes.

4.9. This paper formally describes evaluation metrics used and explains the motivation for choosing these metrics (yes/partial/no) Yes.

4.10. This paper states the number of algorithm runs used to compute each reported result (yes/no) Yes.

4.11. Analysis of experiments goes beyond single-dimensional summaries of performance (e.g., average; median) to include measures of variation, confidence, or other distributional information (yes/no) Yes.

4.12. The significance of any improvement or decrease in performance is judged using appropriate statistical tests (e.g., Wilcoxon signed-rank) (yes/partial/no) Yes.

4.13. This paper lists all final (hyper-)parameters used for each model/algorithm in the paper's experiments (yes/partial/no/NA) Yes.

