# OpenReview forum: "From Lab to Line: Deployment-Aware NMR–Text Expert Routing for Real-Time Apple Moldy Core Disease Screening and Explanation"
_ICLR.cc/2026/Conference — ICLR 2026 Conference Withdrawn Submission_

### Official Review · Reviewer_SM2H · 2025-11-02

**Soundness:** 2
**Presentation:** 3
**Contribution:** 2
**Rating:** 2
**Confidence:** 3

**Summary:**

This paper proposes a deployment-aware multimodal framework for real-time diagnosis of Apple Moldy Core Disease (AMCD).
The authors introduce a new dataset, AppleNMR-MM V1.0, which combines low-field NMR imaging with expert textual annotations to enable multimodal learning under a small-sample regime.

The proposed system integrates Mixture-of-Experts (MoE) and Chain-of-Thought (CoT) reasoning techniques, together with a deployment-oriented metric (TAAPM) that reflects both economic and operational constraints.
Experiments and discussions are conducted with consideration of real-world deployment scenarios.

Overall, the paper presents a coherent and well-engineered system addressing a practical industrial problem. However, the methodological novelty appears limited, and several concerns remain regarding the empirical foundations and the realism of the deployment setup.

**Strengths:**

### **1. Practical Motivation**

The paper addresses a realistic and underexplored industrial problem—real-time diagnosis of Apple Moldy Core Disease (AMCD)—which has clear practical and economic significance.

---

### **2. System-Level Design**

The framework integrates multiple components—NMR imaging, textual expert annotations, a task-aware mixture-of-experts fusion module, and a multi-agent RAG-based reasoning layer—into a cohesive pipeline.
The overall system design is coherent, and the engineering quality appears solid.

---

### **3. New Dataset and Metric**

The introduction of the AppleNMR-MM V1.0 dataset and the TAAPM metric represents meaningful contributions.
TAAPM reflects an interesting attempt to formalize deployment trade-offs among accuracy, latency, and operational profit.

**Weaknesses:**

### **1. Limited Data Scale and Statistical Validity**

The dataset (237 samples, 79 diseased) is extremely small for a multimodal deep learning study, raising concerns about overfitting and statistical reliability.
The significance of reported improvements remains uncertain given the limited sample size.

---

### **2. Lack of Real Deployment or Field Validation**

Although the paper emphasizes “from lab to line” deployment, all experiments appear to be conducted under controlled laboratory settings.   Figure 2 represents a conceptual system architecture rather than a real industrial setup.
The system’s robustness under realistic production variability—such as fruit orientation, size, or hardware calibration—has not been demonstrated.

---

### **3. Limited Generalization and Scalability**

The framework is highly specialized for Apple Moldy Core Disease, with no evidence of transferability to other fruits, crops, or broader NMR-based inspection tasks.
This narrow domain focus limits the general impact and scalability of the approach.

---

### **4. Heuristic Nature of TAAPM**

While TAAPM is conceptually interesting, attempting to bridge predictive performance and deployment economics, it remains a handcrafted profit-based metric.
It is unclear whether optimizing TAAPM leads to consistent performance improvements or stable trade-offs in practice.

**Questions:**

### **1. Data and Empirical Foundation**

The AppleNMR-MM V1.0 dataset includes only 237 samples (79 diseased), which is extremely small for a multimodal deep learning study.
It remains unclear how the authors prevent overfitting under such limited data conditions.

> **Questions:**
> - Was any external dataset or cross-domain validation used to assess generalization?

---

### **2. Deployment Realism and Robustness**

While the paper emphasizes “deployment-aware” optimization, all reported results appear to come from laboratory simulations rather than real production data.
This raises concerns about the framework’s robustness in practical settings, where factors such as fruit size, rotation angle, and hardware calibration may vary substantially.

> **Questions:**
> - Has the system been tested under real conveyor-line or field conditions?
> - Does it require recalibration or fine-tuning for different batches or devices?
> - Given its strong task-specific design, can the framework generalize to other internal fruit disorders or related NMR-based inspection tasks?

---

### **3. Mixture-of-Experts (T-MoE) Novelty**

The proposed T-MoE module seems to follow standard gating and early-fusion mechanisms.
The distinction between this design and existing dynamic fusion or ensemble architectures remains unclear.

> **Questions:**
> - What concretely differentiates T-MoE from prior MoE-based fusion models?
> - Could the observed performance gains primarily result from ensembling effects rather than genuine expert specialization?

---

### Official Review · Reviewer_VheV · 2025-11-04

**Soundness:** 2
**Presentation:** 1
**Contribution:** 2
**Rating:** 2
**Confidence:** 3

**Summary:**

This paper proposes AppleNMR-MM, a comprehensive system for the real-time, interpretable diagnosis of Apple Moldy Core Disease (AMCD, a latent internal disease). It built a new small multimodal dataset (237 apples, 79 diseased) combining low-field Nuclear Magnetic Resonance (LF-NMR) images with expert-generated textual descriptions.

AppleNMR-MM is the core of the system. It is a multimodal fusion model for that leverages both the NMR images (e.g., via ResNet) and the textual data (e.g., via BERT) for classification. A novel task-aware evaluation metric, Task-Aware Apple Pathology Metric (TAAPM) is proposed, which moves beyond simple accuracy to model the real-world economic profit, quality-control constraints, and operational cost. Another key module for explanation is an "NMR-Apple-Expert" system. It is a dialogue agent built on a Mixture-of-Experts (MoE) architecture that uses Retrieval-Augmented Generation (RAG) and a Multi-agent Collaborative Chain-of-Thought (MACCT) framework to provide evidence-based explanations for its diagnoses.

Experiments show that the multimodal approach outperforms unimodal baselines, TAAPM is effective for model selection under production constraints, and the dialogue agent achieves high-quality for AMCD diagnosis interpretability.

**Strengths:**

1. The paper focuses on "Lab to Line", which has a high impact on real-world applications. The entire research effort is clearly motivated by the constraints and goals of a real-world industrial deployment. The "Deployment-Aware" emphasis that the ultimate goal of this paper is to build AI models is to solve real-world problems, that is to say, reducing post-harvest food loss through non-destructive screening.

2. The Task-Aware Apple Pathology Metric (TAAPM) is a standout contribution. Formulating a custom evaluation metric that directly reflects the economic utility and operational constraints of the target environment is a fantastic example of deployment-aware AI. It provides a much more meaningful way to select a model (e.g., based on production profits, efficiency, and costs) than merely relying on standard metrics like F1 or AUC.

3. Though in a small-scale, this multimodal dataset is useful. Combining a non-standard imaging modality like LF-NMR with structured expert text provides a unique resource for the community interested in multimodal learning for specialized domains like healthcare.

**Weaknesses:**

1. The keywords on the abstract and title are not clearly demonstrated in the main paper. For example, in the abstract, it claims "a Task-Aware Mixture-of-Experts (T-MoE) function is proposed to route among NMR and text experts conditional on predictive uncertainty and compute budget...", but how to compute "uncertainty" and what is the "expert routing"? The experiment shows a series of combinations of models, which is more likely a Model Selection rather than Expert Routing. Also, the detailed description of"Multi-agent Collaborative Chain-of-Thought (MACCT) with retrieval-augmented generation (RAG)" is missing. It seems a fine-tuned Qwen3-4B works as *triage -> diagnose -> explain*, but then where is the Multi-agent Collaboration? Such vague explanation cannot support the key contribution claimed on the abstract.

2. The paper has two key modules, a classification module (AppleNMR-MM, in Section 3.2) and an interpretability module (NMR-Apple-Expert, in Section 4.4), however, the connection between these two is not clearly articulated. Is the dialogue agent's diagnosis and explanation explicitly conditioned on the output of the classifier chosen by TAAPM? Or are they two separate systems that both perform diagnosis? The evaluation of the dialogue agent (Table 6) is based on human ratings of dialogue quality, which seems independent of the quantitative evaluation of the classifiers (Tables 1-4).

3. The experiments compare AppleNMR-MM with single-modality baselines, but do not include any other multimodal baselines. Tables 3-4 can only show that the proposed multimodal model (e.g., roberta-base + resnet18 + Gated Fusion) achieves better results than text-only/vision-only results in Tables 1-2. Some classic and advanced multimodal models (e.g., ViLT[1]) are recommanded to compare.

4. The dataset is in a very small-scale from the single site. While the difficulty of data collection is acknowledged, this sample size raises concerns about the statistical significance and generalizability of the findings. More cross-site or cross-season validation is needed.

5. The advantage of TAAPM requires more analysis. TAAPM is an attractive idea, but it depends on business parameters (e.g.,  required delivery quantity $Q$, defect tolerance threshold $\alpha$). The appendix includes sensitivity plots, yet the main text should more directly show how model selection under TAAPM compares to selection by AUC/F1 in terms of real-world outcomes. Providing an explicit example is also helpful. For instance, a case where model A (higher AUC) is not chosen by TAAPM while model B (lower AUC but better latency/params) is chosen.

6. The presentations of this paper can be polished.
(1) Some key words and key contributions require clearly demonstrations in the paper;
(2) The motivation and industry impact could be emphasized in the introduction;
(3) More figures of pipelines could be added to introduce the framework. For instance, the system overview part in Section 4.4 is too simple, and a figure is helpful to understand the pipeline;
(4) The structure of the main paper could be optimized. For example, before introducing the details of the classifier in Section 3.2 and the agent in Section 4.4, a summary paragraph to outline these two key parts would be helpful.
(5) The key insignts of experiments could be highlighted.

[1] ViLT: Vision-and-Language Transformer Without Convolution or Region Supervision

**Questions:**

1. In Multi-agent Collaborative Chain-of-Thought (MACCT), could you provide a more concrete example from your system of how the triage -> diagnose -> explain agents would interact to answer a complex query?

2. Given the small dataset, how can we be confident that the performance gains of the multimodal models are statistically significant?

3. GPT-Score in Table 6 requires clearly definition. What is the prompt to define this score based on the four dimensions (i.e., contextual coherence, factual consistency, domain-specific relevance and language fluency) mentioned in the paper?

4. The caption of figures in the Appendix are too simple and in the wrong format. More description could be added for the figure caption. For instance, the caption of Figure S6 only says "TAAPM Sensitivity analysis"; more explanation of each sub-figure and the key points of sensitivity analysis are recommended to report in the caption. Besides, the figure caption is supposed to be lower case (except for first word and proper nouns).

5. A minor point on presentation: Please define the acronym like TAAPM by providing its full name on its first use in the abstract.

---

### Official Review · Reviewer_E4SR · 2025-11-04

**Soundness:** 2
**Presentation:** 2
**Contribution:** 2
**Rating:** 6
**Confidence:** 3

**Summary:**

This paper presents a practical multimodal system (AppleNMR-MM) that combines low-field NMR imaging, vision models, and expert textual knowledge to detect internal Apple Moldy Core Disease (AMCD) under real-line constraints. Contributions include: (i) a small LF-NMR multimodal dataset (AppleNMR-MM V1.0, n=237 apples, 79 diseased); (ii) a Task-Aware Mixture-of-Experts (T-MoE) routing scheme that trades off uncertainty and compute budget for in-line inference; (iii) a Multi-Agent Collaborative Chain-of-Thought (MACCT) pipeline with RAG to produce evidence-grounded explanations and expert Q&A; and (iv) a deployment-centric metric, TAAPM, that encodes business/procurement constraints into model selection. Empirical results show multimodal fusion (text+NMR) improves AUC/F1 over unimodal baselines and the RAG explainer attains high expert pass rates (≈89–92%) on held-out dialogue samples.

**Strengths:**

1) The paper is strongly motivated by a concrete industrial problem (AMCD screening) and stresses operational constraints (throughput, latency, cost). TAAPM is an interesting attempt to formalize business-aware model selection.

2) Combining LF-NMR (T2-TSE scans) with textual expert knowledge and lightweight fusion/backbones is novel in this specific domain and valuable for food-safety applications. The AppleNMR-MM dataset itself (though small) is a useful contribution.

3) The AppleNMR preprocessor (YOLOv9-S), T-MoE routing, and MACCT+RAG expert system reflect a careful engineering effort oriented toward production deployment. The paper reports latency/parameter tradeoffs and compares many backbones and fusion strategies, which is useful to practitioners

4) Including human expert pass rates and GPT-based scoring for the dialogue/explanation component strengthens the claim that the system yields interpretable, evidence-grounded outputs.

**Weaknesses:**

1) The core supervised dataset contains 237 apples with 79 diseased samples; models and architectures are trained and evaluated routinely with many configurations (384 configs claimed). Such a small corpus raises strong concerns about overfitting and result stability. Although the paper reports repeated runs, I would like to see stronger statistical treatment (mean ± std for all main results), confidence intervals, and, ideally, evaluation on an external or cross-domain test set (e.g., apples from another orchard, season, cultivar, or a small held-out origin) to demonstrate generalization beyond the collection site.

2) The expert evaluation for dialogue/explanation uses 100 held-out samples with binary pass/fail. While promising (89% pass), the sample size is small and inter-annotator agreement is not reported

3) TAAPM is an interesting, deployment-aware metric, but it is highly dependent on chosen business parameters (Q, psell, cbuy, cextra, α, rbad). The paper references Appendix sensitivity analysis (Fig. S6) but the main text overinterprets TAAPM scores as absolute indicators of deployment-readiness. Please:

(a) Expand the sensitivity analysis in the main paper (or add a compact table) showing how TAAPM ranking of top models changes under realistic variations in defect rate rbad, penalty thresholds α, and cost parameters.

(b) Report not only TAAPM values but also the underlying standard metrics (AUC, F1, latency) for the top TAAPM models so readers understand the tradeoffs. (Some of this is present but better integration is needed.)

**Questions:**

1) The authors should:

a) Report mean ± std across independent seeds for all primary metrics (AUC, F1, TAAPM) and perform statistical tests where appropriate.

b) Provide clear train/validation/test split protocol (how many folds, stratification by batch/scan session, whether slices are grouped per apple) and evidence that no leakage exists (e.g., same-apple slices not split across train/test).

c) If possible, add a small external validation (even a few dozen apples from a different harvest or location) or leave-one-batch-out cross-validation to show robustness.

2) Supply the LLM prompt templates used to generate the textual descriptions and dialogues, along with a few raw (pre-expert) and final (post-expert) examples.

---

### Official Review · Reviewer_JDZX · 2025-11-05

**Soundness:** 1
**Presentation:** 1
**Contribution:** 2
**Rating:** 2
**Confidence:** 3

**Summary:**

This paper proposes to tackle the problem of Apple Moldy Core Disease (AMCD) by leveraging both text descriptions and images of affected apples. The authors propose a multi-modal fusion based approach for combining text descriptions of the apples, produced by various LLMs, and images of the cores of the apples. The authors also propose a new metric for evaluating their model, TAAPM, while also planning to release the dataset used to generate these results.

**Strengths:**

The application area is novel, I had never heard of AMCD disease before reading this paper. It seems like a very important problem to enable better utilization of batches of apples for transport & selling.

**Weaknesses:**

1. The writing of this paper needs improvement.

a. From the start of the paper, there are many different acronyms that are not defined which makes the understanding of this paper very difficult. For example what do the letters after Apple stand for here: AppleNMR-MM ?

b. The abstract of this paper seemingly just lists the results section of the submitted manuscript which does not summarize and briefly contextualize the work for the reader.

c. The results (in section 4.2 for example) shouldn't include the various metrics in the text (i.e. line 305: 'with an F1 score of 0.786 ± 0.015, AUC of 0.805 ± 0.019, MCC of...'). Please just refer the reader to the table/figure where the results are. It clouds the writing and makes it harder to understand.

d. For the tables of results, what does bolding the results mean? Are these results statistically significant from one another? How are the results compared?

2. From the submitted manuscript, it is not clear why the authors are tackling this problem. In the related works section, there are multiple instances of previous works leveraging images only that produce an accuracy of >96%. The proposed method doesn't reach that level while also not making the case for a multi-modal method clear.

3. In the related works section, there are several works that are mentioned that seem to have high performance on the same task. These works should be used as baselines, but results from these works are not included in the experimental results section of the submitted work.

4.  I don't see the importance of the TAAMP metric. If we are to train a model for a classification task, then we generally already have access to the various individual components of the TAAMP metric. Why do we need a unified metric?

5. Line 281: "to ensure experimental reproducibility, all random number generators were seeded with a fixed value of 42." can you explain what this means? Are the authors saying that they would seed random number generators (i.e. torch.manual_seed(42)) for each training run of their method? Does this mean that any variation in the methods is from CUDA non-determinism and not from training the models using different random initializations?

6. The figures in the appendix (i.e. S7) should likely be in the main text, while some of the details of the dataset acquisition and TAAMP could likely be moved to the appendix. In addition, the main figure that summarizes the method (Figure 2) is very hard to read because the text is so small.

Minor Issues
Appendix overload: 51 pages of supplementary tables that could be summarized


In summary, I think that the authors need to (a) motivate the problem better (why multi-modal? How have previous works approached this problem?), (b) improve the writing quality, (c) communicate the results in a more understandable fashion, and (d) ensure that the results are statistically significant across multiple random seeds.

**Questions:**

1. From the past 10-15 years of machine learning research, we have learned about the importance of lots of data for training ML models. The dataset that the authors use for this task is very small (n=237 samples). How does the small dataset size affect the performance of the models here? What prevents overfitting with many different configurations on 237 samples?

2. Why is a multi-modal model needed for this task?

3. What do human experts achieve in this task?

---

### Official Review · Reviewer_U1E9 · 2025-11-08

**Soundness:** 2
**Presentation:** 3
**Contribution:** 2
**Rating:** 4
**Confidence:** 4

**Summary:**

The paper introduces a small vision–language dataset (AppleNMR-MM V1.0) of apple diagnoses curated by domain experts, combining LF-NMR images and textual descriptions. Using this dataset, the authors trained and compared text-only, vision-only, and multimodal models with different fusion strategies, showing that multimodal fusion yields superior performance. The study includes a detailed ablation analysis across modalities and architectures, proposes a task-aware evaluation metric (TAAPM) that links diagnostic accuracy with deployment efficiency, and demonstrates an expert-in-the-loop dialogue system for explainable, real-time apple quality screening.

**Strengths:**

(1) The paper includes a strong and comprehensive ablation study covering different modalities, model architectures, and fusion strategies, effectively demonstrating robustness and performance under various conditions and providing justification for the overall pipeline design.

(2) The analysis of the presented results is detailed and thorough, providing meaningful insights into model behavior and comparative performance.

(3) The paper introduces a new evaluation metric that is task-aware and closely aligned with the practical context and objectives of the study.

**Weaknesses:**

(1) The introduction focuses mainly on the importance of apples, AMCD, and disease detection. Since this is an ICLR paper, it should emphasize machine learning aspects more and reduce the agricultural or biological context. One paragraph on ML methods is not sufficient.

(2) he paper’s contributions are not substantial—limited mainly to a small dataset creation that is insufficient in scale to meaningfully impact the large models used in the experiments—and they are not clearly articulated in the introduction.

(3) The flowchart of the model—especially the architecture of multimodal fusion and how it integrates with other feature extraction modules—is confusing. It would be clearer if the diagram included explicit input–output arrows to show the connections.

(4) Although the paper text is coherent, the abstract and the method names mentioned there appear somewhat inconsistent. In particular, the framework does not seem to implement a real mixture of experts (the term is never discussed in the paper). It only fuses two modalities, and within each modality, there is no more than one model or core architecture. There is an important difference between a “human expert” and an “expert” in an MoE architecture. The models are merely experimented with separately. Furthermore, the so-called multi-agent chain-of-thought is neither discussed nor shown—where exactly are the “multi-agent” and “chain-of-thought” components?

(5) The phrase “conditioned on predictive uncertainty and compute budget” is also unsupported. This conditional property is never discussed or backed by any theoretical or experimental evidence.

**Questions:**

(1) In the main results section, the authors state that different models—text-only, vision-only, and multimodal with fusion—were trained using both pretrained and scratch initialization on the proposed dataset (237 images with corresponding text descriptions) and report good performance. However, it is unclear whether such a small dataset is sufficient for training—or even fine-tuning—models of this scale, including:
- four text encoders (~12–125M parameters)
- eight vision backbones (~2–138M)
- several fused multimodal variants (~13–136M)
- one dialogue model (~4B)
It would also be important to compare these results with zero-shot evaluations (i.e., before any training) to better understand how much actual learning occurred from such limited data.

(2) The section titled “Different Fusion Methods” appears to be more of an ablation study on various fusion strategies and would fit better under an Ablation Study section.

---

### Note · Authors · 2026-05-28

I have read and agree with the venue's withdrawal policy on behalf of myself and my co-authors.

---

### Meta-Review · Area_Chair_b9BP · 2026-01-09

**Summary:**

While the paper is well-motivated by a real-world industrial problem and includes a careful engineering effort with extensive ablations, it falls short in the current form along several dimensions:  (i) the claimed methodological contributions (task-aware MoE routing, multi-agent chain-of-thought, uncertainty-conditioned fusion) are either standard, weakly specified, or not empirically demonstrated; (ii) the dataset is extremely small (237 samples from a single site),  (iii) empirical validation lacks strong multimodal baselines, ; and (v) presentation and clarity issues.

**Reviewer Scores:**

N/A

---

### Decision · Program_Chairs · 2026-01-26

Reject